

# ORCHIDEE-MICT (revision 4126), a land surface model for the high-latitudes: model description and validation

Matthieu Guimberteau[1]*, Dan Zhu[1]*, Fabienne Maignan[1], Ye Huang[1], Chao Yue[1], Sarah Dantec-Nédélec[1], Catherine Ottlé[1], Albert Jornet-Puig[1], Ana Bastos[1], Pierre Laurent[1], Daniel Goll[1], Simon Bowring[1], Jinfeng Chang[2], Bertrand Guenet[1], Marwa Tifafi[1], Shushi Peng[1,3], Gerhard Krinner[4], Agnès Ducharne[5], Fuxing Wang[6], Tao Wang[7,8], Xuhui Wang[1,9], Yilong Wang[1], Zun Yin[1], Ronny Lauerwald[10,1,11], Emilie Joetzjer[1,12], Chunjing Qiu[1], Hyungjun Kim[13] and Philippe Ciais[1]

* These authors contributed equally to this work

[1] Laboratoire des Sciences du Climat et de l'Environnement, LSCE/IPSL, CEA - CNRS - UVSQ, Université Paris-Saclay, 91191 Gif-sur-Yvette, France

[2] Sorbonne Universités (UPMC), CNRS-IRD-MNHN, LOCEAN/IPSL, 4 place Jussieu, 75005 Paris, France

[3] Sino-French Institute for Earth System Science, College of Urban and Environmental Sciences, Peking University, Beijing 100871, China

[4] CNRS, Univ. Grenoble Alpes, Institut des Géosciences de l'Environnement (IGE), 38000 Grenoble, France

[5] UMR 7619 METIS, Sorbonne Universités, UPMC, CNRS, EPHE, 4 place Jussieu, 75005 Paris, France

[6] Laboratoire de Météorologie Dynamique, Ecole Polytechnique, F 91128 Palaiseau, France

[7] Key Laboratory of Alpine Ecology and Biodiversity, Institute of Tibetan Plateau Research, Chinese Academy of Sciences, Beijing 100085, China

[8] CAS Center for Excellence in Tibetan Plateau Earth Sciences, Chinese Academy of Sciences, Beijing 100085, China

[9] Laboratoire de Météorologie Dynamique, Université Pierre et Marie Curie, 75005 Paris, France

[10] Université Libre de Bruxelles, Belgium

[11] University of Exeter, Exeter, United Kingdom

[12] CNRS, Université Paul Sabatier, ENFA; UMR5174 EDB (Laboratoire Evolution et Diversité Biologique), 118 route de Narbonne, Toulouse F-31062, France

[13] Institute of Industrial Science, The University of Tokyo, Tokyo, Japan

**Abstract.** The high-latitude regions of the northern hemisphere are a nexus for the interaction between land surface physical properties and their exchange of carbon and energy with the atmosphere. At these latitudes, two carbon pools of planetary significance – those of the permanently frozen soils (permafrost), and of the great expanse of boreal forest – are vulnerable

5    to destabilization in the face of currently observed climatic warming, the speed and intensity of which are expected to increase with time. Improved projections of future Arctic and boreal ecosystem transformation require improved land surface models that integrate processes specific to these cold biomes. To this end, this study lays out relevant new parameterizations



in the ORCHIDEE-MICT land surface model. These describe the interactions between soil carbon, soil temperature and hydrology, and their resulting feedbacks on water and $CO_2$ fluxes, in addition to a recently-developed fire module. Outputs from ORCHIDEE-MICT, when forced by two climate input data sets, are extensively evaluated against: (i) temperature gradients between the atmosphere and deep soils; (ii) the hydrological components comprising the water balance of the largest high-latitude basins, and (iii) $CO_2$ flux and carbon stock observations. The model performance is good with respect to empirical data, despite a simulated excessive plant water stress and a positive land surface temperature bias. In addition, acute model sensitivity to the choice of input forcing data suggests that the calibration of model parameters is strongly forcing-dependent. Overall, we suggest that this new model design is at the forefront of current efforts to reliably estimate future perturbations to the high-latitude terrestrial environment.

## 1   Introduction

In the high latitudes, the complex coupling between soil thermal and hydraulic processes, snowpack properties, and plant and soil carbon pools is of great importance. Snow accumulation and freezing of soil water lead to a net storage of water from October to April. Through the processes of snow melt and the onset of soil thaw in spring, water is made available for plant uptake and growth. Simultaneously however, much of this is 'lost' as runoff to rivers, causing peak discharge rates in May-June (Yang et al., 2003) and the flooding of large flatland areas from May to September (Papa et al., 2008; Biancamaria et al., 2009). In summertime, the peak in incident solar radiation causes a temperature maximum that increases water evaporative demand on the land surface. Many boreal and Arctic regions thus have a negative water balance in summer (Schulze et al., 1999), which may impose powerful constraints on plant growth. Siberian and Canadian boreal forests have thus been shown to experience water stress, with ratios of surface sensible to latent heat flux of up to ~2 (Jarvis et al., 1997; Baldocchi et al., 1997; Schulze et al., 1999) causing further heating of the near-surface atmosphere.

These large seasonal shifts of the high-latitude water balance – how water input from precipitation is shared between changes of water storage in snow, ice and soil moisture, and balanced against losses from evapotranspiration, sublimation and river discharge – can now be better assessed and evaluated using state-of-the-art observation datasets. In addition, in terms of realistic process representation, land surface models (LSMs) focusing on high-latitude phenomena require the inclusion of the following non-exhaustive series of pivotal hydrological and biogeochemical interactions:

1- A representation of permafrost physics and seasonal freeze-thaw cycles, which determine the soil hydrologic and thermal budgets and the volume and timing of lateral water flows to rivers.

2- The impact of winter snow acting as an insulating 'barrier' between soils and overlying air from Autumn to early Spring. These have subsequent effects on soil temperature and water content, feeding back onto snow thickness itself.

3- The seasonal mediation of plant water availability via snowmelt water, transpiration losses and the depth of the permafrost table (active layer thickness), which in turn determine the availability of the lateral water flows that feed rivers in the warmer months.



4- The limitations on plant productivity and biomass due to acute climatic conditions in high-latitude regions. These primarily involve biotically prohibitive cold temperatures from Autumn to late Spring, low soil moisture in dry-summer regions, and fire events, caused by hot and dry conditions.

5- The buildup of large soil carbon stocks under cold conditions through the slow burial of organic matter in the permafrost via cryoturbation and sedimentary soil formation processes (e.g. Hugelius et al., 2013; Tarnocai et al., 2009).

6- Feedbacks between high soil carbon concentrations and profiles of soil temperature, water and permafrost carbon content (e.g. Lawrence and Slater, 2008; Decharme et al., 2016).

We represent the above processes in an updated version of the ORCHIDEE LSM (ORganizing Carbon and Hydrology in Dynamic EcosystEms), known as ORCHIDEE-MICT (aMeliorated Interactions between Carbon and Temperature), which we describe in this study. Since the comprehensive description of the model ORCHIDEE by Krinner et al. (2005), the model went through major modifications and improvements; we present here the major ones linked to high-latitude processes. ORCHIDEE-MICT is evaluated over the last two to three decades (depending on the variable) against empirically generated datasets. Against these, we are able to evaluate model performance regarding the distribution of permafrost and the effect of snow on soil thermics (mechanisms 1 and 2); the different components of the water cycle over a wide range of high-latitude basins (mechanism 3); plant primary productivity as constrained by high-latitude conditions (mechanisms 3 and 4), and replication of soil carbon stocks and feedback dynamics (mechanisms 5 and 6).

## 2 ORCHIDEE model overview

The starting point for our updated land surface model (ORCHIDEE-MICT) is ORCHIDEE-TRUNK revision 3976. As detailed in Section 3, its description of soil temperature and vertical water transport dynamics is based on coupled diffusion equations with identical vertical discretization (Wang et al., 2016a), and includes soil freezing, its effect on water infiltration, and phase change-induced heat sources and sinks in the soil column (Gouttevin et al., 2012b). The snow model described by Wang et al. (2013) is incorporated in this version, where snow is discretized into three layers of variable thickness, conductivity and density, accounting for snow liquid water content (Boone and Etchevers, 2001). In terms of large-scale hydrology, a river routing scheme including floodplains and their dynamics (d'Orgeval et al., 2008; Guimberteau et al., 2012) is coupled to simulated grid-cell runoff (Section 3.2), permitting the calculation of 'natural' river discharge (i.e. in the absence of dams or human water withdrawals).

The carbon cycle model includes half-hourly photosynthesis (GPP), daily allocation of GPP assimilates to autotrophic respiration and 8 plant biomass pools (foliage, roots, above/below-ground sapwood and heartwood, fruits and carbon reserves), and prognostic phenology (Botta et al., 2000). These pools are characterized by different turnover times, mortality rates and subsequent litter and soil carbon decomposition rates. Litter carbon is funneled between structural and metabolic fractions, and soil carbon between active, slow and passive pools, following Parton et al. (1987).

The model divides vegetation into 13 plant functional types (PFTs). Each PFT follows the same suite of equations but with PFT-specific parameter values and phenology functions (Krinner et al., 2005). PFT fractions are assigned to three soil tiles





corresponding to bare soil, short vegetation (grass and crop PFTs) and forests (all tree PFTs). The soil moisture budget of each soil tile is calculated separately, but different PFTs in the same soil tile interact as they share the same soil moisture source. While transpiration is calculated separately for each PFT, and soil moisture for each soil tile, the energy budget of a grid cell with multiple PFTs is calculated using the area-weighted average of those PFTs. This in turn defines mean grid-cell

land surface temperature, giving the upper boundary condition for the vertically discretized soil thermal module.

Temperature, water and carbon interactions described in ORCHIDEE revision 3976 are summarized in Fig. 1 by the black arrows. Air temperature and humidity impact phenology, photosynthesis, autotrophic respiration and the water and heat fluxes comprising the surface energy budget. Soil moisture in the root zone modulates photosynthesis and transpiration, which depends on wilting point and field capacity. In ORCHIDEE revision 3976, while soil carbon decomposition is impacted by soil

water and temperature, soil carbon stocks themselves exert no feedback on the soil physical state.

The key model developments presented here in ORCHIDEE-MICT (revision 4126) thus include the feedback effects of soil organic carbon (SOC) concentration on both soil thermic and soil water dynamics (Fig. 1, red arrows). Because these SOC-affected soil physics alter the above and below-ground components of the carbon cycle, as well as plant transpiration via hydraulic stress, we can expect complex indirect effects on the energy, water and carbon budgets (Fig. 1). Note that several

other updates were implemented in ORCHIDEE-TRUNK (revision 3976) and passed to ORCHIDEE-MICT (revision 4126), including a revised background albedo based on satellite observations, and updates of photosynthesis scheme. These will be described in an upcoming paper for the ORCHIDEE-TRUNK (version close to revision 3976) that will be used for CMIP6 exercise.

In the following, we describe the parameterizations that define ORCHIDEE-MICT (revision 4126). Section 3 briefly sum-

marizes snow and soil freeze developments (Gouttevin et al., 2012a; Wang et al., 2013) that are incorporated already in ORCHIDEE-TRUNK revision 3976. In Section 4, we focus on new features of ORCHIDEE-MICT compared to revision 3976, including SOC vertical discretization and processes leading to SOC burial in frozen horizons (Section 4.1), and functions describing how the SOC content in each soil layer alters soil thermal and hydraulic properties (Section 4.2). The simulation protocol, forcing data and different observations used to evaluate ORCHIDEE-MICT are described in Section 5. Model out-

puts are evaluated against multiple observations for grid-based simulations forced by two climate datasets, CRUNCEP v7 and Global Soil Wetness Project Phase 3 (GSWP3) v0, at 1° resolution over the Northern Hemisphere over periods within the range 1960-2007. In the result sections, Section 6 evaluates snow depth and soil temperature against measurements from in-situ stations, and land skin temperature against satellite-derived dataset. Section 7 evaluates modeled components of the water budget and their sum, using remote-sensing based observations of total land water storage changes, snow water equivalent, and evap-

otranspiration, combined with in-situ river discharge data, to close the water budget at the scale of large northern hemisphere catchments. Sections 8 and 9 evaluate carbon fluxes and stocks, using observation-based gridded GPP, fire emissions, forest biomass and SOC, together with site-based measurements of NPP (net primary production) and GPP, and regional net $CO_2$ fluxes from atmospheric inversions. In the discussion (Section 10), we address some of the critical issues in the model performances for processes related to seasonal changes in water budgets, thermal gradients from atmosphere to soils, and carbon

budgets.



## 3 High-latitude processes in the initial ORCHIDEE version

### 3.1 Soil freezing and snow processes

The soil freezing scheme developed by Gouttevin et al. (2012a) describes phase changes of soil water, simulating the latent heat exchanges involved in the freezing and melting of soil water, and subsequent changes in thermal and hydrological ground properties. Soil heat conductivity and heat capacity are dependent on soil ice content. The hydraulic conductivity of the soil is parameterized according to its liquid water content and decreases with the frozen soil fraction. Heat transfer through the soil column is represented by a one-dimensional heat conduction equation, with latent heat acting as a source or sink term (Gouttevin et al., 2012a), in the following function:

$$c \frac{\partial T}{\partial t} = \frac{\partial}{\partial z} \left( \lambda \frac{\partial T}{\partial z} \right) + \rho_{ice} L \frac{\partial \theta_{ice}}{\partial t} \tag{1}$$

where c is volumetric soil heat capacity ($J\,K^{-1}\,m^{-3}$); T is soil temperature (K); $\lambda$ is soil thermal conductivity ($J\,m^{-1}\,s^{-1}\,K^{-1}$); $\rho_{ice}$ is ice density ($kg\,m^{-3}$); L is latent heat of fusion ($J\,kg^{-1}$); $\theta_{ice}$ is volumetric ice content ($m^3\,m^{-3}$); t is time (s) and z is depth (m). In ORCHIDEE-MICT, this equation is discretized on the 32 vertical layers of the model with a total soil depth of 38 m (Fig. S1). Note that the soil hydrology has only 11 layers up to 2 m, so the volumetric contents of water and ice below 2 m take the values of the bottom layer (i.e. the 11[th] layer).

Snowpack is represented by a 3-layer snow model of intermediate complexity, as described in Wang et al. (2013). This scheme was implemented to resolve the energy and water budgets inside the snowpack, accounting for thawing and refreezing of liquid water. The snow model produces prognostic snow temperature, density and SWE for the 3 snow layers. Modifications were recently implemented to represent snowpack sub-grid scale variability. A snow cover fraction over the grid cell was introduced as a function of SWE. This was used for improving albedo and surface temperature estimates. Although this snow cover fraction was calculated for glacier and vegetated-surface areas separately, it is not dependent on the vegetation cover. Additional modifications were implemented (uniformization of the energy budget calculation, update of the snow-covered vegetation albedo...) and will be described in the upcoming CMIP6 ORCHIDEE paper as mentioned in Section 2.

### 3.2 Soil hydrology and river routing

ORCHIDEE simulates soil water fluxes and storage through a multi-layer soil hydrology scheme described by de Rosnay et al. (2000, 2002) and Campoy et al. (2013). Soil moisture is redistributed in the column by solving the Richards equation for vertical unsaturated flow under the effect of root uptake. The hydraulic conductivity and diffusivity depend on soil moisture, following the Mualem-van Genuchten model (Mualem, 1976; Van Genuchten, 1980), and using parameters defined by Carsel and Parrish (1988). These variables depend on the dominant soil texture in each grid-cell, based on the 12 USDA texture classes provided at the 0.08° resolution from Reynolds et al. (2000). For frozen soils, the decrease of the hydraulic conductivity (Gouttevin et al., 2012a) reduces infiltration into the soil and drainage, and increases surface runoff. The 2m soil column is divided into



11 layers, with layer thickness increasing geometrically with depth (Fig. S1). The saturated hydraulic conductivity is modified according to the scheme in d'Orgeval et al. (2008). This decreases exponentially below a top-30 centimeter depth boundary to account for increased soil compaction, as suggested by Beven and Germann (1982), and increases above that boundary towards the soil surface due to the enhanced infiltration capacity afforded by vegetative roots, whose presence increases soil porosity

in the root zone (Beven, 1984). The canopy throughfall rate and soil hydraulic conductivity govern the partitioning between surface runoff and soil infiltration. This partitioning involves a time-splitting procedure inspired by Green and Ampt (1911), describing the propagation of the wetting front. The second physical factor contributing to total runoff is free gravitational drainage at the bottom of the soil.

The runoff routing module (Polcher, 2003; Ngo-Duc et al., 2005; Guimberteau et al., 2012) aggregates surface runoff and
drainage produced at a 30-minute time step to calculate daily flow between grid cells and discharge to the ocean. Grid cells are subdivided into basins in which water is transferred through a series of linear reservoirs along the drainage network, derived from a 0.5 degree resolution data set (Vörösmarty et al., 2000; Oki et al., 1999). In a given basin, a 'slow' reservoir collects drainage water, while a 'fast' reservoir collects surface runoff, each with different linear response timescales. Corresponding outflows are transferred to the stream reservoir of the downstream basin. The process is fully detailed in Guimberteau et al.
15 (2012).

The routing scheme includes a parameterization of floodplains (d'Orgeval et al., 2008; Guimberteau et al., 2012), the maximum extent of which is prescribed by the GLWD (Global Lakes and Wetlands Database) map (Lehner and Döll, 2004). In grid-cells with flooded areas, river discharge from upstream basins is diverted to a floodplain reservoir, which then feeds a delayed return flow back to the stream reservoir of the basin.

## 4 New processes and parameterizations

### 4.1 Soil carbon discretization

In ORCHIDEE-MICT, the three soil carbon pools (active, slow and passive) share a common 32-layer discretization scheme as that of soil temperature, to a maximum depth of 38 m. Carbon inflows to the soil pools from decomposed litter are partitioned along this depth using an exponential function that corresponds to the prescribed PFT root profile. Decomposition of soil carbon
is calculated at each layer as a function of soil temperature, moisture, and texture (Koven et al., 2009; Zhu et al., 2016). Vertical mixing of soil carbon due to cryoturbation (mixing of soil layers induced by repeated freeze-thaw cycles) and bioturbation are accounted for by adding a diffusion term in the soil carbon equation:

$$\frac{\partial C_i(z,t)}{\partial t} = I_i(z,t) - g_i(z,t) C_i(z,t) + D \frac{\partial C_i^2(z,t)}{\partial z^2} \tag{2}$$

where $C_i(z,t)$ is carbon content of pool i at depth z and time t ($gC\,m^{-3}$); $I_i(z,t)$ is carbon input ($gC\,m^{-3}\,d^{-1}$); $g_i(z,t)$ is
decomposition rate ($d^{-1}$); D is diffusive mixing rate, set as $10^{-3}\,m^2\,yr^{-1}$ through the active layer and decreases linearly to zero





at 3 m in permafrost regions, to represent cryoturbative mixing (Koven et al., 2009), and set as $10^{-4}\,\mathrm{m^2\,yr^{-1}}$ above 2 m in non-permafrost regions to represent bioturbation (Koven et al., 2013).

## 4.2 SOM-dependent soil thermal and hydraulic parameters

Soil organic matter (SOM) significantly modifies soil thermal and hydraulic properties. SOM lowers thermal conductivity
and increases heat capacity (e.g. Lawrence and Slater, 2008; Decharme et al., 2016), and increases soil porosity, which in turn increase saturated hydraulic conductivity and available water capacity (e.g. Hudson, 1994; Morris et al., 2015). As a consequence, the presence of SOM modulates heat transfer from the surface through the soil column, typically leading to cooler soil temperature during summer. SOM-effected increases in soil water holding capacity also enhances plant available water and thus primary productivity (Krull et al., 2004) and transpiration. SOM impacts on soil thermics and hydraulics have
previously been parameterized in the global LSMs CLM (Lawrence and Slater, 2008), JULES (Chadburn et al., 2015) and ISBA (Decharme et al., 2016). In ORCHIDEE, SOM thermal insulation was previously investigated by Koven et al. (2009) but its parameterization was imbedded in a prior model version which employed bucket-type soil hydrology. This however, is not applicable to ORCHIDEE-MICT, which uses a new vertically discretized hydrology scheme and its coupling with the thermal module. In addition, the Koven et al. (2009) study did not include SOM effects on soil hydraulic properties, which are
addressed in ORCHIDEE-MICT and described in detail below.

**Thermal conductivity and heat capacity**

By default, soil thermal conductivity and heat capacity in ORCHIDEE are calculated in each soil layer as empirical functions of the 12 USDA soil texture classifications (see Table S1) and soil water and ice contents, following Wang et al. (2016a):

$$\lambda_i = Ke_i\,\lambda_{i,sat} + (1 - Ke_i)\,\lambda_{i,dry} \tag{3}$$

with:

$$\lambda_{i,sat} = \lambda_{i,solid}^{(1-\theta_{i,sat})}\,\lambda_{liq}^{\left(\theta_{i,sat}\frac{\theta_{i,liq}}{\theta_{i,liq}+\theta_{i,ice}}\right)}\,\lambda_{ice}^{\left(\theta_{i,sat}\frac{\theta_{i,ice}}{\theta_{i,liq}+\theta_{i,ice}}\right)} \tag{4}$$

$$c_i = c_{i,dry} + \theta_{i,liq}\,c_{liq} + \theta_{i,ice}\,c_{ice} \tag{5}$$

where $\lambda_{i,sat}$ and $\lambda_{i,dry}$ are saturated and dry thermal conductivities for layer i; $\lambda_{liq}$ and $\lambda_{ice}$ are thermal conductivities of liquid water and ice, equaling to 0.57 and 2.2 respectively ($\mathrm{W\,m^{-1}\,K^{-1}}$); $\lambda_{i,solid}$ is thermal conductivity of soil solids (see Table
S1); $c_{liq}$ and $c_{ice}$ are heat capacities of liquid water and ice, equaling to $4.18\,10^6$ and $2.11\,10^6$ respectively ($\mathrm{J\,K^{-1}\,m^{-3}}$); $c_{dry}$ is





dry soil heat capacity depending on soil texture; $\theta_{i,sat}$ is volumetric moisture content at saturation (porosity), and it varies with soil textures; $\theta_{i,liq}$ and $\theta_{i,ice}$ are prognostic volumetric liquid water and ice contents ($m^3\,m^{-3}$) that are computed by the soil hydrology model; $Ke_i$ is the Kersten number given by:

For unfrozen soils:

$$Ke_i = \begin{cases} \log_{10}(S_r) + 1 \\ 0.7\log_{10}(S_r) + 1 \quad \text{if} \\ 0 \end{cases} \begin{cases} S_r > 0.1 \\ 0.05 < S_r \leq 0.1 \\ S_r \leq 0.05 \end{cases} \tag{6}$$

with:

$$S_r = \frac{\theta_i}{\theta_{i,sat}} \tag{7}$$

For frozen soils:

$$Ke_i = S_r \tag{8}$$

where $S_r$ is the degree of saturation.

To account for the impacts of organic carbon on soil thermal properties in ORCHIDEE-MICT, we follow Lawrence and Slater (2008) in assuming that soil physical properties are weighted averages of mineral soil (as the default values in standard ORCHIDEE) and pure organic soil, with the organic soil fraction $f_{i,soc}$ calculated as:

$$f_{i,soc} = \min\left(1, \frac{\rho_{i,soc}}{\rho_{soc,max}}\right) \tag{9}$$

where $\rho_{i,soc}$ is the carbon content for layer i ($kgC\,m^{-3}$), derived from observation-based soil organic carbon map from NCSCD (Hugelius et al., 2013) in permafrost regions and from HWSD (FAO, 2012) in non-permafrost regions, after linear vertical interpolation from their original soil horizons to fit ORCHIDEE-MICT vertical layers; $\rho_{soc,max}$ equals to $130\,kgC\,m^{-3}$, a typical soil carbon density of peat (Lawrence and Slater, 2008). Therefore, the red-colored parameters in Eqs. 3-7 are calculated as:

$$P_i = \left(1 - f_{i,soc}\right) P_{mineral} + f_{i,soc}\, P_{soc} \tag{10}$$





where $P_i$ represents different properties $\lambda_{i,dry}$, $\lambda_{i,solid}$, $c_{i,dry}$, and $\theta_{i,sat}$. The values of $P_{mineral}$ for each soil texture and $P_{soc}$ are listed in Table S1.

**Available water capacity**

Plant available water capacity, defined as the difference in the amount of water held by each soil layer between field capacity ($\theta_{fc}$) and permanent wilting point ($\theta_{wp}$), determines the capacity of the soil to store and supply water for plants and is therefore an important aspect of soil fertility (Hudson, 1994). For mineral soils in ORCHIDEE, $\theta_{fc}$ and $\theta_{wp}$ are derived from measurements of the soil matric potential at field capacity and wilting point, based on the soil water retention curve described by van Genuchten equation (Van Genuchten, 1980):

$$\theta = \frac{(\theta_{sat} - \theta_r)}{\left[1 + (\alpha\,(-\Psi))^n\right]^{\left(1-\frac{1}{n}\right)}} + \theta_r \tag{11}$$

where $\psi$ is soil matric potential (kPa), and $\psi = -33\,\text{kPa}$ (or -10 kPa for the three sandy soils, see Table S2) corresponds to field capacity $\theta_{fc}$, while $\psi = -1500\,\text{kPa}$ corresponds to wilting point $\theta_{wp}$ for all textures; $\theta_r$ is the residual volumetric water content ($\text{m}^3\,\text{m}^{-3}$); $\alpha$ and n are empirical fitting coefficients, with their values for different soil textures listed in Table S2.

SOC has been shown to significantly increase water retention (Rawls et al., 2003). To parameterize this SOM effect, we assume that $\theta_r$ and the coefficients in Eq. 11 do not change with carbon content, while porosity $\theta_{sat}$ increases with organic carbon (Eq. 10). Therefore, both $\theta_{fc}$ and $\theta_{wp}$ increase under higher carbon contents, but $\theta_{fc}$ increases faster, resulting in a higher available water capacity (Fig. S2), consistent with the patterns observed in Hudson (1994).

### 4.3 Reformulation of soil hydric stress above the permafrost table

It is known that reduced soil moisture availability decreases the rate of photosynthesis, but the parameterization of this photosynthetic stress differs amongst models (Medlyn et al., 2015). ORCHIDEE-MICT lacks a fully mechanistic plant hydraulic structure that calculates plant internal water movement via constraints from water potential ($\psi$) and conductance of roots, stems and leaves. Instead, a stress factor, which ranges from 0 to 1, is calculated based on the relative moisture content at each soil layer. This factor is applied to stomatal conductance and mesophyll conductance, as well as the maximum RuBisCO activity rate (Vcmax) and maximum electron transport rate (Jmax), in order to account for experimentally observed effects of drought on stomatal and non-stomatal photosynthetic limitation (Zhou et al., 2014). The stress factor ($\gamma$) of water limitation is calculated as:

$$\gamma_i = \frac{\theta_i - \theta_{wp}}{\theta_{wp} + \rho\left(\theta_{fc} - \theta_{wp}\right)} \tag{12}$$





$$\gamma = \sum_{i=1}^{11} \gamma_i \, w_i \tag{13}$$

where $\gamma_i$ is relative moisture content at each soil layer i, bounded between 0 and 1; $\rho$ represents the fraction above which photosynthesis rate is not limited by soil moisture, and is set at 0.8; $w_i$ is the weighting factor for each layer.

In the initial version of ORCHIDEE, the profile of $w_i$ was assumed to be constant over time, although differed between tree
and grass PFTs, with the highest value at 1.5 m depth for trees and 0.37 m depth for grasses. We considered this description inappropriate for the high latitudes, and in particular for permafrost regions, where trees develop shallow and lateral roots above the permanently frozen layer (Kajimoto et al., 2003). Thus, in ORCHIDEE-MICT, $w_i$ is modified to be a dynamic profile which optimizes plant water use, in a manner inspired by the representation given in Beer et al. (2007):

$$w_i = \frac{\gamma_i}{\sum_{i=1}^{11} \gamma_i} \tag{14}$$

where if layer i is below modeled active layer thickness, $w_i$ is set to zero, and the remaining w are re-normalized to one.

### 4.4   Fires

The prognostic fire module SPITFIRE (SPread and InTensity of FIRE), which has been previously integrated to and calibrated for a standard version of ORCHIDEE (Thonicke et al., 2010; Yue et al., 2014), was merged into ORCHIDEE-MICT. Ignitions were re-calibrated using the GFED4s data set (http://www.globalfiredata.org/data.html) by using region-specific scaling factors
(see Table S3) and the exclusion of cropland fires to ensure that simulated mean annual burned area for 1997-2013 was equal to that of the GFED4s data set. Note that this method only calibrated for mean annual regional burned area, and that simulated latitudinal distributions and grid cell spatial patterns of burned area and fire carbon emissions, and their interannual and seasonal variabilities, could be still compared with observation-based data. Deforestation and peatland fires are not explicitly simulated, but as both fire types rely on suitable weather conditions to occur, which could be partly captured by SPITFIRE (Yue et al.,
2015), model simulations are expected to partially include these fire types.

### 5   Simulation protocol, forcing and evaluation data sets

### 5.1   Simulation protocol and forcings

### 5.1.1   Simulation set-up

Two separate runs using different climate forcing input data – CRUNCEP v7 (hereafter CRUNCEP) and GSWP3 – were
performed with ORCHIDEE-MICT for the terrestrial Northern Hemisphere (> 30°N) at 1° spatial resolution. Both sets of runs



encompass the 20[th] and beginning of 21[st] century, and were preceded by separate spin-ups for each climate dataset, forced by fixed pre-industrial conditions of atmospheric $CO_2$ (286 ppm) and vegetation map. The dynamic vegetation model is de-activated throughout both runs. In order to accumulate soil organic carbon in the model, which requires substantial computing time before reaching near-equilibrium in the presence of the slow mixing processes described in Section 4.1, the spin-up

procedure comprised three steps: 1) The full ORCHIDEE-MICT model was forced by looped climate fields over the period 1960-1990 for 100 years to reach equilibria for soil temperature, soil moisture, vegetation productivity, soil carbon inputs from dead plants, etc. We used the 1960-1990 loop, instead of pre-industrial climate, to approximate the higher Holocene temperatures relative to the 'pre-industrial' period that have been reconstructed in Marcott et al. (2013); 2) A soil carbon sub-model was run for 20,000 years, forced by the outputs from the preceding step; 3) The full ORCHIDEE-MICT model was run

for 100 years, forced by looped 1901-1920 climate data, to approach to the pre-industrial equilibrium for physical variables, carbon fluxes, and fast carbon pools. A final transient simulation from 1861 to 2007 (using 1901-1920 climate loop for the period 1861-1900 due to the lack of climate forcing before the 20[th] century) was then run from the last year of spin-up stage 3, forced by historical climate forcing and land cover maps, and rising $CO_2$ concentrations, as detailed below.

### 5.1.2   Atmospheric forcing datasets

The use of two different forcing datasets represents a first step in documenting atmospheric forcing based uncertainty in model output. Runoff has been shown for instance to be particularly affected by differences in precipitation from different datasets (Fekete et al., 2004; Biancamaria et al., 2009), and by the methods to partition total precipitation volumes between rainfall and snowfall during the cold season (Haddeland et al., 2011). The bias of meteorological driver also impacts the carbon budget (Zhao et al., 2012). A description of the two datasets used follows.

**GSWP3 v0**

This 3-hourly 0.5° global forcing product (1901-2007) was developed for the third phase of GSWP3 (http://hydro.iis.u-tokyo. ac.jp/GSWP3/). It is based on the 20[th] Century Reanalysis (20CR) version 2 performed with the NCEP land-atmosphere model (Compo et al., 2011). 20CR was dynamically downscaled to T248 (0.5°) resolution using the Global Spectral Model (GSM) by data assimilation using the spectral nudging technique (Yoshimura and Kanamitsu, 2008). Bias correction for precipitation,

temperature and longwave and shortwave downward radiations were made using GPCC v6 (Global Precipitation Climatology Centre), CRU TS v3.21 (Climate Research Unit), and SRB (Surface Radiation Budget) datasets, respectively. Precipitation was partitioned into rainfall and snowfall referring to the ratio of the downscaled 20CR, and wind induced undercatch correction (Motoya et al., 2002) were applied separately. We upscaled the GSWP3 forcing for 1° spatial resolution.

**CRUNCEP v7**

This 6-hourly 0.5° global forcing product (1901-2015) is a combination of the annually-updated CRU TS v3.24 monthly cli-mate dataset (New et al., 2000) and NCEP reanalysis (Kalnay et al., 1996). The latter is only used to generate diurnal and daily



anomalies added to CRU TS monthly means, after bi-linear interpolation to the 0.5° resolution of CRU, except for the precipitation which were linearly interpolated. A threshold of 0°C in 2-m temperature were used to partition the precipitation into rainfall and snowfall in the CRUNCEP forcing. Rainfall, cloudiness, relative humidity and temperature are taken from the CRU while the others fields (pressure, longwave radiation, windspeed) were directly derived from NCEP (see more details at: https://vesg.ipsl.upmc.fr/thredds/fileServer/store/p529viov/cruncep/readme.html). We upscaled the forcing to a 1° spatial resolution dataset, which can be found at https://vesg.ipsl.upmc.fr/thredds/catalog/store/p529viov/cruncep/V7_1901_2015/catalog.html.

### 5.1.3 Vegetation and soil texture map

The ESA CCI Land Cover map (Bontemps et al., 2013) was used to produce the PFT map for ORCHIDEE. The ESA CCI land cover product is given by 3 maps at a 300 m spatial resolution, corresponding to the years 2010, 2005 and 2000. These maps were derived from the interpretation of MERIS full and reduced resolutions and SPOT-Vegetation time series. Land cover was classified according to the 22 classes used in the UN-LCCS (land cover classification system) scheme, which was translated into PFT fractions used in ORCHIDEE, following the cross-walking method presented by Poulter et al. (2011, 2015). Historical land use maps from the Harmonized Global Land Use dataset (Chini et al., 2014) were incorporated to reconstruct PFT fractions since 1860, following Peng et al. (2017), which will be detailed in the upcoming ORCHIDEE-TRUNK paper for CMIP6.

For soil texture, we use the 12 USDA texture classes provided at a global 0.08° resolution from Reynolds et al. (2000) and upscaled these to the resolution of the atmospheric dataset (1°x1°). Only the dominant texture type for a grid cell is used at the 1° resolution for defining soil hydraulic parameters (Carsel and Parrish, 1988) in ORCHIDEE-MICT.

### 5.2 Evaluation datasets for the water budget

Table 1 gives an overview of the selected datasets to evaluate ORCHIDEE-MICT. For the water budget evaluation, we selected six Arctic river basins (Fig. 2b) which are important contributors to total Arctic Ocean river inflow: the four largest Eurasian Arctic basins (Ob, Yenisei, Lena and Kolyma), the Mackenzie basin in northwestern Canada and the Yukon basin in Alaska. The four Eurasian basins (with the smaller Pechora and the Severnaya Dvina basins) drain about two-thirds of the Eurasian Arctic landmass (Peterson et al., 2002), while the Mackenzie is the largest North American river bringing freshwater to the Arctic Ocean (Woo and Thorne, 2003). We also evaluated the Volga basin (Fig. 2b), which is subject to snowfall events during the year but is not underlain by permafrost, in order to compare results with the high-permafrost Arctic basins (Fig. 2a and Table S4).

### 5.2.1 Total terrestrial water storage

The change of total terrestrial water storage (TWS) is an integrated measure of the ability of a LSM to partition the cold season storage of water as snow and ice, and its decrease from losses to downstream river discharge, plant and soil evapotranspiration, and soil moisture increase. Alkama et al. (2010) showed with the ISBA-TRIP model that seasonal variations in TWS over high-



latitude basins resulted primarily from snow accumulation in the cold season, despite a wintertime underestimation of TWS values. Decreasing soil moisture contributed to a small decrease in springtime TWS, while exports with riverflow increased dramatically during the snowmelt period. The GRACE (Gravity Recovery And Climate Experiment) satellite mission permits estimation of monthly TWS variations through measurements of the Earth's gravitational field. Three solutions, at 1° resolution,

based on spherical harmonic coefficients (Release 05), are obtained by different processing centers: CSR (Center for Space Research), JPL (Jet Propulsion Laboratory) and GFZ (GeoForschungsZentrum Potsdam) and provided in the GRACE Tellus dataset (Swenson and Wahr, 2006; Swenson, 2012; Landerer and Swenson, 2012). We use the ensemble mean of these solutions to reduce uncertainty in GRACE data (Sakumura et al., 2014). To compare TWS simulated by ORCHIDEE-MICT to GRACE data over the common time period between the two meteorological forcing data sets and GRACE data (July 2003-December

2007), we summed the water stocks simulated by ORCHIDEE-MICT, i.e., soil moisture, snowpack, water on the canopy and water stored for the routing reservoirs. In each grid cell, the corresponding 5-year average is removed from the 2003–2007 time series of TWS output from ORCHIDEE-MICT. The comparison of simulated TWS with GRACE data over seven basins in this study (Fig. 2b) is statistically satisfactory given large surface areas occupied by these basins (> 400,000 km² (see Table S4), Swenson et al., 2003).

**5.2.2   Snow water mass**

The GlobSnow v2.0 SWE dataset is based on a data-assimilation approach combining space-borne passive radiometer data (SMMR, SSM/I and SSMIS) with data from ground-based synoptic weather stations. The record provides SWE information on a daily, weekly and monthly basis, at a spatial resolution of approximately 25 km (in EASE-Grid format) and applies to non-mountainous regions of the northern hemisphere, excluding glaciers and Greenland. The product is based on the SWE

retrieval methodology developed by Pulliainen (2006) complemented by a time-series melt-detection algorithm (Takala et al., 2009). A complete description of its methodology is given in Takala et al. (2011). SWE estimates are complemented with uncertainty data at the grid cell scale. This work uses monthly-averages of original daily data, interpolated to a 1° grid for appropriate comparison with ORCHIDEE-MICT output.

**5.2.3   Evapotranspiration (split into components)**

The Global Land Evaporation Amsterdam Model (GLEAM) (Miralles et al., 2011) estimates the daily terrestrial ET rate and its components (transpiration, bare-soil evaporation, open-water evaporation, interception loss and sublimation), as well as root-zone soil moisture, at a spatial resolution of 0.25°. It is driven by microwave remote sensing observations and uses satellite soil moisture to constrain potential evaporation rate. The latter is computed with the equation of Priestley and Taylor (1972), which is based on air temperature and net radiation. In the GLEAM v3 product (Martens et al., 2017), representation

of evaporative stress has been improved by the use of microwave vegetation optical depth as a proxy for vegetation water content. Other improvements include the use of satellite based SWE, reanalysis air temperature and radiation, and a multi-source precipitation product. GLEAM v3 contains three sub-versions: Version v3.0a is an assimilated product derived from satellite observations and multiple reanalysis climate forcing, while v3.0b and v3.0c are purely satellite-based products. As the





latter two do not provide full coverage of high-latitude regions (only 50°S – 50°N), the v3.0a, which covers the period 1980-2014, is used in our study. We use monthly-averages of original daily data, interpolated to a 1° grid for appropriate comparison with ORCHIDEE-MICT output.

### 5.2.4 River discharge

We use two river discharge databases, one global and another exclusively covering Siberia: The Global Runoff Data Centre (GRDC) product is a global database which collected river discharge data from nearly 7800 stations in 156 countries for the [maximum] period 1807-2017. We use one gauging station per river, the closest to the mouth of the river. For Siberia, we also use daily river discharge at the 3 gauges in three large Siberian rivers (the Ob, Yenisei and Lena) which have been reconstructed (naturalized discharge) to exclude the human impact from the data. This has been performed by using the Hydrograph Routing

Model (HRM) developed at the University of New Hampshire (USA) in collaboration with the Arctic and Antarctic Research Institute (Russia) (Shiklomanov and Lammers, 2009). We use monthly-averages of original daily data.

### 5.2.5 Top-soil moisture

The global ESA CCI SM product (v2.2) provides daily soil moisture (in volumetric units $m^3 m^{-3}$), at a spatial resolution of 25 km with quality flags specifying potential sources of errors linked to the presence of water bodies, dense vegetation, snow

or frozen soils in the pixel area. Three datasets are provided based on different combinations of active and passive satellite sensors. In this work, we used the combined active and passive product which covers the whole period from November 1978 to December 2014. The product results from the merging of soil moisture estimations inversed from two types of instruments and two methodologies: passive microwave radiometers (SMMR, SSM/I, TMI, AMSR-E, AMSR2 and WindSat) using the methodology developed by Owe et al. (2008) and active microwave instruments (ERS-1&2 and ASCAT) using the algorithm

developed by TU-Wien (Wagner et al., 1999; Bartalis et al., 2007). These two records were first rescaled together and merged with topsoil (10 cm) soil moisture simulations of the GLDAS-NOAH LSM using a CDF-matching approach. Data accuracy was estimated against in-situ measurements (Dorigo et al., 2015) to $0.05 m^3 m^{-3}$.

### 5.2.6 Root-zone soil moisture

The GLEAM root-zone soil moisture v3.0a database (Martens et al., 2017) is based on both satellite observations and reanalysis

data. It is a mixture of soil water content of three soil tiles: bare soil (0-10cm), herbaceous (0-100cm) and tall vegetation (0-250cm). The tile fractions are static and derived from MODerate resolution Imaging Spectroradiometer (MODIS) Global Vegetation Continuous Fields product (MOD44B, Hansen et al. 2005). Obviously, it is hard to compare the ORCHIDEE soil moisture with the GLEAM root-zone soil moisture directly not only due to the mismatch of defined soil tile depth but also due to the difference of soil tile fractions. Thus we implement the comparison after normalization of the relative soil moisture

values with their respective standard deviation, following Koster et al. (2009).



## 5.3 Evaluation datasets for air-to-soil temperature continuum

### 5.3.1 Snow depth

Realistically simulating snow depth is one of the prerequisites for accurately modeling soil thermal dynamics, particularly in winter. Daily snow depth in-situ data (1975-2005) from 524, 72, 528, 128 and 528 stations in Europe, Canada, USA, Russia and China were obtained from European Climate Assessment & Dataset (ECA&D), the National Climate Data and Information Archive of Environment Canada, the United States Historical Climatology Network (USHCN), the Russian Research Institute of Hydrometeorological Information—World Data Center (RIHMI-WDC) (Bulygina et al., 2011), and the National Meteorological Information Center of the China Meteorological Administration (Peng et al., 2010), respectively. Monthly averages were calculated when more than 2/3 of daily snow depth data were available.

### 5.3.2 Surface soil temperature

The soil temperature product used here (André et al., 2015) resulted from the use of combined passive microwave and thermal infrared data to estimate land surface temperature (LST), during summer snow-free periods (snow-covered pixels are masked) in the northern high latitudes. The product is based on the use of SSM/I-SSMIS 37GHz measurements at both vertical and horizontal polarizations and is provided on a 25 km resolution EASE grid at a one-hour time step for the period 2000-2011. LST retrievals are based on the assumption of a relationship between surface emissivities at both polarizations (Royer and Poirier, 2010) which was calibrated at pixel scale using cloud-free independent LST data from MODIS instruments. The SSM/I-SSMIS and MODIS data were synchronized by fitting a diurnal cycle model built on skin temperature with reanalysis provided by the European Centre for Medium-Range Weather Forecasts (ECMWF). This product was evaluated at local and circumpolar scales against MODIS LST, and the results show a mean Root Mean Square Error (RMSE) of the order of 2.5 K.

### 5.3.3 In-situ air and soil temperatures

Simultaneous measurements of snow depth, near-surface air temperature and top soil (at 20 cm depth) temperature from Russian meteorological stations (Bulygina et al., 2011; Sherstiukov, 2012) are used to evaluate the thermal insulating effect of snow during winter, by comparing the relationships between snow depth and the air-to-topsoil temperature gradient ($\Delta T$) (Wang et al., 2016b). This dataset includes monthly mean values at 268 sites for the period 1980-2000.

### 5.3.4 Active-layer thickness

The Circumpolar Active Layer Monitoring (CALM) network's in-situ data set was used to evaluate modeled soil active layer thickness. The CALM network aims to observe the response of near-surface permafrost and the active layer to climate change over long time scales. Thaw depth is measured at the end of the thawing season, so that it should be comparable to the maximum active layer thickness in the model. Before comparing to ORCHIDEE-MICT, the data from 221 sites were averaged from 1990 to 2015.



In addition, we used the active layer thickness map derived by Beer et al. (2013) over Yakutia at a spatial resolution of 0.5°. It is based on regional surveys of landscapes and permafrost conditions in Yakutia during the time period 1960–1987. The gridded datasets can be accessed at the PANGAEA repository. Values of active layer thickness (ALT) in this region range between 0.3 m north of 70°N to 3 m south of 65°N. Uncertainty increases with ALT and is highest (up to 1m) in the south because of the occurrence of discontinuous and sporadic permafrost landscapes.

## 5.4 Evaluation datasets for leaf area, carbon stocks and fluxes

### 5.4.1 Leaf Area Index

The Leaf Area Index (LAI) can be derived from satellite measurements, using reflectance measurements and land cover maps. Comparing the seasonal cycle of the satellite products with simulated LAI gives information on the appropriateness of the phenology modules. The amplitude of the cycle is also informative with respect to several photosynthetic parameters (e.g. maximum carboxylation rate Vcmax or specific leaf area).

**GIMMS**

The Global Inventory Modeling and Mapping Studies (GIMMS) LAI3g dataset is derived from Advanced Very High Resolution Radiometers (AVHRR) measurements. It is more specifically computed from the GIMMS Normalized Difference Vegetation Index (NDVI) 3g, using a neural network and the International Geosphere Biosphere Programme (IGBP) land cover classes (Zhu et al., 2013). The 1/12 degree files cover 30 years from July 1981 to December 2011, at a bi-weekly resolution. They were aggregated at a 1.0° spatial resolution on the model grid, to enable comparisons of the time-series and maps.

**GLASS**

The Global Land Surface Products (GLASS) LAI product (Liang and Xiao, 2012) is built from AVHRR reflectances over the period 1982-1999 and then from MODIS data over 2000-2012 (Collection 5). The processing is also based on a neural network and the 8 biomes of the MODIS land cover type 3 (Xiao et al., 2014). The files are available at a spatial resolution of 0.05° with an 8-day frequency and were aggregated at 1.0° spatial resolution.

### 5.4.2 NEE from atmospheric inversions

We use estimates of NEE (here the net land-atmosphere $CO_2$ fluxes excluding fossil fuel $CO_2$ emissions) from two atmospheric inversions where NEE is optimized at the resolution of a global atmospheric transport model each month to match observed $CO_2$ concentration gradients from a global network of ground based stations. The inversions are the Copernicus Atmosphere Monitoring Service (CAMS) inversion system (Chevallier et al., 2010) and the Jena CarboScope (Rödenbeck, 2005).

The inversion from CAMS uses atmospheric $CO_2$ concentration observations from a total of 132 sites covering the period 1979-2015 and includes all sites being added to the network though time, combined with the LMDZ INCA atmospheric transport model and prior information about fossil-fuel $CO_2$ emissions (from EDGAR3.2 FastTrack 2000 database) as well as



land/ocean-atmosphere fluxes, to minimize a cost function in order to calculate NEE. Here we use the latest version (15r4), in which monthly fluxes are calculated at 1.875° x 3.75° lat/lon resolution.

The Jena CarboScope atmospheric inversion uses the same set of atmospheric $CO_2$ observation sites available during the temporal observation period (1996-2015) referred to as Jena s96. Jena CarboScope inversion uses fossil fuel emissions from

EDGAR 4.2 and the TM3 global atmospheric transport models. Here we use monthly NEE from the latest version of Jena s96 (v3.8), which is provided at 3.8° x 5.0° lat-lon resolution.

### 5.4.3   Site-level GPP and NPP observations

**In-situ GPP and NPP measurements**

While there are several NPP datasets especially for forest (e.g. Cannel et al., 1992; Olson et al., 2001; Michaletz et al., 2014),

few provide both GPP and NPP at the same locations. We used a recent database of in situ co-located GPP and NPP measurements (Campioli et al., 2015), which is an extension of the Luyssaert et al. (2007) database for forests. The database consists of 131 sites, for which annual NPP obtained from biometric measurements and associated uncertainties are provided. The criteria employed by Campioli et al. (2015) for selecting the sites was the availability of methodologically independent and site-specific estimates of NPP and GPP. Carbon Use Efficiency (CUE) was calculated at each site by taking the ratio of NPP to

GPP. Site-level biomes are classified as one of tundra, boreal peatlands, marshes, forests, grasslands or croplands. To study the temperature sensitivity of GPP and NPP, we selected only sites northward of 30°N, where model grid cells were representative of both the vegetation type and meteorological conditions observed at each site. We thus restricted the selection to sites whose IGBP vegetation type had a corresponding fraction greater than 0.5 in the grid cell of ISLSCP II MODIS IGBP Land Cover product (Friedl et al., 2010). Site-years where the absolute difference between local and the Mean Annual Temperature (MAT)

of the GSWP3 or the CRUNCEP climate forcing fields was higher than 4 degrees were discarded. As no information was given about the time-period of site NPP and GPP observations in Campioli et al. (2015), we compared them to mean values of the model over the period 2000-2007. The subset of selected sites consisted then in 52 sites out of 131 in the full dataset.

### 5.4.4   Gridded GPP and NPP observation-based data

**GPP**

In addition to GPP from sites, we used the monthly gridded GPP observation-based product Model Ensemble Tree GPP (MTE-GPP) from Jung et al. (2009, 2011) from year 1982 to 2010. This product is the result of a statistical model that combines measurements at FLUXNET sites with geospatial information from satellite remote sensing (fAPAR) and meteorological data. MTE-GPP is obtained as the median from an ensemble of the 25 best model trees out of an initial set of 1000 and the Mean Absolute Deviation (MAD) of the ensemble is used to define the uncertainty of MTE-GPP, which is also provided as a monthly

gridded product. This uncertainty estimate was mentioned by Jung et al. (2011) to be underestimated as compared to the true RMSE. That is why we consider the RMSE from the cross-validation sites as an estimate of uncertainty. The RMSE of mean




MTE-GPP at cross validation sites is $270\,\mathrm{gC\,m^{-2}\,yr^{-1}}$. At global scale and over the period 1982-2008, the mean MTE-GPP is $933 \pm 46\,\mathrm{gC\,m^{-2}\,yr^{-1}}$ and the total is $119 \pm 6\,\mathrm{GtC\,yr^{-1}}$, the uncertainties being derived from the MAD of the MTE (Jung et al., 2011).

**NPP**

We use the MOD17A3.005 global annual NPP model driven by satellite observations from the NASA Land Processes Distributed Active Archive Center, at 1 km resolution and aggregated to the 1° grid of the model over the period 2000-2010. Yearly MODIS NPP is computed from yearly and daily components. Among the daily components, MODIS GPP is based upon a LUE model:

$$GPP = E * fAPAR * PAR \qquad (15)$$

where PAR is the Photosynthetically Active Radiation, fAPAR the fraction of absorbed PAR and E is the radiation conversion efficiency. E depends on vegetation type and on meteorological conditions (temperature, Vapor Pressure Deficit (VPD)). PAR, temperature and VPD used in MOD17A3.005-NPP are from the NASA Data Assimilation Office (DAO) and fAPAR is a MODIS product (Knyazikhin et al., 1998). A daily Maintenance Respiration (MR) was computed for leaves (l) and fine roots (fr), using the LAI MODIS product (Knyazikhin et al., 1998). On a yearly basis, MR was computed for livewood (lw) and

Growth Respiration (GR) for leaves, fine roots, livewood and deadwood (dw). These respirations were computed using a biome-specific parameter look-up table, derived from the BIOME-BGC terrestrial biosphere model (Heinsch et al., 2003). This set of parameters was recalibrated for the MOD17A3.005-NPP collection 5 (Zhao et al., 2005). Yearly MODIS-NPP was thus computed as follows (Running et al., 2004):

$$NPP = \left\{ \sum_{days} [GPP - MR\,(l) - MR\,(fr)] \right\} - MR\,(lw) - GR\,(l) - GR\,(fr) - GR\,(lw) - GR\,(dw) \qquad (16)$$

Turner et al. (2006) evaluated MODIS NPP at nine sites representing various biomes, using local meteorological, LAI and above ground NPP measurements as inputs for the BIOME-BGC model to derive NPP over a 25 km² area. They showed globally no bias but NPP was underestimated at the most productive sites (attributed to a negative bias of E) and overestimated at low productivity sites (attributed to a positive bias of MODIS fAPAR). Zhao et al. (2005) evaluated MODIS NPP against the Ecosystem Model-Data Intercomparison (EMDI) global database of NPP measurements (Olson et al., 2001) and found

a good agreement except for tropical forests (spatial $R^2 = 0.77$). Zhao et al. (2006) estimated the sensitivity of the MODIS NPP product to the use of different meteorological reanalysis input data, showing differences at global scale up to more than $20\,\mathrm{PgC\,yr^{-1}}$, with the largest differences in the tropics.





### 5.4.5 Soil carbon inventories

**NCSCD**

The Northern Circumpolar Soil Carbon Database quantifies storage of organic carbon in soils of the northern circumpolar permafrost region, comprising four depths up to 3m (0-30cm, 30-100cm, 100-200cm and 200–300 cm) (Hugelius et al., 2013).

Total SOC storage in the 0-2m and 0-3m depth ranges in northern permafrost soils are estimated to be 827±108 and 1035±150 PgC respectively.

**ISRIC**

ISRIC is a global soil information database with spatial predictions for a selection of soil properties, at 1 km resolution and six depths, up to a maximum of 2m ( 0-5cm, 5-15cm, 15-30cm, 30-60cm, 60-100cm and 100-200cm). About 110,000 soil profiles

were used to generate the product, using statistical models based on climatic and biomass indices, lithology, and taxonomic mapping units (Hengl et al., 2014). The gridded maps of soil organic carbon mass fraction, soil bulk density, and volumetric fraction of coarse fragments was used to calculate SOC density in the unit of $kgC\,m^{-2}$ to be comparable with model output (see Eq. 6 in Hengl et al., 2014). However, we found in this product a systematic overestimation of bulk density for organic-rich soils, which is similar to the issue found in the HWSD database (Köchy et al., 2015). Therefore, we followed the adjusting

method in Köchy et al. (2015) to correct bulk density for histosols and other soils with organic carbon mass fraction larger than 3%. This adjustment decreases total SOC stock in northern permafrost regions (the same domain as NCSCD) in the 0-2m depth range from the original 1724 PgC to 1177 PgC, which we considered acceptable though higher than NCSCD.

### 5.4.6 Biomass carbon stocks

Two forest biomass datasets are used to evaluate simulated forest biomass. The first is that from Avitabile et al. (2016), which,

at a spatial resolution of 0.01°, merges two tropical aboveground forest biomass (AGB) datasets from Saatchi et al. (2011) and Baccini et al. (2012) with northern hemisphere volumetric forest stock growth data from Santoro et al. (2015). The second forest biomass dataset from Thurner et al. (2014) is given at a spatial resolution of 0.01° for 2010 and confined to the northern hemisphere (30°N–80°N). This is derived from the Santoro et al. (2011) forest standing stock data. The two datasets were re-gridded to 1° resolution for comparison with model output. The Avitabile et al. (2016) AGB was converted into total biomass,

assuming a constant AGB to total biomass ratio of 0.8 and divided by two to obtain total biomass carbon, under the assumption that the carbon content of dry biomass is 50%. Liu et al. (2015) synthesized ratios of aboveground to total biomass for different forest biomes in temperate and boreal regions and found that they range between 0.76 and 0.84 according to regional forest inventory assessments. Our uniform 0.8 factor for converting the Avitabile et al. (2016) data thus yields a potential error of ~5% in inferred total biomass carbon, which is within the reported data uncertainties given by Avitabile et al. (2016). Simulated

equivalent total forest biomass carbon over the period 2000–2007 was compared with observations.



### 5.4.7 Burned area and fire emissions

We compared the simulated burned area and fire carbon emission with the Global Fire Emissions Database (GFED4s) data set, which is based on the GFED modeling framework (van der Werf et al., 2010). The GFED4s burned area data is derived from the MODIS sensor data that is then complemented by specific algorithms to retrieve "small fires" (Randerson et al., 2012).

Emissions are computed from a combination of burned area with the revised Carnegie-Ames-Stanford-Approach biogeochemical model (CASA-GFED, described in van der Werf et al., 2010), that provides an estimation of fuel loads and combustion completeness. GFED4s data is provided at a 0.25° spatial resolution with a monthly time-step for 1997–2015.

Cropland fires are not yet implemented in the ORCHIDEE-MICT model, but are included in the GFED4s dataset. In order to compare the simulation with GFED4s, it was necessary to correct the simulated burned area for the omission of cropland fires.

To do so, we computed for each grid cell the fraction of existing natural PFTs (i.e. all non-cropland PFTs), and divided the simulated burned area in each grid cell by this value. For carbon emissions, the GFED4s dataset provides separate emissions data for cropland and natural fires. In this way, we were able to simply remove cropland fire emissions from GFED for comparison with ORCHIDEE-MICT output.

## 6  Evaluation of the atmosphere-snow-soil continuum

In the following, we analyze model performance in replicating the transfer of heat from atmosphere to deep soils. This is done by evaluation against measured temperature gradients, starting from snow depth controls on winter $\Delta T$, followed by evaluation of surface (skin) temperature in summer, and the temperature gradients between the near-surface and deeper soils.

### 6.1  Snow insulation controls on the temperature gradient between air and topsoil

#### 6.1.1  Snow depth

ORCHIDEE-MICT correctly captures the spatial distribution of maximum monthly average snow depth (Fig. 3a,b), and the seasonal decrease of snow depth from March to June (Fig. 3c), but modeled snow depth strongly depends on the atmospheric forcing used. GSWP3 climate forcing tends to produce a larger maximum snow depth than CRUNCEP, greater than those observed in all Northern regions, especially over boreal Europe (BOEU) (Fig. 3c). This shows that uncertainty from climate forcing data is as large as the model bias compared with observations, making it difficult to attribute a model bias to a particular

component of the snow model. However, the rate of sublimation in winter (Pomeroy et al., 1998) and the prescribed albedo value of fresh snow have been shown to be critical in determining the peak value and phase of both snow depth and SWE (Wang et al., 2015b).

#### 6.1.2  Snow conductivity and snow density

Mean snow density and mean snow thermal conductivity are computed at the month of maximum snow depth over the 1981-

2007 period as weighted averages over the three snow layers. Gouttevin et al. (2012b) report density values of $200\,\mathrm{kg\,m^{-3}}$




for taïga and 330 kg m$^{-3}$ for tundra and conductivity values of 70 mW m$^{-1}$ K$^{-1}$ for taïga and 250 mW m$^{-1}$ K$^{-1}$ for tundra, from Sturm and Johnson (1992) and Domine et al. (2010). These higher values over tundra were attributed to snow compaction by wind. This process is not modeled in ORCHIDEE-MICT and we thus simulate similarly high values of conductivity for both biomes (Fig. 4): approximating tundra with C3 grass PFT between 55 and 85°N, and taïga with the boreal forests PFTs between

45 and 70°N and considering only grid cells with a fraction of the dominant biome above 0.6. The model yields a mean snow conductivity of 266 ± 203 (GSWP3) and 219 ± 197 (CRUNCEP) mW m$^{-1}$ K$^{-1}$ for tundra compared to 221 ± 113 (GSWP3) and 182 ± 100 (CRUNCEP) mW m$^{-1}$ K$^{-1}$ for taïga and a mean density of 269 ± 102 (GSWP3) and 239 ± 103 (CRUNCEP) kg m$^{-3}$ for tundra and of 233 ± 67 (GSWP3) and 207 ± 63 (CRUNCEP) kg m$^{-3}$ for taïga. Note that a recent study (Domine et al., 2016) suggests for tundra a complex structure with depth-hoar developing at the base of snowpack during the course of

the snow season, causing conductivities as low as 20 mW m$^{-1}$ K$^{-1}$ in late winter, whereas snow-compacted upper layers have conductivities of 200 to 300 mW m$^{-1}$ K$^{-1}$, more comparable to ORCHIDEE-MICT.

### 6.2   Summer land surface temperature

ORCHIDEE-MICT overestimates summer (June-August) LST of about 1.36°C when forced by GSWP3 (Fig. 5b) and 0.49°C by CRUNCEP (Fig. 5c). The bias is larger in northern and southern Siberia, where it can reach 4°C. These differences may be

linked to the overall underestimation of ET (see Fig. 12). This is further addressed in the Discussion (Section 10).

### 6.3   Soil temperature

The simulated spatial patterns of mean annual topsoil (0.2 m) temperature generally reproduce the observed gradient along a southwest-northeast transect in Siberia (Fig. 6a,b). However, the CRUNCEP-forced simulation results are colder than those from the GSWP3 ones as well as relative to observations in the permafrost region (Fig. 6a,b), mainly driven by a strong cold

bias in CRUNCEP-based winter soil temperatures (Fig. 6c,d).

During winter, the snowpack acts as an insulating layer above soil surface, reducing soil heat loss. Snow thus causes a large positive temperature gradient (ΔT), which is controlled by both snow depth and snow thermal properties, such as thermal conductivity, density and albedo. Generally, the model underestimates snow insulation in the early (Nov to Jan) and late (Feb to Apr) cold seasons for the same snow depth, as compared to observations (Fig. 7). This indicates that relatively congruent

wintertime soil temperatures in the GSWP3-forced simulation in permafrost regions (Fig.6c) may be due to a bias compensation from overestimated snow depth (Fig. 3) and underestimated snow thermal insulation.

A prominent component of the ΔT / snow depth relationship observed at Russian stations (black in Fig. 7) is the significantly lower insulation during the late cold season, probably due to snow compaction and densification leading to higher snow conductivity (Decharme et al., 2016). This differential sensitivity of ΔT to snow depth between the two periods is ef-

fectively captured by ORCHIDEE-MICT, despite small modeled negative ΔT values (i.e. topsoil colder than air) compared to observations at the termination of the snow season, when snow depth is diminished (< 20 cm).

Summer soil temperatures are higher in the GSWP3-forced simulation relative to those of CRUNCEP, and warmer than observations by 1~2 °C at 0.8 and 1.6 m depth at thaw season (Fig. 6c,d). This is consistent with an overestimation of ALT



(see Section 6.4). Differences between the two simulations in summertime soil temperatures may be partly driven by warmer GSWP3 land skin temperatures during summer (Fig. 5) than CRUNCEP. In addition, the cold bias in winter soil temperatures in the CRUNCEP-forced simulation might be carried over to summer via soil thermal inertia. This would partly offset the warm bias in CRUNCEP land surface temperatures during summer (Fig. 5c).

## 6.4    Active layer thickness and permafrost area

Figure 8a,b shows the simulated spatial pattern of ALT, as calculated from modeled soil temperatures using a linear interpolation between soil layers to locate the first depth that remains frozen (below 0°C) year-round. Compared with CALM observations, in which most of the sites have ALT <1m, the GSWP3-forced model generally overestimates ALT by more than 1 m (also see Fig. S3 which compares modeled ALT of Yakutia, East Siberia with the map of Beer et al., 2013), whereas CRUNCEP-forced output shows better agreement with the observations. This is consistent with our diagnosis that GSWP3-forced soil temperatures (Fig. 6) are overestimated in permafrost regions during summertime at depths of 0.8 m and 1.6 m. Similar results have been reported in several modeling studies (e.g. Ekici et al., 2014; Decharme et al., 2016), which show deeper modeled ALT compared to observations. Apart from the uncertainty induced by climate forcing differences, the model-data mismatch may also arise from scale differences, since models forced by 1°x1° resolution data cannot capture the sub-grid scale heterogeneity of land cover, soil texture, organic matter-modulated soil physical properties and topographic microclimates.

For simulated permafrost extent, two typical definitions of permafrost in LSMs, one defined as ALT less than 3 m to give "near surface" permafrost (e.g. Koven et al., 2013), the other defined if any of the soil layers stay frozen (e.g. Ekici et al., 2014; Burke et al., 2017b), produce quite different permafrost extents (Fig. 8c,d). This result highlights that the intercomparison of permafrost areas among different LSMs with differing soil vertical discretizations may include uncertainties brought by the arbitrarily chosen definition, whereas evaluation and comparison directly for soil temperatures and ALT should be more robust. A qualitative comparison against the empirical IPA (International Permafrost Association) permafrost map (Brown et al., 2002) shows better agreement for CRUNCEP compared to GSWP3-forced output, since CRUNCEP-forced simulation general matches the distribution of continuous permafrost using the former definition, while GSWP3-forced simulation seems to underestimate permafrost extent (Fig. 8c,d). This is consistent with the deeper simulated ALT under GSWP3 climate forcing.

## 7    Evaluation of large-scale water storage and fluxes

Simulated water budget components are evaluated over selected northern basins (Fig. 2b), most of which are underlain by permafrost (e.g. Lena, Kolyma), with the exception of the warmer Volga. The Ob and Yenisei catchments have contrasting north-south precipitation and temperature regimes, with attendant impacts on Arctic Ocean discharge and sub-basin scale water budgets. Here, only basin-scale averages are discussed.





### 7.1 Total terrestrial water storage change

A realistic phase and amplitude of TWS is simulated with both forcing datasets, although peak-to-peak amplitude is slightly overestimated in the Volga, Yenisei and Kolyma basins under GSWP3 input and the seasonal amplitude underestimated in the Yukon with both forcings, and in the Mackenzie and Lena with CRUNCEP forcing (Fig. 9). The positive temporal trend of

TWS in the Ob, Lena and Kolyma basins is captured well by ORCHIDEE-MICT, where it reflects upward precipitation trends (not shown). In general, the GRACE TWS is well captured by ORCHIDEE-MICT with both forcings, at the seasonal scale and for the 5-year trends, except in the Yukon basin where observed TWS decreases are not reproduced in our simulations, with no precipitation decrease in the GSWP3-forced model. The Yukon TWS decline is likely due to glacier melt in the northwest Cordillera (Wang et al., 2015a). As glaciers are not represented in ORCHIDEE-MICT, the model does not capture these TWS

trends. Note that groundwater storage changes related to the development of closed and open taliks (Muskett and Romanovsky, 2009) which contribute to existing TWS trends – increasing storage in the Lena and Yenisei, decreasing it in the Mackenzie basin, and no change in the Ob – are not modeled either in ORCHIDEE-MICT. Despite this, the model reproduces observed trends in these basins.

### 7.2 Snow related processes controlling land water storage in the cold season

The modeled seasonal cycle of the SWE and length of the snowmelt period are in agreement with observations (Fig. 10), suggesting a good parameterization of the snowmelt and sublimation processes. Yet, results differ strongly according to forcing inputs. In basins with a large permafrost fraction (Yukon, Lena and Kolyma) and, to a lesser extent, in the Mackenzie, Ob and Yenisei basins, SWE is underestimated throughout the year compared to GlobSnow data when ORCHIDEE-MICT is forced by CRUNCEP, and it is significantly larger in the GSWP3-forced simulation (in the Volga basin, the SWE is overestimated

in the two simulations). This is due to the low basin-specific snowfall rate in CRUNCEP forcing compared to GSWP3 (Fig. S10), which is probably the result of the criterion used to partition rainfall and snowfall in CRUNCEP, and strongly affects the simulation of snow depth and SWE (Loth et al., 1993; Wen et al., 2013). By contrast, the GSWP3-forced model captures the early winter SWE accumulation in these basins. In spring, the SWE is systematically overestimated except over the Lena, whose seasonal cycle is well reproduced by ORCHIDEE-MICT. This corresponds to an excessive persistence of the snow

cover, which may be explained by the absence of hysteresis in the snow depletion curve relating the snow cover extent and the SWE (e.g. Magand et al., 2014). In the Yenisei and the Mackenzie basins, the SWE in winter is closer to observations with the GSWP3 forcing, with the exception of springtime values, which are better under CRUNCEP forcing. In basins where permafrost area is near-zero (Ob and Volga), the SWE from CRUNCEP-forced simulation is closer to Globsnow than those from GSWP3, in which SWE is overestimated in winter and spring. This is related to the large difference in snowfall over these

basins between the two forcings (Fig. S10).





### 7.3 Soil moisture

**In the topsoil**

Seasonal evolution of topsoil moisture (first 2 cm) is compared to the ESA-CCI-SM product over the seven basins (Fig. 11a). Liquid soil moisture values from the model are used for the comparison because ESA-CCI-SM product measures only the topsoil moisture when temperature is above 0°C. Because of scaling issues (the satellite product is rescaled to the 10 cm top layer of NOAH model, as already noted, but is more representative of a thinner soil layer of about 2 cm), the comparison was performed on relative liquid soil moisture values after normalization with their respective standard deviation. The observed seasonal moisture variations are well captured by ORCHIDEE-MICT with both forcings over the seven basins (Fig. 11a). The maximum values occur in summer, in contrast to lower latitudes, because of the thawing processes occurring in summer. The local minimum simulated in summer in the Volga and Ob basins is underestimated, suggesting a too slow infiltration front of the water in the top soil layers of the model. Thus, less water in the root zone is available for transpiration which is found to be underestimated when compared to GLEAM (see Fig. 12b). Compared to observations, a more rapid increase (decrease) of the modeled topsoil moisture in spring during snow melt (in fall) is found in the Yukon and Mackenzie basins, related to a more rapid thawing (freezing) of the top soil.

**In the root zone**

A soil water comparison between GLEAM and ORCHIDEE-MICT is difficult because of differences in soil depth, which in GLEAM varies with vegetation cover, but is fixed at 2 m in ORCHIDEE-MICT. Moreover, the fraction of soil tile of short vegetation and forests used in the two products is not the same. We therefore normalize the relative root soil moisture (Fig. 11b) by its standard deviation to compare the dynamic of the soil moisture rather than the total amount of water in the soil. The intensity of water uptake by the roots of the vegetation in summer is generally well simulated by the model in both simulations. In basins underlain by permafrost, except in the Lena basin, water uptake is delayed by one month for both forcing sets, while the rate of decrease in root soil moisture is underestimated for GSWP3-forced output only. The similarity in output in this respect, despite very different SWE, highlights the low impact of the latter on the root soil moisture, and further underscores how the snowmelt differential is lost through runoff rather than being available for vegetation.

### 7.4 Evapotranspiration and component fluxes

The amplitude of the ET seasonal cycle is generally well captured by the model in most basins, when compared to the GLEAM product, despite a systematic underestimation by ORCHIDEE-MICT, whichever the input forcing (Fig. 12a). The disparity between modeled peak ET and GLEAM data is reduced under GSWP3 forcing in the Yukon, and under CRUNCEP forcing in the Yenisei and Lena. ET increase is underestimated in spring and early summer for both forcings, except in the Volga basin. This is consistent with modeled LAI increasing too late in spring (see Fig. 14) which could be due, at least partially, to an excessive persistence of the snow cover in spring. In autumn, the timing of the decrease in ET is reproduced by both forcings.





Model biases with respect to GLEAM data in the sublimation, soil evaporation and transpiration components of ET are shown in Fig. 12b. In all basins, sublimation bias in simulations forced by GSWP3 is ~0 in winter, in agreement with GLEAM. By contrast, CRUNCEP forced simulations slightly overestimate sublimation in early spring across basins with the exception of the Volga. These results are consistent with SWE underestimation (except in the Volga) (Fig. 10) and higher downward

shortwave radiation (Fig. S12) that results when CRUNCEP forcing is used. The general underestimation of summer ET by ORCHIDEE-MICT in the Yukon, Mackenzie and Kolyma basins is explained mainly by a too low transpiration despite bare soil evaporation being slightly over-estimated (Fig. 12b). When forced by CRUNCEP data, ORCHIDEE-MICT overestimates interception loss in all basins but the Kolyma and Yukon, which is consistent with CRUNCEP LAI overestimation (see Fig. 14).

**7.5   River discharge**

By comparing the two simulations, it is clear that the meteorological forcing exerts a significant influence on the simulated river discharge (Fig. 13): GSWP3 leads to systematically higher river discharge than CRUNCEP, which is perfectly consistent with the SWE biases (Fig. 10). In a majority of basins, GSWP3-forced simulations better capture the seasonal cycle of observed discharge than CRUNCEP-forced ones, especially in the Yukon and the eastern Siberian basins, where the discharge is strongly

underestimated under CRUNCEP forcing (between 60% in the Yenisei and 83% in the Yukon).

A first feature of the nival regime characterizing the studied high-latitude basins is the occurrence of low flows in winter, when water is frozen in the snowpack, soils, and river ice. This is well simulated by ORCHIDEE-MICT, despite a small underestimation in the Yukon, Mackenzie and Yenisei. Naturalized river discharge is available in the latter, and lower in winter than GRDC values, which reflects the effect of reservoir operations. The winter discharge simulated by ORCHIDEE-MICT in

the Yenisei is closer to the naturalized estimates, as the model does not account for artificial reservoirs. No natural lakes are simulated by ORCHIDEE-MICT, which may contribute to the winter discharge underestimation, especially for the Mackenzie which includes massive lakes (Great Slave, Great Bear, and Athabasca).

The nival regime of the studied basins is also characterized by peak flow in late spring, which is broadly captured by the ORCHIDEE-MICT simulations, even if the magnitude of peak flow is strongly biased in some cases, with a strong link to the

forcing used and the SWE biases. As an example, the Volga is the only river where both peak flow and SWE are overestimated with both forcing, and closer to river discharge observations under CRUNCEP forcing. In this human-altered basin, the absence of the simulation of water withdrawals for irrigation in the model can explain the peak flow overestimation.

An almost systematic weakness of the simulated hydrographs is the underestimation of river discharge during summer and fall, which is very strong with both forcing in the Yukon, Mackenzie, Lena and Kolyma, and to a lesser extent in the

Yenisei. This makes discharge closer to observations during the second half of the year under GSWP3 than under CRUNCEP, particularly in the Ob. This summer-fall underestimation propagates to underestimated annual mean discharge in a majority of basins (reaching -25 and -30% under GSWP3 in the Lena and Kolyma rivers) despite the overestimation of peak flows. Eventually, the discharge underestimation found in summer and fall, and on annual means in many basins, is not coherent with



the low simulated ET compared to GLEAM data. Yet, according to Wang et al. (2015a), biases in precipitation and ET datasets, that are used to evaluate the models, are source of errors for the water imbalances found in the northern high-latitude basins.

These river discharge results highlight deficiencies in the model representation of water infiltration in frozen soils, which appear to be too drastic in their prevention of snowmelt infiltration under conditions of frozen topsoils. Alternative parame-
terizations of these dynamics are underway. For example, infiltration of meltwater into frozen soil could be permitted when accounting for sub-grid scale variability of topsoil freezing and drying which would enhance infiltration (Gray et al., 2001). Other perspectives are the improvement of the floodplain parameterization in ORCHIDEE-MICT (Lauerwald et al., 2017) and the inclusion of natural lakes and artificial reservoirs.

## 8   Evaluation of the leaf area index, gross and net CO$_2$ fluxes

### 8.1   Leaf area index

According to the two evaluation products, the seasonal cycle of LAI (Fig. 14) is similar across the basins, with values near zero during winter, and a maximum in summer. There is a consistent phasing of seasonal LAI between the two evaluation datasets, however maximum LAI in GIMMS is systematically higher than in GLASS. In all the basins, the LAI simulated by ORCHIDEE-MICT has a phase delay of up to one month compared to both products. This is due to a delay in the start of the
growing season, which may be related to excessive persistence of the snow cover (Fig. 10). Note here that ORCHIDEE-MICT is prone to overestimate the timing of senescence (MacBean et al., 2015). This is true in particular for conifers, for which the model lacks an explicit senescence inception model. Except in the Yukon and the Mackenzie basins, the maximum LAI simulated by ORCHIDEE-MICT lies between the two satellite product estimates. Winter LAI of ~1.0 are overestimated in the Yukon, Mackenzie, Ob and Yenisei basins, however observations show values around zero. Given that these basins are covered
by a larger fraction of evergreen forests compared to the others, these simulated values look reasonable. The mismatch with the observations could be explained here by data errors, the assessment of solar reflectance from space in winter in the high latitudes being less reliable.

### 8.2   Gross (GPP) and net (NPP) primary productivity

#### 8.2.1   Spatial distribution and seasonal cycle of GPP

GPP in the high latitudes is co-limited by cold temperatures, constraining the duration of the growing season, and by summer water stress (Schulze et al., 1999) in northern Canada and Siberian boreal forests, for which the water balance is usually negative at this time. The simulated spatial pattern of GPP in Eurasia and North America is close to the MTE-GPP dataset (Fig. 15). Values lower than MTE-GPP were simulated in Eastern Siberia under the GSWP3 forcing, mainly due to water stress (see also underestimated biomass for eastern Siberia in Figs. 21). The seasonal GPP cycle in Fig. 15 (lower panel) is generally
accurate with respect to observations at the scale of large boreal regions; however peak GPP is strongly overestimated for boreal North America (BONA).



### 8.2.2 Spatial distribution of NPP and site level comparison

The distribution of NPP for both ORCHIDEE-MICT simulations is compared to forest site data from Campioli et al. (2015) (Fig. 16), in which NPP measurements are collocated with GPP (118 sites in total north of 30°N). Despite no sampling from Western Russia and Siberia, the model is able to reasonably capture NPP gradients in BONA and BOEU. In temperate forests

of western Europe and eastern US, modeled NPP is too low (see also Fig. 17b for warm sites), possibly due to high water stress in ORCHIDEE-MICT (see below) or due to a lack of N-deposition combined with soil fertility effects in modeled NPP.

Considering only the 52 sites that were selected (as described in Section 5.4), we computed the distributions of GPP and NPP by 5 °C mean annual temperature bins for the sites, the MTE-GPP and MODIS-NPP products, and for the ORCHIDEE-MICT simulations. We plot the 95th percentiles of these distributions in Fig. 17, which arguably defines an upper envelope for

temperature-limited GPP and NPP (see Fig. 3 of Luyssaert et al., 2007), and allows us to evaluate the spatial sensitivity of GPP and NPP to mean annual temperature. The behavior of all products is similar to the sensitivities derived from site observations, with a strong positive relationship between GPP, NPP and mean annual temperature over the range -10 to 10°C. ORCHIDEE-MICT then captures the decrease of GPP (NPP) at warmer sites but underestimate the values above 5°C. Global gridded data products generated independently from the Campioli et al. (2015) dataset exhibit sensitivities comparable to those derived from

local sites, whereas modeled GPP and NPP saturate for temperatures above 10°C, indicating water stress dominant controls.

The local measurements and the global observations products give similar median CUE values (51% and 49% respectively) and first and third quartiles whereas the model gives a narrower distribution range, with higher median values (57% for GSWP3 and 53% for CRUNCEP) (Fig. 18). This high CUE bias can be expected, as ORCHIDEE-MICT omits the effects of low nutrient availability on CUE (Vicca et al., 2012).

### 8.3 Spatial distribution of burned area and fire emissions

The spatial distribution of burned area is largely reproduced by ORCHIDEE-MICT, with a higher fraction of burned area in central Eurasia, and a decrease in burned area toward higher latitudes (Fig. 19). For the GSWP3 simulation, modeled total burned area over the period 1997-2007 (in the unit of $\text{Mha yr}^{-1}$) is smaller than GFED4s for BONA (1.7 vs. 2.2 in GFED4s), and higher for BOEU (8.1 vs. 5.0) and BOAS (16.1 vs. 10.4). Modelled spatial distribution of natural fire emissions has greater

discrepancies with respect to evaluation data than burned area, bearing in mind that GFED4s data is based on a biosphere model (CASA), not observations. Higher emissions in eastern Eurasia are reproduced by ORCHIDEE-MICT. However, modelled emissions are overestimated for central Eurasia and underestimated in boreal America, with respect to GFED4s. As a result, regional carbon emissions (in the unit of $\text{TgC yr}^{-1}$) in ORCHIDEE-MICT are lower than in GFED4s for BONA (20 vs. 48), much higher for BOEU (45 vs. 6), and in good agreement for BOAS (104 vs. 111). One possible reason for the discrepancy

in BOEU is the lack of forest management and fire suppression measures in ORCHIDEE-MICT, leading to higher simulated burned area and carbon emissions than in GFED4s. Changing the climate forcing from GSWP3 to CRUNCEP yields a smaller burned area (Fig. 19 b and c). Because the reduction of burned area mainly occurs in grassland, the impact on carbon emission is small.





Carbon emissions peak in summer for both GFED4s and ORCHIDEE-MICT for BONA and BOEU (Fig. S6d,e). However, for BONA the fire season starts one month later in ORCHIDEE-MICT (Fig. S6a). For BOAS and BOEU, there are stronger discrepancies in seasonal carbon emission (Fig. S6e,f). In particular, ORCHIDEE-MICT fails to account for the April peak in carbon emissions. A possible explanation for the missing April emissions in ORCHIDEE-MICT may be the late timing of

snowmelt (Fig. 10) and the delayed spring increase in LAI (Fig. 14). This would cause an unavailability of fuel in spring time. Changing the climate forcing from GSWP3 to CRUNCEP has only a very small effect on burned area and carbon emission seasonality.

## 8.4   Seasonal cycle of NEE

Monthly NEE from inversions, originally provided at the spatial resolution of each transport model, was aggregated at the

scale of the 3 high-latitude sub-regions (Fig. 2a). Spatially averaged NEE is expected to be more consistent between different inversions at coarser spatial scale, given the sparseness of atmospheric $CO_2$ stations and differences in transport models. Nevertheless, the seasonal cycle of NEE is generally consistently estimated by the two inversions in each sub-region, although Jena CarboScope estimates generally higher seasonal NEE amplitude, and in BOEU NEE from CAMS peaks one month earlier than Jena CarboScope (Fig. 20). Simulated seasonal NEE (defined as GPP minus autotrophic and heterotrophic respiration, fire

emissions and emissions from LUC) magnitude and evolution is very similar in both simulations and, in general, ORCHIDEE-MICT was able to reproduce the timing and magnitude of the transition between winter release and spring NEE uptake, despite a later onset of spring uptake for all three boreal sub-regions, and a smaller peak summer NEE for BOAS.

Since the timing of NEE uptake in Spring and release in Autumn is well constrained in the inversions from the observed periodical drawdown and buildup of $CO_2$ at atmospheric stations, the modeled delayed onset of Spring uptake is consistent

with the ~1-month lag between ORCHIDEE-simulated and satellite-observed LAI in Fig. 14 (and ET in Fig. 12a) in the basins of the three sub-regions. Even though GPP is overestimated in BONA during spring and summer (Fig. 15), the resulting NEE gives a slightly weaker uptake.

Modeled autumn and winter emissions (positive NEE) from soil respiration are well reproduced in the BOEU sub-region, even though the release is overestimated in autumn and underestimated in winter. The underestimation of NEE from November

to March in BONA and BOAS suggests that wintertime respiration in soils is underestimated in the model. This may be because: (i) decomposition in the model is cut off at T = -1°C, whereas observations suggest it can be sustained well below the freezing point in liquid films of soil pores (Schaefer and Jafarov, 2016); and (ii) insufficient snow insulation of soils (see 7). This suggests that we should either allow decomposition below the freezing point, perhaps to account for heat produced in the soil by microbial decomposition (Zimov et al., 1993; Hollesen et al., 2015), improve the snow insulating, or incorporate an

additional organic layer of insulating topsoil (e.g. mosses, O-horizons observed in boreal forests, see O'Donnell et al., 2011) into the soil module.

NEE discrepancies may be related to the different seasonal contributions of evergreen and deciduous trees to GPP and LAI (Fig. S7), as well as to the relative sensitivity of heterotrophic respiration, autotrophic respiration and disturbances to the warming (cooling) at the beginning of the growing season (winter).



## 9   Evaluation of carbon stocks

### 9.1   Biomass

Consistent with the LAI and GPP results, the simulation forced by CRUNCEP shows higher forest biomass carbon densities than the GSWP3 simulation (Fig. 21). The two simulations, however, exhibit similar patterns, which reproduce the general
spatial pattern of observed biomass, being higher in western and eastern regions of North America, with a declining gradient across Eurasia from the west to the east. However, in general, the model tends to overestimate carbon density in regions of observed high carbon density (e.g., Northwestern Europe and European Russia, eastern North America, Korean Peninsula and Japan). The model also misplaced the region of the highest biomass density in Northwestern Europe and European Russia rather than Central Europe as shown by the observation data. Likewise, the model estimates extremely large biomass in eastern
North America, especially by CRUNCEP forcing, which almost doubles the observed amount. For the rest of the study region, simulated carbon density is slightly lower than observations, by around 5–25 MgC ha$^{-1}$. The two observation data sets show considerable similarities.

Both model output and observation data are subject to the spatial uncertainties introduced by the use of satellite-derived land cover maps. When latitudinal distributions of total forest biomass are examined by using forest area distribution as prescribed in
the model input for all data sets, strong agreement is found between observed data and GSWP3-forced model output (Fig. S5a) for the whole study region. CRUNCEP-forced model output shows much higher biomass at all latitudes. For BOEU, the model estimated higher biomass observations for regions above 55°N (Fig. S5c), whereas for BOAS, model output fell between the observation datasets at latitudes of >50°N (Fig. S5b). The overestimation of biomass in BONA is consistent with that of GPP (Fig. 15), while biomass is more overestimated in BOEU compared with GPP, probably because of simulated longer biomass
turnover and lack of forest management and forest age structure. Over the whole study region, ORCHIDEE-MICT simulated a total forest biomass carbon stock of 95 PgC under GSWP3 forcing (165 PgC under CRUNCEP), close to estimates from forest inventory data in Pan et al. (2011) of 92.1 PgC. Somewhat lower estimates are derived by Avitabile et al. (2016) and Thurner et al. (2014), of 73 and 84 PgC, respectively.

### 9.2   Soil carbon

SOC stocks simulated by the model fit the spatial pattern from observed inventory data, including that of NCSCD's permafrost region near-surface SOC density (Fig. 22), but generate lower values than those from ISRIC in BONA and BOAS. ORCHIDEE-MICT tends to underestimate SOC density in deep soils below ~1m. This underestimation is maybe because the model does not include the sedimentation of soils that characterized peat formation during the Holocene, nor simulation of loess carbon or yedoma deposits during the last glacial period (Zimov et al., 2009; Zhu et al., 2016). Further, spin up was performed with
the coarse approximating assumption that Holocene climate is equal to recent climate. Nevertheless, it is important to note that ISRIC and NCSCD products also have important dissimilarities in overlapping regions suggesting that high systematic uncertainties exist in boreal and Arctic SOC inventories. We also observe that modeled SOC stocks are rather dependent on



climate forcing inputs, particularly for BONA and BOAS (Fig. 22). Higher stocks are produced under CRUNCEP forcing, mainly due to the higher primary productivity and therefore higher inputs to SOC associated with this forcing data.

## 10   Discussion

The model performances against datasets being documented in previous sections, we focus here on key mechanisms expected from a "high-latitude" model: 1) the conversion of winter snow and ice storage into a peak river discharge in spring, 2) the ability to capture the rectification of the seasonal amplitude of air temperature through the insulating snowpack and its attenuation at depth in the soil profile, and 3) the ability to reproduce the large scale gradients of carbon input to ecosystems from GPP and NPP and its further partitioning into live biomass and SOC pools driving soil heterotrophic respiration.

### 10.1   Conversion of winter water storage to spring and summer river discharge

We have shown that the model simulated very different river discharge values according to the atmospheric forcing used. The spatial distribution estimate of winter snowfall in high latitudes in climate datasets is very difficult. The first reason is the low density of meteorological stations, in particular in the CRU data (Burke et al., 2013), which leads to lower total precipitation in winter and spring (November to April) in CRUNCEP compared to GSWP3 forcing (Fig. S11). Another well known reason is the difficulty to catch snowfall in a gauge, because of its lower density; the wind prevents the snowfall from being vertical, leading to systematic undercatch (Yang et al., 2005). Then, the way the total precipitation is partitioned into rainfall and snowfall in the atmospheric forcing can also change the SWE in winter and spring. The threshold of 0°C in 2-m temperature, used to partition the precipitation in the CRUNCEP forcing, can lead to very different results compared to the physical partitioning performed within the dynamical downscaling for the GSWP3 forcing. In the Yukon and Kolyma basins, the proportion of snowfall to total precipitation in winter and spring being 100% in both forcings, the precipitation partitioning has no influence in these basins. By contrast, the proportion of snowfall to total precipitation in winter and spring largely differs between the two forcings in the other basins, in particular in the Volga (77% in GSWP3 and 53% in CRUNCEP), Ob (84% and 72%), Yenisei (87% and 80%) and Lena (96% and 89%) basins.

 We can assume that most of the forcing-related bias in SWE is converted into a river discharge bias in spring. Regressing the spring (April to June) river discharge bias for each forcing against the bias of SWE during the pre-melt season (February to March), over the same period of spring discharge and pre-melt season than Rawlins et al. (2007), shows that across the seven basins, 69% of the spatial variance of the bias of the spring flow is explained by the SWE bias during the pre-melt season (Fig. 23). With GSWP3 forcing, the positive ratio of pre-melt SWE bias to spring flow bias ranges between 7 and 97% in the different basins (Table S5). With the CRUNCEP forcing, this range is from 9 to 34%. In some basins such as the Ob or the Mackenzie with CRUNCEP or the Lena with GSWP3, the errors in spring discharge cannot be evidently related to the SWE bias (negative ratio). This can be explained by the periods chosen for spring and pre-melt which are not necessarily the same for all basins, in particular in the Ob, Mackenzie and Lena, the rivers for which the advance of simulated peak flow is the largest. The higher sublimation in the end of winter with CRUNCEP compared to GSWP3 (Fig. 12b), probably related to higher



downward shortwave radiation and lower specific humidity in CRUNCEP (Figs. S12 and S14 respectively), also contributes to the reduced spring discharge between the two simulations (Fig. 13). Over the Lena basin, higher soil evaporation together with higher interception loss (Fig. 12b), which is coherent with higher LAI (Fig. 14), lead to higher ET in summer with CRUNCEP (Fig. 12a), which reduces the spring discharge compared to GSWP3 (Fig. 13). The excessive persistence of the snow cover can

explain the delay of the LAI increase occurrence in the simulations, which may in turn contribute to too early simulated peak flow, also coupled to floodplain buffering. The Yukon, Mackenzie, Ob and Lena rivers have large floodplain areas and wetlands (and, in the Yukon and Mackenzie basins, lakes that enhance the floodplain buffer) that retain snowmelt water (Ringeval et al., 2012), and should attenuate the peak discharge in spring and sustain a significant summer discharge. The ORCHIDEE-MICT model uses a predefined floodplain map from GLWD (Lehner and Döll, 2004), but flooded area data products (e.g. Prigent

et al., 2016) show that wider areas are flooded in these catchments in spring, suggesting that better discharge could be obtained by connecting these larger floodplains and wetlands to the river routing scheme. This highlights complex potential interactions between the snow, vegetation, and river discharge dynamics, the overall results of which remain highly uncertain.

**10.2   Seasonal rectification of soil temperature in the atmosphere-snow-soil continuum**

A realistic modeling of soil temperature in cold regions requires not only a good soil thermal scheme, but also a good parame-

terization of snow insulation processes which strongly modulate soil temperatures during winter, as well as a realistic soil-plant hydrology which affects heat transfer in the air-soil interface during summer through ET, and in spring and autumn through latent heat uptake and release. Therefore, a better understanding and evaluation of model behaviors call for an integrated examination of all these elements. In this study, we showed that biases in modeled winter soil temperatures (Fig. 6) are connected with biases in snow depth (Fig. 3), while error compensations seem to exist in terms of snow depth and snow thermal conduc-

tivity. Model forced by the GSWP3 climate forcing, which has higher snowfalls than CRUNCEP forcing (Fig. S10), produces significantly larger snow depth and warmer winter soil temperatures than by CRUNCEP. However, the diagnosis from the relationship between $\Delta$T and snow depth, which reveals the model's intrinsic parameterization of snow insulation irrespective of snow depth, shows similar patterns with both GSWP3 and CRUNCEP forcings. The model can generally capture the broad characters in the $\Delta$T-snow depth relationship, including the different $\Delta$T regimes for different periods of the snow season,

yet underestimates the insulation given the same snow depth, compared to observations. This indicates an overestimation of effective thermal conductivity of snow. In the snow module of ORCHIDEE-MICT, snow compaction is parameterized such that snow density increases due to the weight of the overlying snow, while the top layer of snow keeps relatively low-density due to fresh snowfall; thermal conductivity of snow then evolves with time and depth, calculated as a non-linear function of snow density (Wang et al., 2013). Recent field measurements for Arctic snow, however, show that the formation of a thin layer of soft

depth hoar at the base of snowpack, with its low thermal conductivity around $0.025 \, \mathrm{W \, m^{-1} \, K^{-1}}$ which is ten times smaller than the intermediate snow layers, significantly re-shapes the vertical profile of snow thermal conductivity (Domine et al., 2016). It is therefore important to account for such complex metamorphic conditions of snowpack, especially in Arctic/sub-Arctic regions, in order to better model the soil thermal regime in permafrost regions.



Soil temperature during the thawing season is a key driver for microbial decomposition of soil organic carbon in the high latitudes; hence its realistic representation is crucial in modeling carbon cycle in permafrost regions. ORCHIDEE-MICT produces reasonable summer-time soil temperature and ALT in general, but significant discrepancies exist between the results of the model forced by the two climate forcings (Figs. 6 and 8). As the monthly mean gridded air temperature of GSWP3

and CRUNCEP forcings are almost identical (Fig. S9), both coming from the CRU data, the differences in the two results are mainly induced by other meteorological fields. A model intercomparison study (Peng et al., 2016) has reported that the surface incident longwave radiation is one of the dominant drivers of soil temperature trends in LSMs. Indeed, GSWP3 forcing has a systematically higher downward longwave radiation than CRUNCEP (Fig. S13), partly explaining the higher land skin and soil temperatures and deeper ALT with GSWP3 than with CRUNCEP forcing (Figs. 5, 6 and 8). What's more, the interactions

with hydrology and vegetation also play an important role. The higher ET in Siberian permafrost regions with CRUNCEP than with GSWP3 forcing (Fig. 12a), which is probably driven by the higher LAI in the same regions (Fig. 14), leads to more cooling for the land surface through latent heat release. Besides, the strong cold bias in winter soil temperatures with CRUNCEP forcing affects summer temperature through thermal inertia, especially for the deeper soils below topsoil. All these elements are interconnected, again highlighting the importance of synthetic evaluation of different yet related model aspects to improve

our understanding of the system.

### 10.3 Ability of ORCHIDEE-MICT to simulate northern land carbon fluxes and pools

Figure 24 shows the land carbon fluxes and pools modeled and derived from observations (datasets used in the text) in the model domain > 30°N. Modeled estimates encompass the range of observations excepted for the deep SOC pool where the model underestimates SOC. The same figures for each northern region are given in Fig. S8. At the regional level, model-observed

agreement still holds.

Arguably, the main variables that should be well simulated for a high-latitude land ecosystem model regarding future carbon-cycle climate feedbacks (Koven et al., 2013; Burke et al., 2017a) are the carbon stocks in biomass, litter and soils. In this respect, ORCHIDEE-MICT performs generally well for biomass, including the latitudinal profile, showing a peak between 50 and 60°N, despite the simple constant background mortality used for forest (Krinner et al., 2005) in a region where harsh climate

is known to induce mortality events. Note that the DGVM version of ORCHIDEE-MICT has a climate and PFT dependent mortality (Zhu et al., 2015) which was adjusted to reproduce the distribution of vegetation types, but it was not activated in this study aiming to reproduce observed stocks from an observed vegetation map. The inclusion of climate dependent fires adds another factor of mortality in this study. From simulated burned area and fire induced biomass suppression, we calculated that fires add a mortality of 0.4% yr$^{-1}$ of biomass in BONA, 0.3% yr$^{-1}$ in BONEU and 0.6% yr$^{-1}$ in BOAS, compared to the fixed

background mortality rates of 1.25%. Thus fire is a non negligible component of biomass mortality, controlling the modeled turnover of forest biomass (Thurner et al., 2016). Interestingly, ORCHIDEE-MICT can reproduce the latitudinal gradient of biomass stocks well, even in warmer regions where NPP is lower than observed (compare Figs. 21 and 16b). This suggests a possible error compensation with too small mortality and NPP in the southern boreal forests. Lack of representation of forest management, and under-representation of other natural disturbances (insect, wind, etc.) could collectively contribute to an





overall underestimation of biomass turnover. Carvalhais et al. (2014) reported an average turnover of 53.5 years for boreal forests, using extrapolated global soil database and satellite-derived biomass and GPP estimations. Their estimation of the turnover time integrates both biomass and soil C. Given a longer soil carbon turnover than biomass, the biomass turnover of boreal forests should be smaller than 53.5 years. While ORCHIDEE-derived turnover is between 54 (adding together fire and

background mortality) and 80 years (assuming only a 1.25% background mortality) with a lower boundary close to Carvalhais et al. (2014), given that forest fires impact only a small fraction of forests each year (0.1–3% on annual basis), the ultimate turnover time in the model could be much longer than Carvalhais et al. (2014). The last factor for overestimation of biomass is that ORCHIDEE represents a mature forest state everywhere due to lack of age structure in the model, whereas in reality disturbances and management have led to forest age younger than a mature one.

Regarding SOC, it is critical for carbon climate feedbacks that models can simulate the large current SOC stocks of the high latitudes. This is a particularly difficult problem since frozen high-latitude SOC was formed during the Pleistocene (Yedoma) or the Holocene (peat and tundra soils) through processes that are not incorporated as routine in current models, despite efforts in this direction, e.g. Zhu et al. (2016) for Yedoma and Kleinen et al. (2012) and Spahni et al. (2011) for Holocene peat deposits. The incorporation of slow forming high-latitude carbon deposits also requires climate history for deeper past periods. In order

to reproduce the burial of SOC below the active layer, the strategy followed in ORCHIDEE-MICT is inspired from Koven et al. (2009) who used a diffusion equation to move carbon in the permafrost where no decomposition occurs. Other studies (Koven et al., 2013; Burke et al., 2017a) further limited the rate of decomposition of SOC at depth, to reproduce the lack of oxygen inhibiting decomposition. Here we argue that key to the simulation of a high and realistic SOC stock is the correct representation of soil thermics, illustrated in Fig. 6. A detailed study of the sensitivity of modeled SOC to soil thermics during

the spin-up phase will be presented in a follow up study but we already found comparing SOC with CRUNCEP and GSWP3 climate forcings, the latter giving warmer soil temperature profiles, that SOC is indeed lower in GSWP3. Overall, the simulated circumboreal SOC stocks are comparable with NCSD observations (Fig. 22). This suggests that simulating the formation of frozen carbon with a diffusion-related burial process gives a good performance for the model, although the too stiff profile of SOC (Fig. 22) indicates that a further inhibition of respiration in deep horizons could improve the modeled vertical distribution.

Other processes that would tend to make the vertical profile of SOC more uniform with depth is the omission of interactions between SOM and the physical soil environment (Doetterl et al., 2015; Tipping et al., 2016). Modeled SOC is too low in regions covered by peat (lowland Hudson Bay, Ob northern and central basin) and Yedoma (Northeastern Siberia and Alaska) (Fig. 22), suggesting that adding these two SOC formation processes should further improve the modeled SOC patterns in longitude, and that in non-peat and non-Yedoma regions, there is no significant model bias for SOC.

Second, for research related to biophysical vegetation-climate feedbacks and understanding the current variability of the carbon cycle, having a correct seasonal representation of GPP, NPP and of the related plant transpiration is a critical requirement for a LSM. The phase of simulated LAI in spring lags satellite observations, in particular for BONA and BOAS sub-regions (Fig. 14). There is also a lag of GPP (Fig. 15), and of ET (Fig. 12a), but of smaller length (~15 days) compared to the lag of LAI. In order to further analyze why the lag of GPP (and ET) in spring is shorter than the one of LAI, we represent

separately monthly GPP and LAI for deciduous and evergreen PFTs (Fig. S7). The phase of GPP coincides with that of LAI




in deciduous, whereas GPP increases in spring as soil thaw faster and earlier than LAI for evergreens. Thus, it is the modeled spring decoupling between GPP and LAI in evergreen forest that explains the phase bias of GPP being smaller than that of LAI. Note that evergreen trees GPP is modeled to increase rapidly in early spring ORCHIDEE-MICT, even if the model has an inhibition of Vcmax when previous monthly air temperature remains below -4°C (Krinner et al., 2005) to reproduce impaired

photosynthesis observed in conifers, a physiological measure to avoid frost damage of photosystems (Leinonen, 1996; Tanja et al., 2003).

Peak summer GPP is overestimated in BONA and BOEU sub-regions compared to MTE-GPP (Fig. 15) and mean GPP is underestimated at southern sites in W. Europe and US where T > 7°C (Fig. 17a). We checked that underestimated GPP at warmer sites and in Eastern Siberia for the deciduous needle-leaved forest PFT (Larch) is due to an overestimation of water

stress in summer by the model, since ET is also lower than observed at all the warmer sites. Without the dynamic root allocation to optimize water use in the root zone (see Section 4.3), GPP in Eastern Siberia would be even lower compared to MTE-GPP (not shown). Last, regarding the timing of GPP decrease in autumn, ORCHIDEE-MICT shows good performances (Fig. 15) despite the lack of an explicit senescence model for conifers.

Interestingly the reasonable seasonal phase of simulated GPP (Fig. 15) can be contrasted with the larger lag of spring NEE

uptake compared to inversions results (Fig. 20). This suggests that in the model, ecosystem respiration may peak too early in spring when topsoil thaws, possibly because it lacks a representation of $CO_2$ trapping in the soil column before complete thawing of the soil, as observed in a tundra site (Elberling and Brandt, 2003).

The connection between modeled GPP and carbon pools takes place through NPP and its allocation to interconnected pools of variable turnovers. Of critical importance is thus the ability of the model to reproduce CUE which defines the ratio of

NPP to GPP, i.e. carbon available for ecosystem pools. Piao et al. (2010) showed that CUE of forest decrease from 0.5 to 0.3 between 10°C and -10°C, due to higher maintenance needs of trees e.g. to recover from cavitation in spring or maintain tissues during the cold-season period. While mean CUE is correctly simulated (Fig. 18), ORCHIDEE-MICT does not capture its observed decrease towards colder temperatures (data not shown), possibly because allocation and NPP are not driven by sink considerations (Körner, 2003; Fatichi et al., 2014) and thus too much GPP is used to make NPP at colder temperatures.

**11  Conclusions**

This study has described the inclusion of parameterizations which link soil carbon content and its decomposition rates with permafrost physics and hydrology in the ORCHIDEE-MICT land surface model. The model was evaluated against temperature gradients between the atmosphere and deep soils, and reasonable captured active layer thickness, northern permafrost extent, and soil carbon stocks and profiles. We have shown that the simulated water balance components and their seasonal transition

between cold season storage, mostly in solid form, and warm season loss are comparable to observations. Naturally, there remains significant room for improvement. The model appears to underestimate evapotranspiration and overestimate surface temperatures in the southern portion of the northern boreal zone, particularly over southeastern Siberia. Through the use of two different climate forcing datasets across all diagnostic variables, we found that for a large number of these, the variation in





output arising from choice of a forcing was as large as the discrepancy between model and observations. This raises a caution flag against 'over-calibrating' the parameters of a LSM to match measured quantities in the presence of large biases in climate input datasets. One critical aspect of forcing data in this respect is the partitioning of precipitation between snowfall and rainfall, which appears to produce a large difference in modeled snow mass and depth. Future improvements of the ORCHIDEE-MICT

model include the need for a better floodplain parameterization (Lauerwald et al., 2017) and a better representation of floodplain areas and wetland effects on river discharge. Including an explicit senescence inception model in ORCHIDEE-MICT may contribute to correct the excessive persistence of LAI during the fall, or even winter in the Yukon and Mackenzie basins. The subgrid-scale representation of permafrost hydrology in discontinuous permafrost areas may be necessary, as might a more realistic description of the slow processes that accumulate carbon in the soil (peat, yedoma, erosion) on very long time scales.

**Author contributions**

M. Guimberteau and D. Zhu led the study. D. Zhu performed the simulations. F. Maignan and A. Jornet-Puig performed the code versioning of ORCHIDEE-MICT; F. Maignan managed the scientific aspects of the updates from the main branch (TRUNK); A. Jornet-Puig optimized the code; they both assisted with technical aspects of the model and its run environment. H. Kim built the GSWP3 dataset and J. Chang adapted its format to be compatible with ORCHIDEE-MICT. D. Zhu developed

SOC-dependant soil physical properties in the model and introduced a reformulation of the hydric stress above the permafrost table. She conducted the evaluation of the soil temperature and permafrost area simulated by ORCHIDEE-MICT. Y. Huang provided the comparison of the snow depth with dataset. C. Ottlé and S. Dantec-Nédélec processed the satellite datasets and interpreted the soil moisture, land surface temperature and albedo results. A. Ducharne contributed to the improvement of the hydrological modeling in the initial ORCHIDEE version and M. Guimberteau led the water budget evaluation. Z. Yin

contributed to the comparison of soil moisture with GLEAM. M. Guimberteau, R. Lauerwald and S. Bowring supervised the routing scheme. C. Yue built an updated version of SPITFIRE in ORCHIDEE-MICT and, with P. Laurent, analyzed the results in biomass carbon stocks, burned area and fire emissions. A. Bastos provided the comparison of NEE with atmospheric inversions. F. Maignan performed the NPP and GPP evaluations over the sites database. B. Guenet, M. Tifafi and D. Zhu conducted the soil carbon evaluation. Y. Wang synthesized the carbon budget evaluation. C. Ottlé, T. Wang, S. Peng and G.

Krinner supervised the representation of the high-latitude processes in ORCHIDEE-MICT in coherence with the initial version of ORCHIDEE. D. Goll, X. Wang, E. Joetzjer, C. Qiu and F. Wang assisted in ensuring consistency between the carbon cycle, water cycle and various other processes in ORCHIDEE. P. Ciais contributed to the theoretical concept of the modeling of high-latitude processes. All authors contributed to the writing of the manuscript.

**Competing interests**

The authors declare that they have no conflict of interest.



**Code availability**

The SVN version of the code branch is https://forge.ipsl.jussieu.fr/orchidee/browser/branches/ORCHIDEE-MICT revision 4126 from the 28th February 2017. Please contact the corresponding author for the code of the ORCHIDEE-MICT if you plan an application of the model and envisage longer-term scientific collaboration.

**Data availability**

Primary data and scripts used in the analysis and other supplementary information that may be useful in reproducing the author's work can be obtained by contacting the corresponding author.

*Acknowledgements.* This work was financially supported by the European Research Council Synergy grant ERC-2013-SyG-610028 IMBALANCE-P. We acknowledge the ORCHIDEE-PROJET group, which developed and tested the ORCHIDEE model revision 3976. We thank Natasha
MacBean who contributed to the production of the land cover map. We thank Arsène Druel who processed the GLASS LAI data. Simulations with ORCHIDEE-MICT were performed using computational facilities of the Institut du Développement et des Ressources en Informatique Scientifique (IDRIS, CNRS, France) and of the CCRT (CEA). S. Dantec-Nédélec, G. Krinner, P. Ciais and C. Ottlé were supported by the CLASSIQUE national project (ANR Grant No. ANR 2010-CEPL-012-02) and the PAGE 21 project (grant agreement number 282700, funded by the EC seventh Framework Programme theme FP7-ENV-2011). A. Ducharne was supported by the IGEM project 'Impact of
Groundwater in Earth system Models' (ANR Grant No. ANR-14-CE01-0018-01). Discharge station data is kindly provided by the Global Runoff Data Centre, 56068 Koblenz, Germany. GRACE land are available at http://grace.jpl.nasa.gov, supported by the NASA MEaSUREs Program. The MOD17A3.005 data product was retrieved from the online Data Pool, courtesy of the NASA EOSDIS Land Processes Distributed Active Archive Center (LP DAAC), USGS/Earth Resources Observation and Science (EROS) Center, Sioux Falls, South Dakota (https://lpdaac.usgs.gov/data_access/data_pool). Jena CarboScope inversion data were provided by Christian Rödenbeck and are available at
http://www.bgc-jena.mpg.de/CarboScope/?ID=s96_v3.8.



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





| Dataset | Variable | Resolution | Period | URL | References |
|---|---|---|---|---|---|
| **Evaluation datasets for water budget** | | | | | |
| GRACE | TWS | 1° | Jul. 2003-Dec. 2007 | http://grace.jpl.nasa.gov | Swenson and Wahr, 2006; Swenson, 2012 |
| | | | | | Landerer and Swenson, 2012 |
| GlobSnow | Snow water mass | 25 km | 1979-2013 | www.globsnow.info | Takala et al. (2009) |
| GLEAM v3.0a | Evapotranspiration | 0.25° | 1980-2014 | http://www.gleam.eu | Miralles et al. (2011) |
| GRDC | River discharge | - | 1981-2007 | http://www.bafg.de/GRDC/EN/Home/homepage_node.html | - |
| Naturalized discharge | River discharge | - | 1981-2007 | http://www.r-arcticnet.sr.unh.edu/ObservedAndNaturalizedDischarge-Website | Shiklomanov and Lammers (2009) |
| ESA CCI SM v2.2 | Top-soil moisture | 25 km | Nov. 1978-Dec. 2014 | http://esa-soilmoisture-cci.org | - |
| GLEAM v3.0a | Root-zone soil moisture | 0.25° | 1980-2014 | http://www.gleam.eu | Martens et al. (2017) |
| **Evaluation datasets for air-to-soil temperature continuum** | | | | | |
| ECA&D | Snow depth | - | 1975-2005 | http://ecad.knmi.nl/dailydata/predefinedseries.php | - |
| National Climate Data and | Snow depth | - | 1975-2005 | http://climate.weather.gc.ca/historical_data/search_historic_data_e.html | - |
| Info. Archive of Env. Canada | | | | | |
| USHCN | Snow depth | - | 1975-2005 | http://cdiac.ornl.gov/epubs/ndp/ushcn/ushcn.html | - |
| RIHMI-WDC | Snow depth | - | 1975-2005 | - | Bulygina et al. (2011) |
| National Meteo. Info. | Snow depth | - | 1975-2005 | - | Peng et al. (2010) |
| Center of the China | | | | | |
| Meteo. Admin. | | | | | |
| - | Surface soil temperature | 25km | 2000-2011 | http://doi.pangaea.de/10.1594/PANGAEA.833409 | André et al. (2015) |
| - | In-situ air and | - | 1980-2000 | - | Sherstiukov (2012) |
| | soil temperatures | | | | |
| CALM | Active-layer thickness | - | 1990-2015 | - | - |
| For Yakutia | Active-layer thickness | - | 1960-1987 | doi:10.1594/PANGAEA.808240 | Beer et al. (2013) |
| **Evaluation datasets for leaf area, carbon stocks and fluxes** | | | | | |
| GIMMS | Leaf Area Index | 0.08° | Jul. 1981 to Dec. 2011 | http://nasanex.s3.amazonaws.com/AVHRR/GIMMS/LAI3G | Zhu et al. (2013) |
| GLASS | Leaf Area Index | 0.05° | 1982-2012 | http://glcf.umd.edu/data/lai | Liang and Xiao (2012) |
| CAMS | NEE | 1.875° x 3.75° | 1979-2015 | https://apps-test.ecmwf.int/datasets/data/cams-ghg-inversions | Chevallier et al. (2010) |
| Jena s96 v3.8 | NEE | 3.8° x 5.0° | 1996-2015 | http://www.bgc-jena.mpg.de/CarboScope/?ID=s96_v3.8 | Rödenbeck (2005) |
| - | GPP | - | Not known | - | Campioli et al. (2015) |
| MTE-GPP | GPP | 0.5° | 1982-2010 | - | Jung et al. (2009, 2011) |
| - | NPP | - | Not known | - | Campioli et al. (2015) |
| MOD17A3.005 | NPP | 1 km | 2000-2010 | - | - |
| NCSCD | Soil carbon inventories | 0.01° | - | http://bolin.su.se/data/ncscd/netcdf.php | Hugelius et al. (2013) |
| ISRIC | Soil carbon inventories | 1 km | - | doi:10.5879/ecds/00000001 | Hengl et al. (2014) |
| - | Biomass carbon stocks | 0.01° | - | - | Avitabile et al. (2016) |
| - | Biomass carbon stocks | 0.01° | - | http://www.biomasar.orghttp://www.bgc-jena.mpg.de/geodb/projects/Home.php | Thurner et al. (2014) |
| GFED4s | Burned area | 0.25° | 1997-2015 | http://www.globalfiredata.org/data.html | van der Werf et al. (2010) |
| | and fire emissions | | | | |

**Table 1.** List of the data sets used for the ORCHIDEE-MICT evaluation, with their references, the original spatial resolution, and period of availability





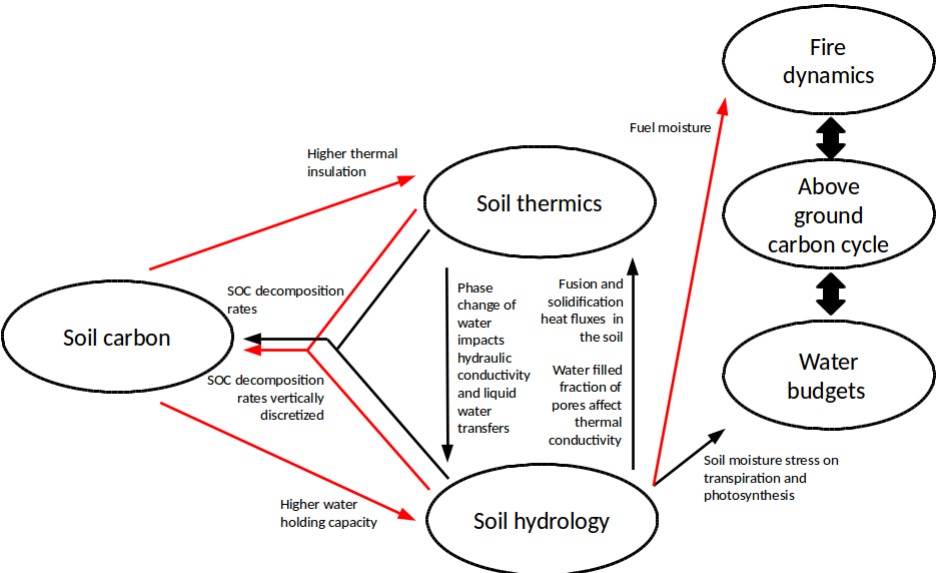

**Figure 1.** Temperature, water and carbon interactions in the initial version of ORCHIDEE (black), and processes included in ORCHIDEE-MICT in this study (red)

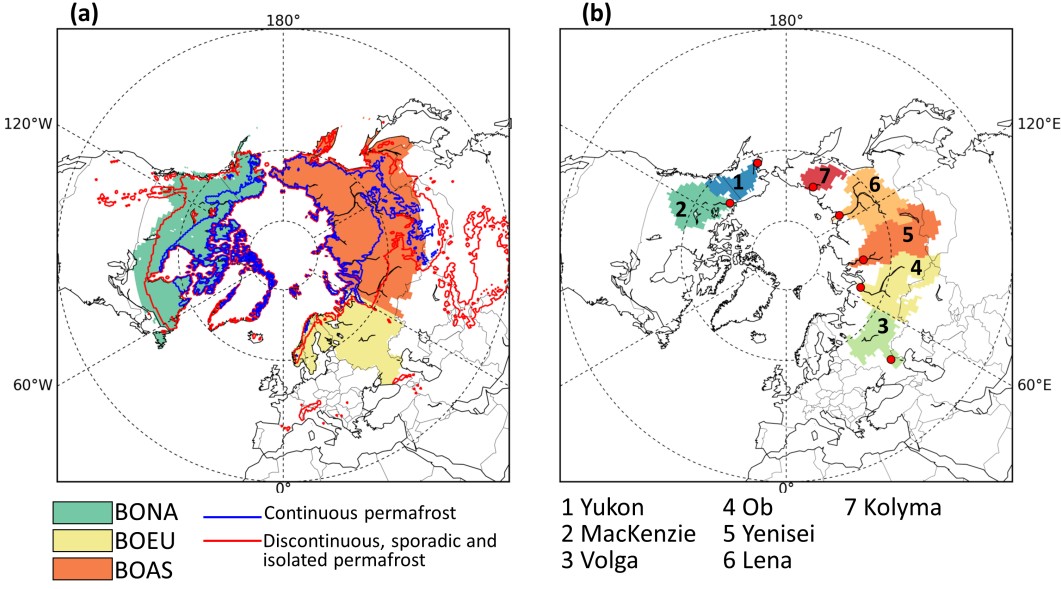

**Figure 2. (a)** The three high-latitude sub-regions used in this study, including boreal North America (BONA), boreal Europe (BOEU) and boreal Asia (BOAS), following McGuire et al. (2016). Blue and red lines indicate the extent of continuous permafrost and all permafrost categories respectively, according to the IPA permafrost map (Brown et al., 2002). **(b)** The seven high-latitude basins selected for this study with the gauge stations (red circles on the map, more information in Table S4).

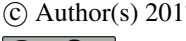



**Figure 3.** Maximum monthly snow depth (m) simulated (background maps) with **(a)** GSWP3 and **(b)** CRUNCEP forcings compared to observations (color filled circles), averaged over the period 1975-2005. **(c)** Monthly mean seasonal snow depth (m) from observation and the two simulations, averaged over the observation sites in the three high-latitude sub-regions (shown in Fig. 2a).





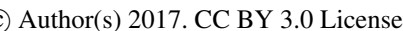

**Figure 4. (a,b)** Mean snow density ($\mathrm{kg\,m^{-3}}$) and **(c,d)** snow conductivity ($\mathrm{mW\,m^{-1}\,K^{-1}}$) computed over the three snow layers, averaged over the period 1981-2007. Results from the **(a,c)** GSWP3 and **(b,d)** CRUNCEP-forced simulation.





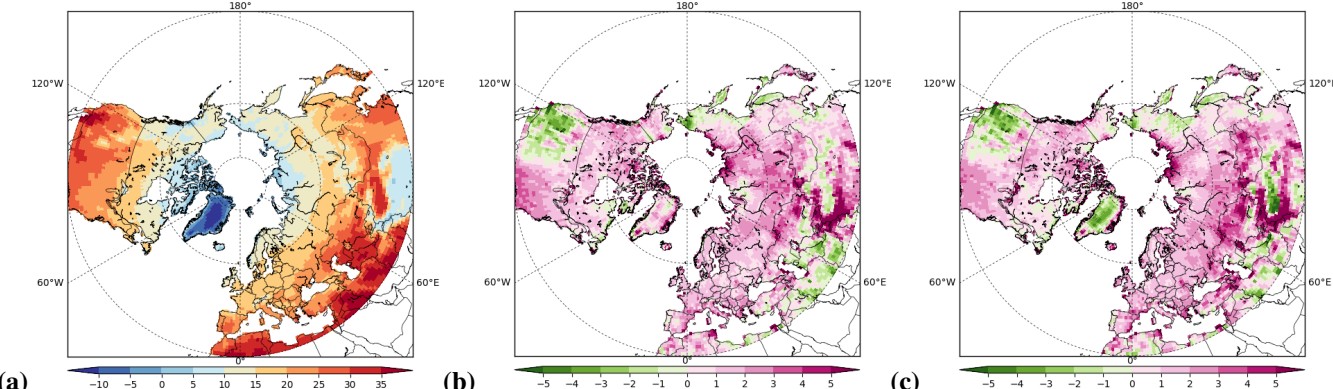

**Figure 5. (a)** Observed mean summer (JJA) land surface temperature (°C) and bias in the **(b)** GSWP3 and **(c)** CRUNCEP-forced simulation, averaged over the period 1996-2007.

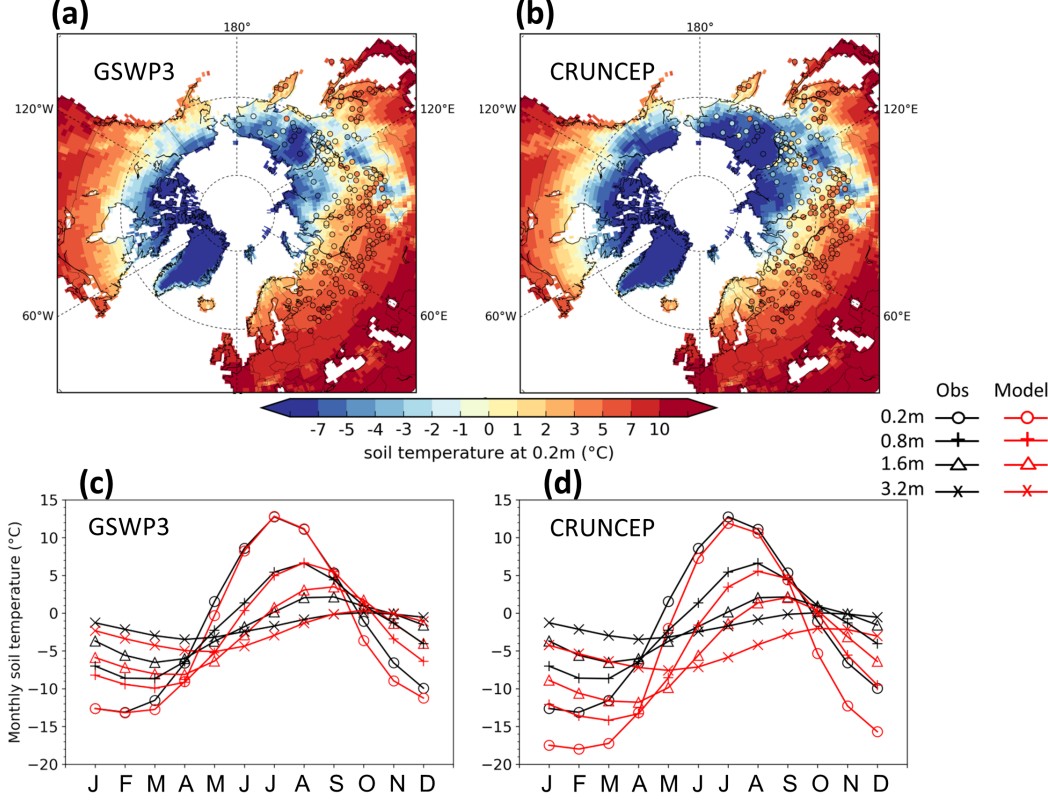

**Figure 6.** Mean annual soil temperature at 0.2 m depth (°C) in the **(a)** GSWP3 and **(b)** CRUNCEP-forced simulation (background maps), compared to the site observations (color filled circles), averaged over the period 1981-2000. Monthly mean seasonal soil temperatures at different depths (°C) in the **(c)** GSWP3 and **(d)** CRUNCEP-forced simulation, compared with the observation, averaged over the 51 sites in continuous permafrost region (according to the IPA map) and over the period 1981-2000.





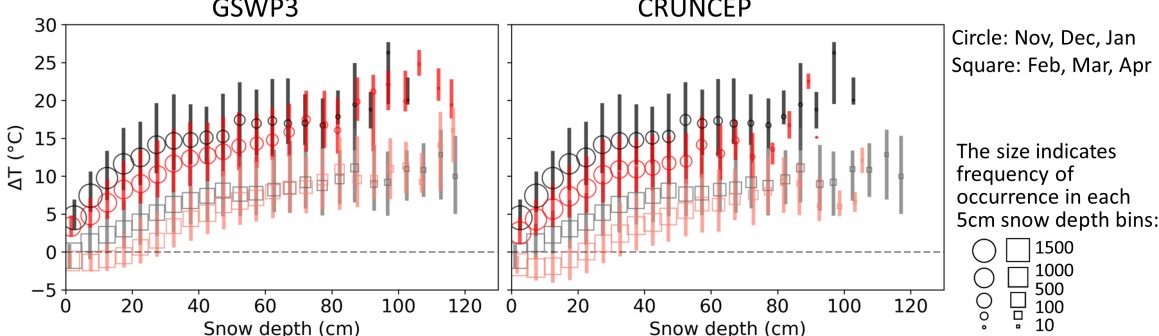

**Figure 7.** Relationship between ΔT (soil temperature at 20 cm depth minus air temperature) and snow depth (cm) over the period 1981-2000, for site-level observations (black), and for model results (red) (9612 site-month values in total), forced by **(left)** GSWP3 and **(right)** CRUNCEP. Circles and squares are medians of 5 cm snow depth bins, representing early (Nov, Dec, Jan) and late (Feb, Mar, Apr) snow season respectively. Upper and lower bars indicate the 25th and 75th percentiles of each bin. The size of circles/squares indicates frequency of occurrence in each bin.

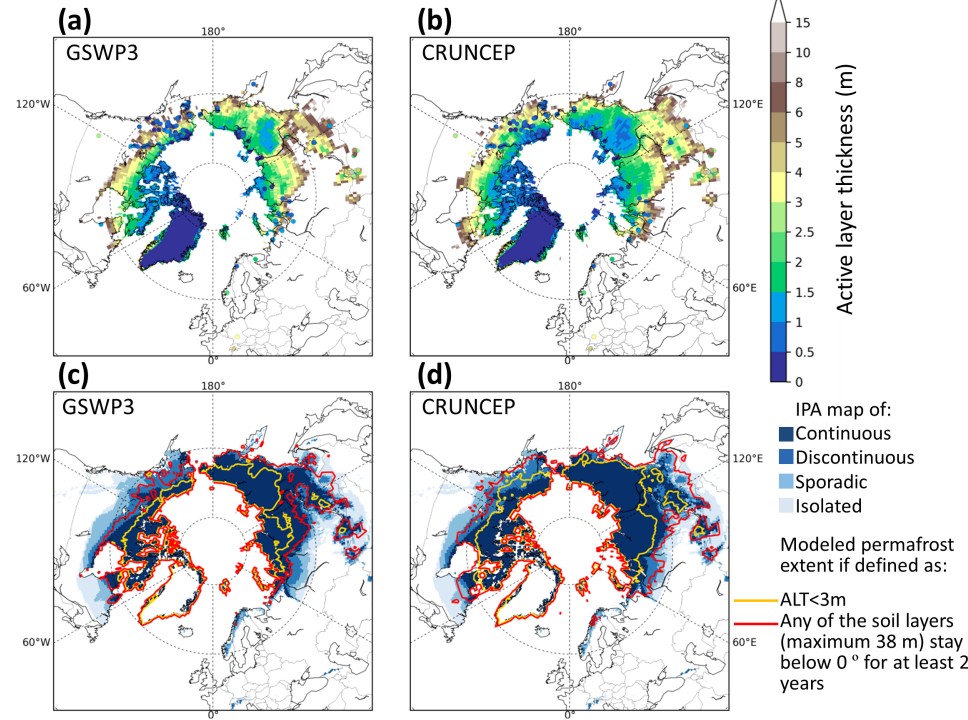

**Figure 8.** Active layer thickness (ALT in m) from the **(a)** GSWP3 and **(b)** CRUNCEP-forced simulation (background maps), compared to the observed ALT from the CALM network (color filled circles), averaged over the period 1990-2007. Permafrost extent from the **(c)** GSWP3 and **(d)** CRUNCEP-forced simulation according to two different definitions (yellow and red lines) on top of the IPA permafrost map (Brown et al., 2002).



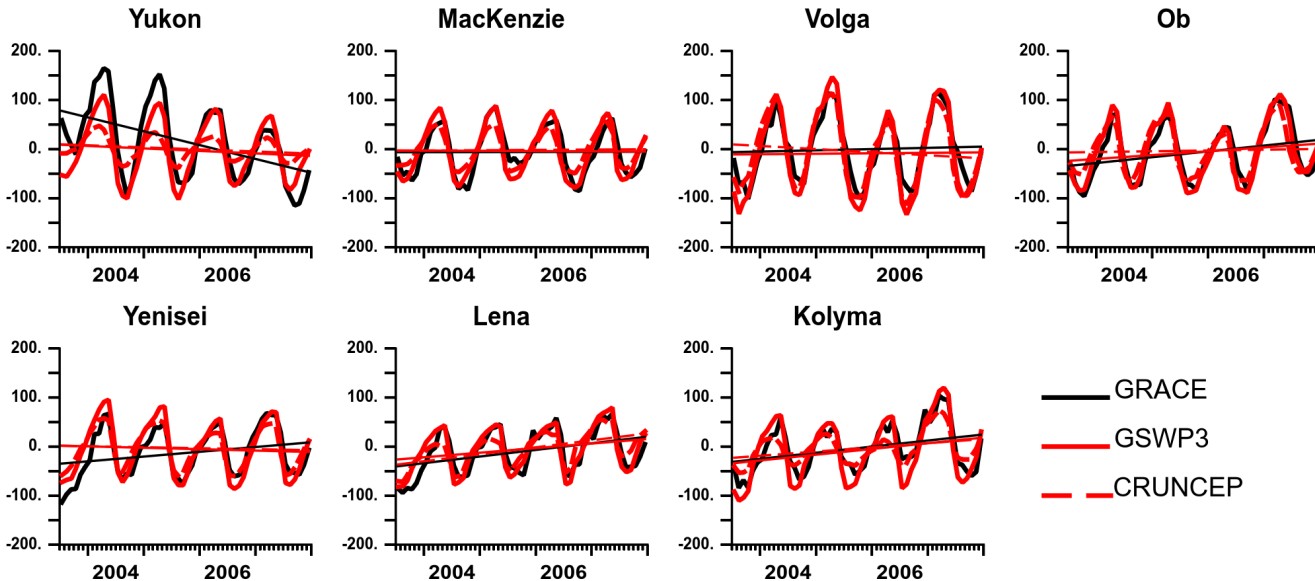

**Figure 9.** Interannual monthly variation and trend (line) of TWS (mm) simulated with the two forcings compared to GRACE data over the seven basins (see Fig. 2b), for the period July 2003-December 2007.

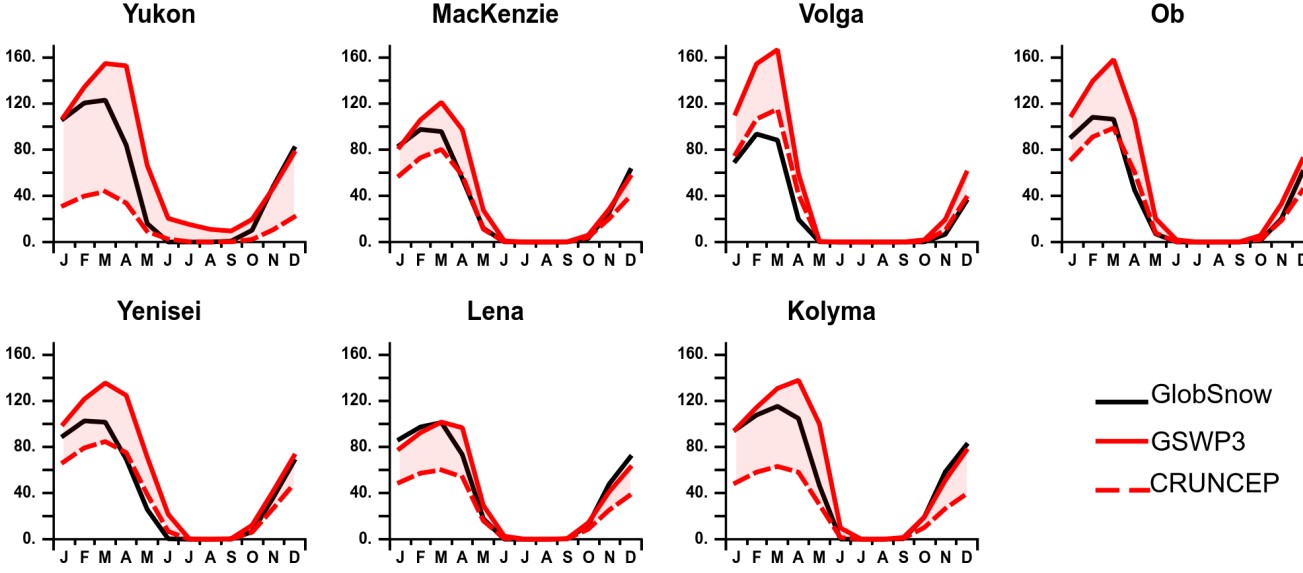

**Figure 10.** Monthly mean seasonal SWE (mm) simulated with the two forcings compared to Globsnow over the seven basins (see Fig. 2b), averaged over the period 1981-2007.





**Figure 11.** Monthly mean seasonal relative **(a)** topsoil (-) and **(b)** root soil moisture (-), both normalized by their multi-year standard deviation, simulated with the GSWP3 and CRUNCEP forcings over the seven basins (see Fig. 2b), averaged over the period 1981-2007. The results are compared with **(a)** satellite-derived observations from ESA CCI and **(b)** the GLEAM data driven model assimilated against satellite data.





**Figure 12. (a)** Monthly mean seasonal evapotranspiration (mm d$^{-1}$) simulated with the two forcings compared to GLEAM data-driven model over the seven basins (see Fig. 2b), averaged over the period 1981-2007. **(b)** Seasonal bias of ET components (mm d$^{-1}$) averaged over the same period, with the GSWP3 (solid line) and CRUNCEP (dashed line) forcings.



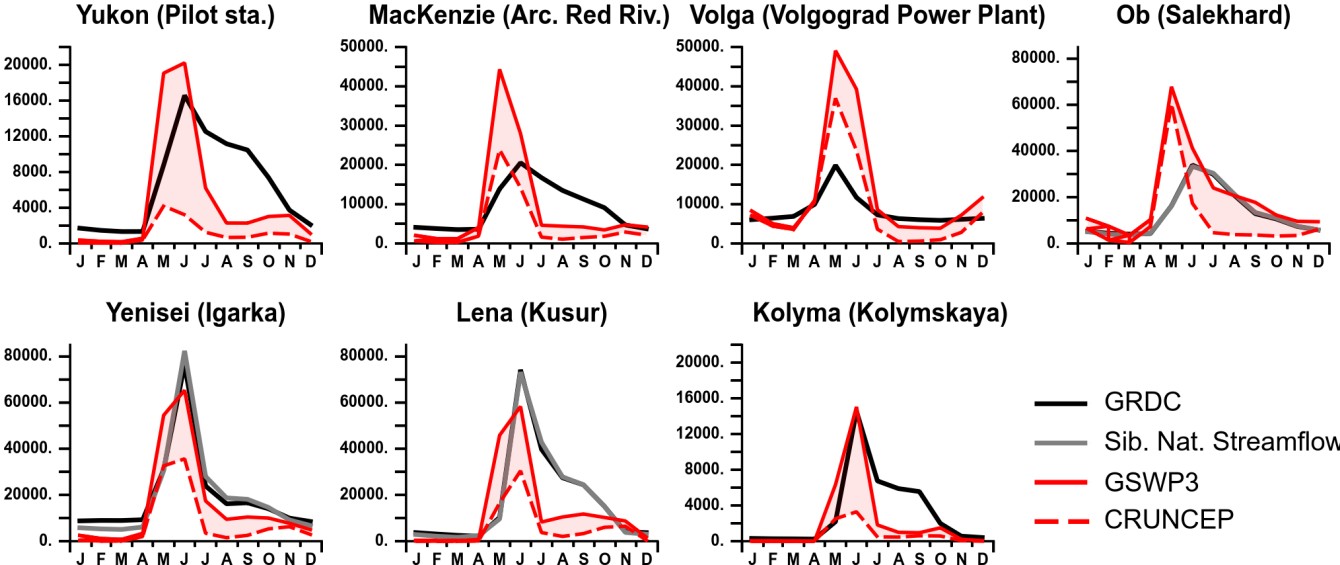

**Figure 13.** Monthly mean seasonal river discharge (m³ s⁻¹) simulated with the two forcings compared to observed non-naturalized (GRDC) and the Siberian naturalized river discharge dataset at the gauge stations of the seven basins (see Fig. 2b), averaged over the period 1981-2007.

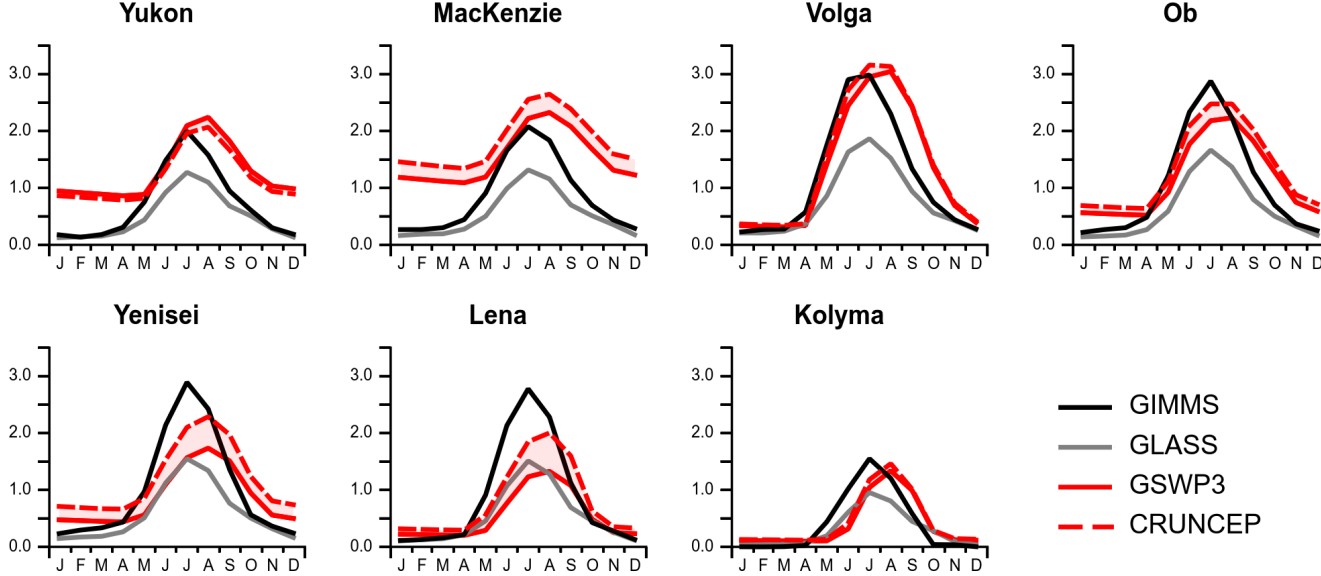

**Figure 14.** Monthly mean seasonal LAI (-) simulated with the two forcings compared to GIMMS and GLASS products over the seven basins (see Fig. 2b), averaged over the period 1981-2007.





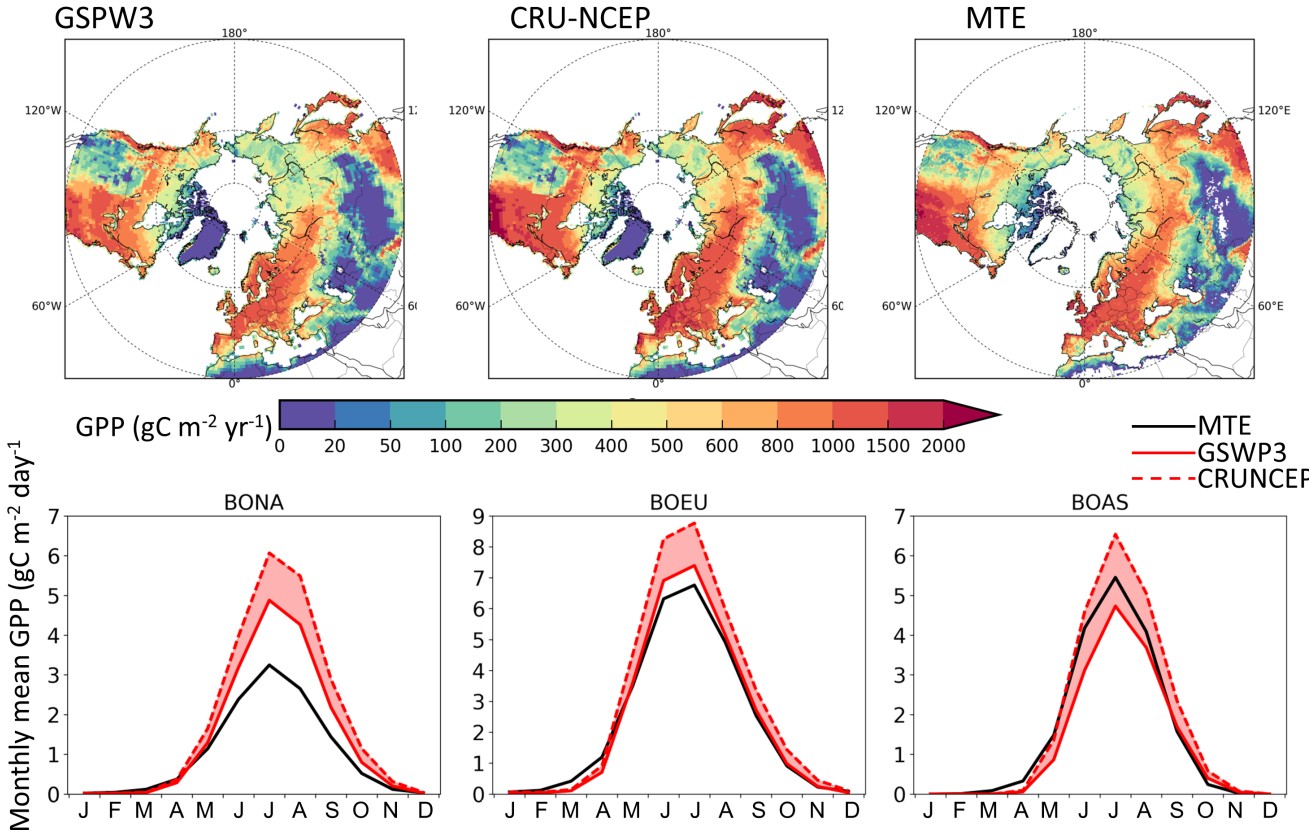

**Figure 15. (Upper panel)** Simulated annual GPP (gC m$^{-2}$ day$^{-1}$) from the two climate forcings, compared to the data-driven MTE-GPP, averaged over the period 2000-2007. **(Lower panel)** Monthly mean seasonal GPP (gC m$^{-2}$ day$^{-1}$) simulated with the two forcings compared to MTE-GPP over the three high-latitude sub-regions (shown in Fig. 2a), averaged over the same period.



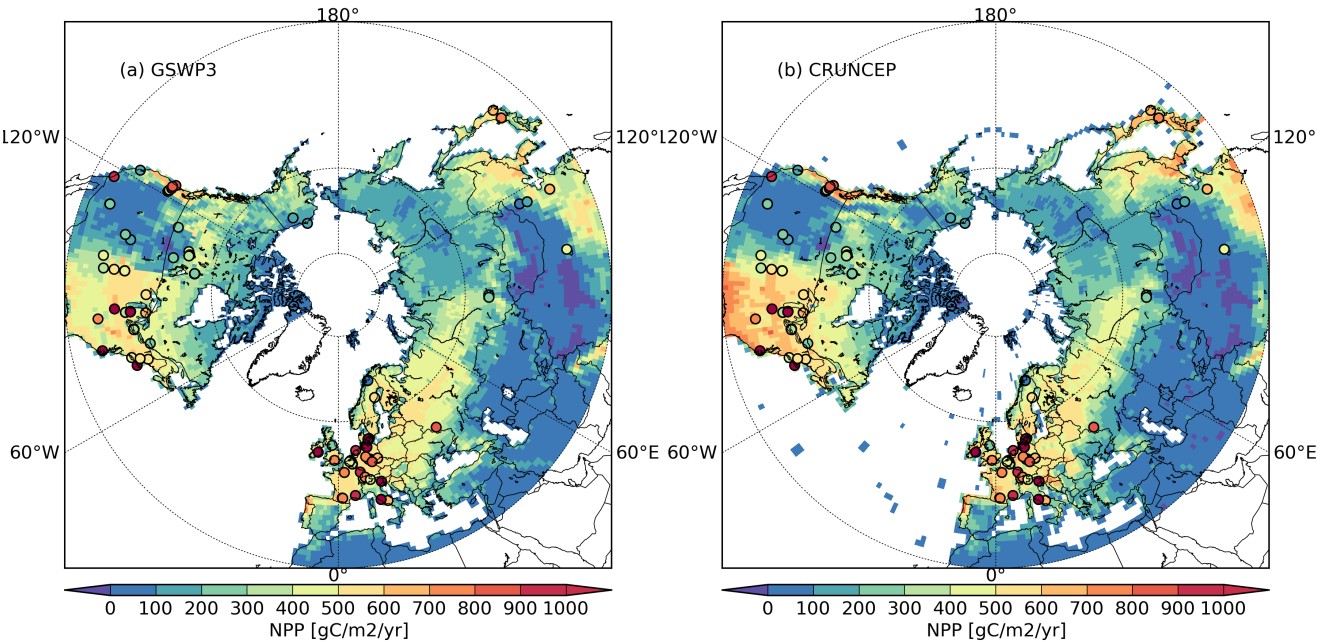

**Figure 16.** Mean annual NPP ($gC\,m^{-2}\,yr^{-1}$) simulated with **(a)** GSWP3 and **(b)** CRUNCEP forcings (background maps) compared to the site observations (color filled circles), on averaged over the period 2000-2007.

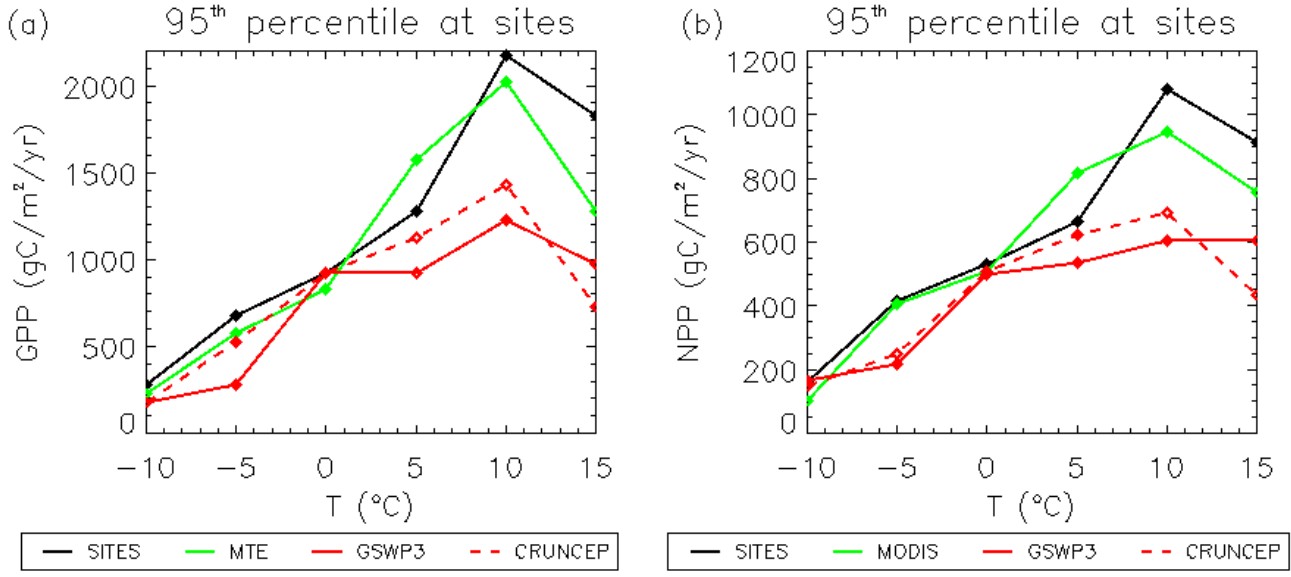

**Figure 17.** 95th percentiles of mean annual **(a)** GPP ($gC\,m^{-2}\,yr^{-1}$) and **(b)** NPP ($gC\,m^{-2}\,yr^{-1}$) distributions per temperature bins of 5°C for in situ measurements, the gridded MTE-GPP (MODIS-NPP) product sampled at the sites location and the two simulation results, averaged over the period 2000-2007.




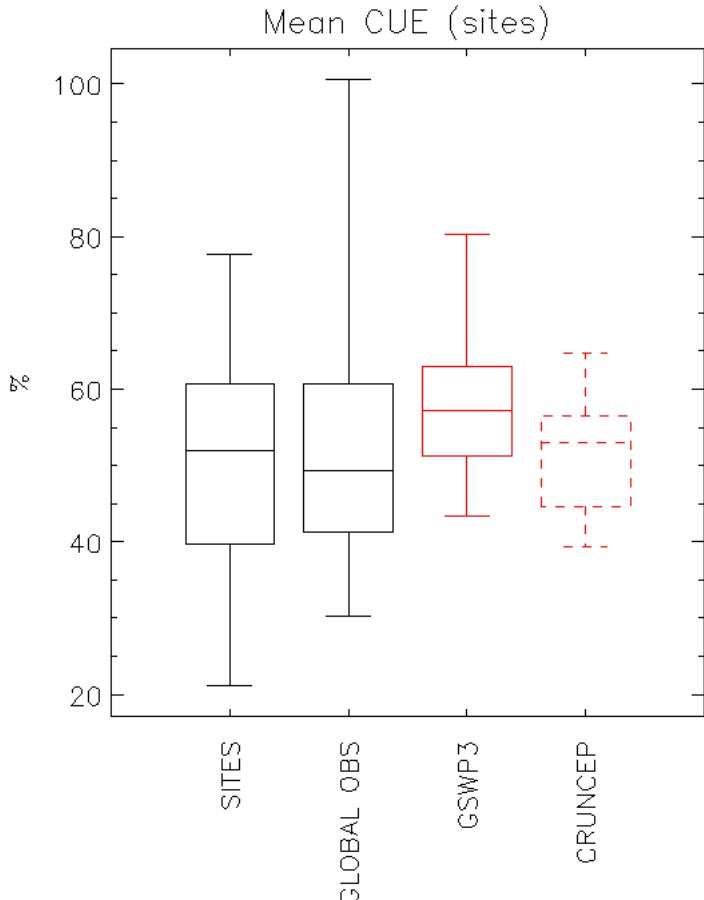

**Figure 18.** Mean annual CUE (%) over the 52 Campioli sites, averaged over the period 2000-2007. The first black boxplot is computed using the local estimations of the Campioli et al. (2015) database and the second one (global observations) using MODIS-NPP and MTE-GPP. The red boxplots uses the values of the two simulations. For each boxplot the median value is the short horizontal bar within the rectangle, whose bottom and top sides illustrate the 25th and 75th percentiles of the distribution, while the vertical segments link those sides to the points representing respectively the minimum and maximum values.



**Figure 19. (a-c)** Mean annual fractional burned area (%) from **(a)** satellite observation in GFED4s and **(b)** GSWP3 and **(c)** CRUNCEP-forced simulations. **(d-f)** Mean annual carbon emissions from natural fires (gC m$^{-2}$ yr$^{-1}$) from **(d)** satellite observation in GFED4s and **(e)** GSWP3 and **(f)** CRUNCEP-forced simulations, averaged over the period 1997-2007. The burned area fraction simulated from ORCHIDEE-MICT is corrected for the omission of cropland fires in the simulation.



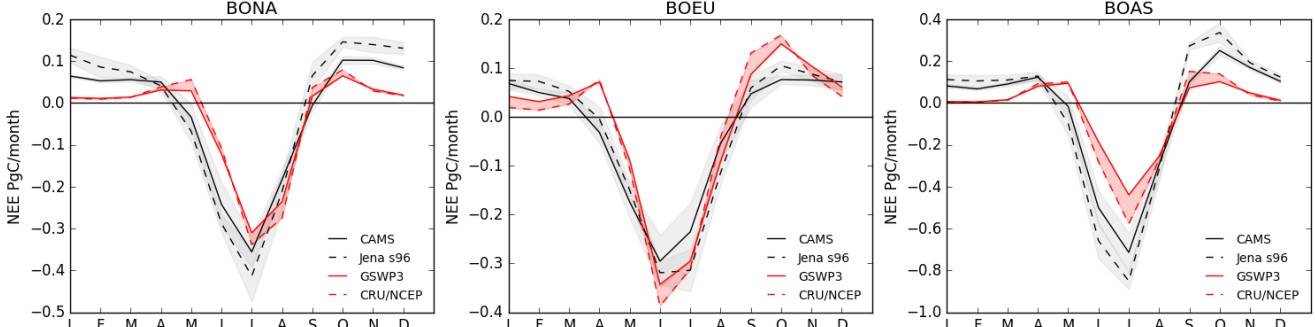

**Figure 20.** Monthly mean seasonal net land-atmosphere $CO_2$ fluxes (NEE in $PgC\,month^{-1}$) from the GSWP3 and CRUNCEP-forced simulations compared to atmospheric inversions, over the three high-latitude sub-regions (shown in Fig. 2a), averaged over the period 2000-2007. The grey shaded areas corresponds to the standard deviation of the monthly values for each inversion. Note that a negative sign in NEE corresponds to $CO_2$ uptake from the atmosphere, and a positive sign to release to the atmosphere.

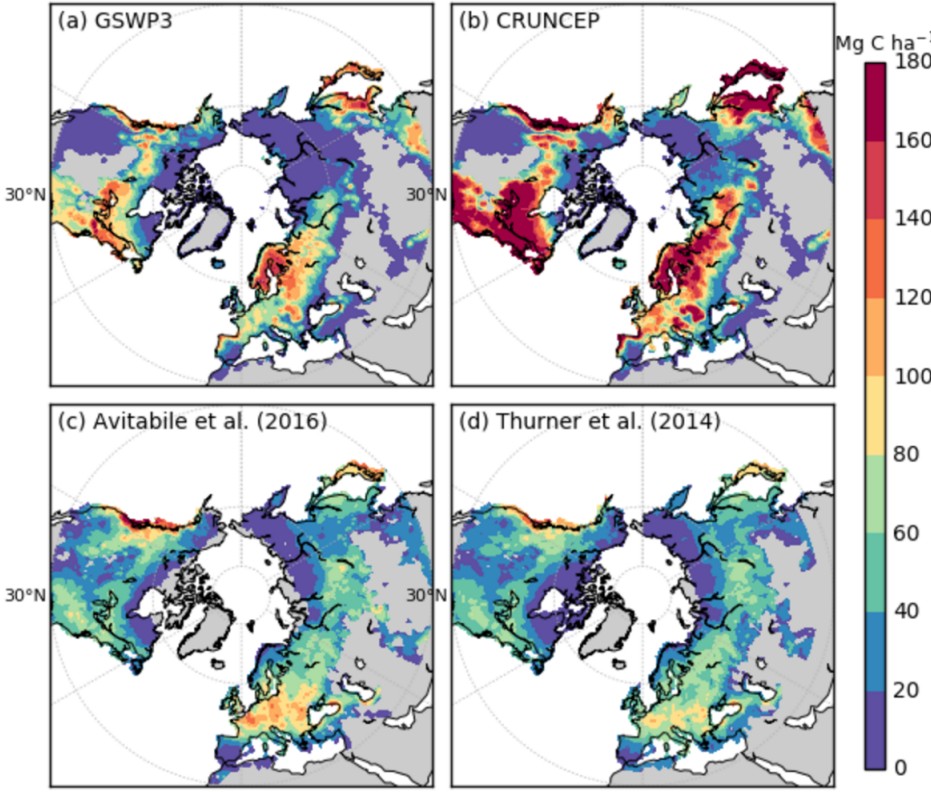

**Figure 21.** Total forest biomass carbon density ($MgC\,ha^{-1}$) from the **(a)** GSWP3 and **(b)** CRUNCEP-forced simulations compared with satellite-derived observation products from **(c)** Avitabile et al. (2016) and **(d)** Thurner et al. (2014), averaged over the period 2000-2007.





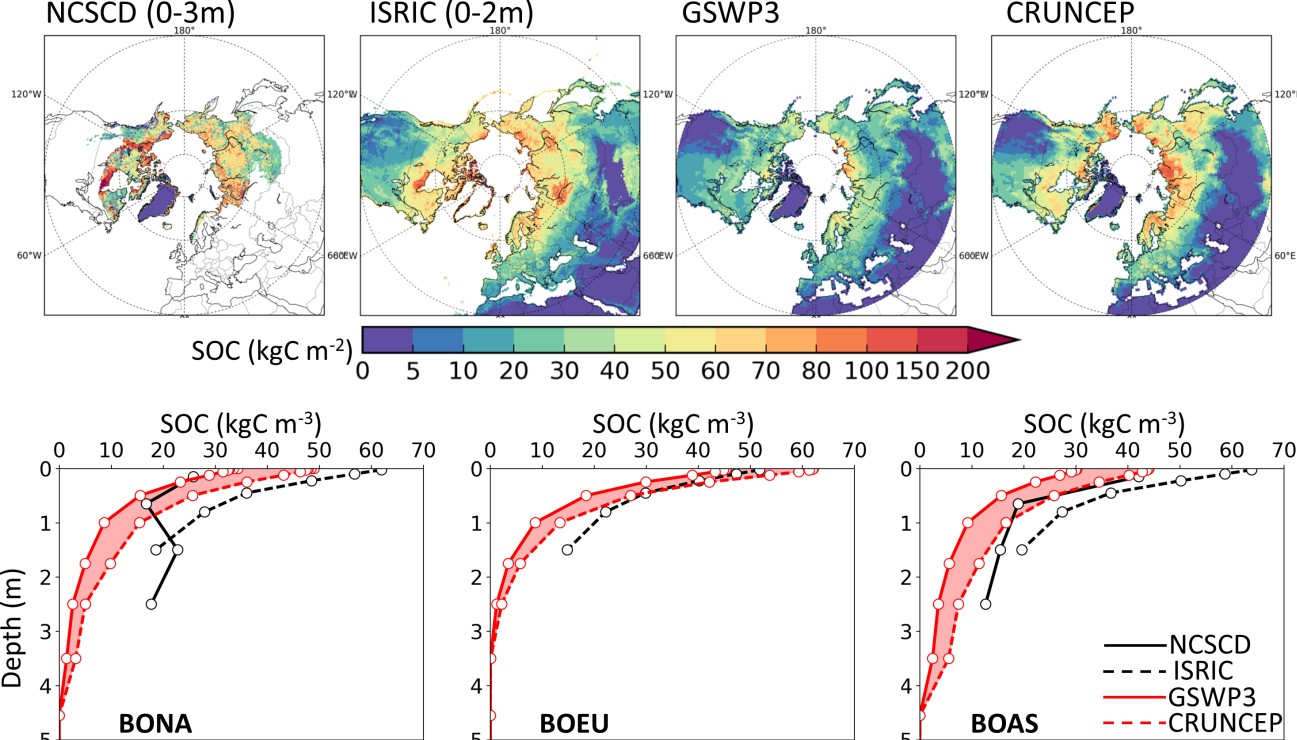

**Figure 22.** Soil organic carbon from the GSWP3 and CRUNCEP-forced simulations compared with the two inventory datasets NCSCD (Hugelius et al., 2013) and ISRIC, averaged over the period 2000-2007. (**Upper panel**) Spatial distribution (kgC m$^{-2}$). (**Lower panel**) Vertical profiles (kgC m$^{-3}$) averaged over the three high-latitude sub-regions (shown in Fig. 2a). Since NCSCD does not encompass the whole domain, only grid cells with available data in NCSCD are averaged for BONA and BOAS so that the four vertical profiles are comparable; while for BOEU, NCSCD is not shown as it has few data in this sub-region.



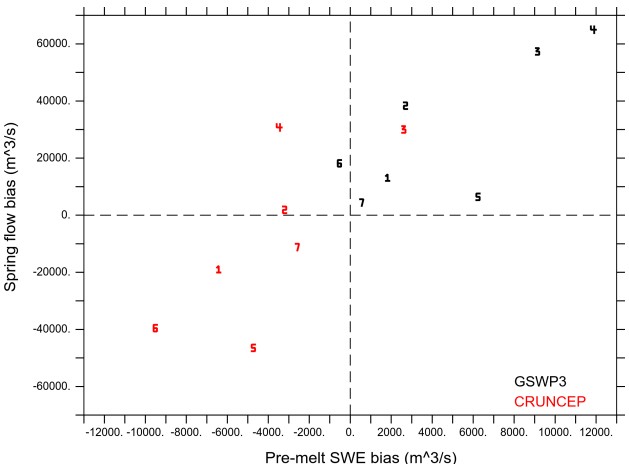

**Figure 23.** Scatter plot of the pre-melt SWE bias (m³ s⁻¹) defined as the difference between simulated and observed values during the pre-melt season (between February and March) vs. the spring river discharge bias (total discharge at the mouth of the river between April and June) (m³ s⁻¹), averaged over the period 1981-2007. Each number corresponds to one basin: 1: Yukon, 2: Mackenzie, 3: Volga, 4: Ob, 5: Yenisei, 6: Lena and 7: Kolyma (see Fig. 2b). One color represents the result with one atmospheric forcing.

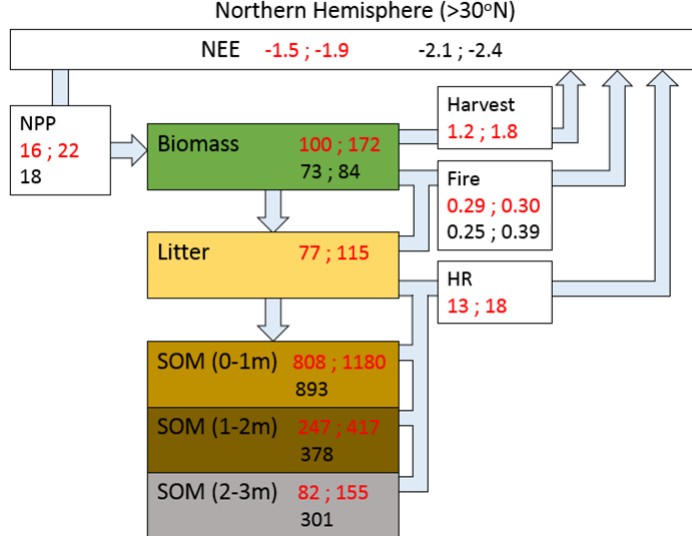

**Figure 24.** Annual mean land carbon fluxes (PgC yr⁻¹) and pool sizes (PgC), averaged over the terrestrial domain higher than 30°N, for the period 2000-2007. The red numbers are the results of the model forced by (left) GSWP3 and (right) CRUNCEP. The black numbers are from observation-based estimates used in the text : NPP from MODIS (NTSG), fire emissions from (left) GFED4s and (right) GFAS datasets, harvest fluxes include crop harvest and wood product decay in the model (for simplicity the change in wood product pools is not represented), NEE from the two atmospheric inversions (left) Jena CarboScope and (right) CAMS, forest biomass from (left) Avitabile et al. (2016) and (right) Thurner et al. (2014), soil carbon from NCSCD (Hugelius et al., 2013) in permafrost regions and HWSD in non-permafrost regions.