# Peer review of "ORCHIDEE-MICT (v8.4.1), a land surface model for the high-latitudes: model description and validation"

_Geoscientific Model Development, 2017_

## Short Comment (SC1) · 11 Jul 2017

The link https://forge.ipsl.jussieu.fr/orchidee/browser/branches/ORCHIDEE-MICT nominated as the access to the code in the manuscript does not work. It seems to require an login. This needs to be fixed.

It is also a bit usual to use an SVN version number to refer to a model version. I would like to suggest that the authors tag the particular revision in SVN and use this tag as a reference instead. Obviously releases of ORCHIDEE have been tagged before although the last release seems to be a few years old. Nevertheless I am strongly encouraging the authors to do this and to change the title accordingly. In order to

guarantee persistent access to the release the use of a DOI is strongly encouraged but not enforced.

It is also recommended to add a brief statement on license of use.

If possible we also strongly encourage authors to provide data and scripts as supplements for the manuscript again in order to guarantee persistent access to the information.

Lutz Gross GMD Executive Editor

---

## Referee Comment (RC1) · Anonymous Referee #1 · 3 Aug 2017

This paper presents and evaluates the ORCHIDEE "high latitude" model version which is known as ORCHIDEE-MICT. The difference compared with the trunk version of OR-CHIDEE is a vertically discretised soil carbon scheme and the coupling of this scheme to the soil thermal/hydrological properties, along with a representation of fire. What I really like about the paper is the coincident evaluation of so many variables (in order to correctly interpret interrelated biases) and the use of multiple datasets for the same variable when those are available (in order to give an idea of observational uncertainties). It is also great that two different forcing datasets were used, and I think it is a valuable conclusion that the uncertainty in forcing datasets needs to be taken into account to avoid "over-calibrating".

[Figure]

Clearly, this paper is extremely long. I think a suggestion of splitting it could be rightly met with the argument that all of the components are interacting so it would be difficult to split. However, depending on what the other reviewers or editors think, there could be a reasonable split into two linked papers that cover thermal/hydrological processes (in one paper) and carbon cycle processes (in another). Given the size of the work, it is well written so that it does not become too confusing to the reader. So for now (aside from the idea of splitting into two papers) I suggest only minor changes.

1. A theme that runs through the paper is the late response of LAI in the spring. The reason that is suggested (several times) in the paper is that this could be linked to the late persistence of snow cover. However, from my experience of such land surface models, they often don't incorporate a direct influence of snow on vegetation - perhaps this is incorporated in ORCHIDEE? But if so, can you make it more clear in the paper how the snow cover influences the vegetation in the model? The late LAI in spring also occurs in ORCHIDEE simulations where the snow does not stay too late (Chadburn et al., 2017), and I have heard that bud burst is simply triggered by the number of growing degree days and requires a rather large number to initiate bud burst. Therefore, I suggest you consider this alternative (possibly, more likely?) explanation, and recommend further study on the phenology scheme.

2. Another bias is the deep active layer, which really suggests that the soil properties should be better representative of the organic carbon content. The high organic content at the surface is quite well simulated (Figure 22), and this should have a great impact on the soil temperature. I would suggest that the problem might be the use of the linear weighted average for thermal conductivity. In terms of water and ice, the geometric mean is used - Equation 4 - so it might make sense to use that form for weighting the soil thermal conductivity as well, particularly the dry soil thermal conductivity which can be very low for an organic soil.

In the seasonal cycle of NEE is an unrealistic peak of emissions in spring, which can be partly due to the late LAI already discussed, but also partly because of soil decom-

position starting too early. You talked about CO2 trapping in the soil (P34 Line16) but in fact it may be much more simply that the ground is thawing too quickly - again, due to the lack of thermal insulation from the organic layer. Although when the seasonal cycle of soil temperature is studied (Figure 6), this is not obviously the case, I think there might be a bias in the Russian dataset as it seemed to behave differently from other in-situ data. The problem is potentially with the removal of vegetation from the surfaces and site disturbance, which can result in the insulation of the 'organic layer' being removed (Frauenfeld et al., 2004). Certainly when comparing with in situ data in Chadburn et al. (2017), the soil in ORCHIDEE is thawing too early and likewise the soil respiration starts a bit too early in the spring.

I suggest adding discussion of the above points relating to the link between soil carbon and soil thermal properties.

3. These two issues that I have mentioned: The phenology and the organic soil thermal properties, both seem quite important to me, and worthy of being mentioned in the conclusion, along with the issue of snow thermal conductivity which is certainly too high. These issues are extensively discussed in the text but not mentioned I the conclusions. (In particular, the organic soil properties are the 'new' process that is included in the paper so it seems important to include them in the conclusion.)

It seems to me that the other processes are appropriately discussed (at least, as far as my expertise goes: I can't comment on fires or say much about forests.)

I would like to suggest some kind of reduction in the text, as the same points are sometimes made a few times, but it is hard to envisage how to do this- I'm sure you have thought about the same thing! However, to shorten I suggest at least moving Figure 4 to the supplementary as it doesn't contain observations and doesn't seem so informative as the others.

Small comments: P29 Lines 25: "SOC stocks simulated by the model fit the spatial pattern from observed inventory data" - this does not seem convincing to me looking at

Figure 22, I think this statement should be more qualified e.g. 'to some extent' ! P28, line 27/8: just says "see 7" - should this be "see Figure 7"?

---

## Referee Comment (RC2) · Anonymous Referee #2 · 4 Aug 2017

The manuscript presents a new version of the global land surface model ORCHIDEE, which aims at a more realistic representation of hydrological processes and carbon fluxes at high latitudes and is called ORCHIDEE-MICT. To this end, several new components are introduced, such as a vertical soil carbon profile, influence of soil carbon on soil thermal properties, and a revised scheme for plant water stress. The new model is thoroughly evaluated by comparing multiple variables, such as snow properties, soil moisture and temperature, runoff and evapotranspiration, GPP, NPP, biomass and soil carbon. Explanations for mismatches between model estimates and observations are provided. The paper is well written and of good scientific quality. I therefore recommend to publish it with some minor revisions (see below).

[Figure]

General comments

(1) I agree with reviewer #1 that the paper is quite long, but I do not think it is necessary to split the manuscript. Instead, I suggest replacing the last paragraph on page 4 with a table or schematic, similar to a table of contents, which lists the sections of the paper. This might help the reader to get a better overview of the paper.

(2) The simulated soil temperatures show, in general, a cold bias, only for one soil depth in combination with the GSWP3 forcing data the temperature is overestimated. However, ALT is significantly overestimated, which does not make much sense to me. Even if soil thermal conductivity was overestimated, leading to a overestimation of ALT, this does not explain why soil temperature is underestimated. Could you please explain this a bit more? Furthermore, I do not understand why spatial heterogeneity should lead to underestimated ALT in the field measurements (page 22, line 14)?

(3) In the abstract, it is stated that the new processes put ORCHIDEE-MICT at the forefront of land surface modelling at high latitudes. However, I would expect more comparison at the process level to other models (e.g. JULES or JSBACH) to substantiate this statement, maybe through a short paragraph in the discussion. I would also like to know why the inclusion of an organic layer or a moss/lichen layer, which has been done in JULES (Chadburn et al, 2015, TC) and JSBACH (Porada et al, 2016, TC) was not considered? Could you please explain in this context the relation of ORCHIDEE-MICT to another ORCHIDEE version which is currently in review in GMD (Druel et al, 'Towards a more detailed representation of high-latitude vegetation in the global land surface model ORCHIDEE (ORC-HL-VEGv1.0)') ?

(4) I agree with the authors that the new processes implemented in ORCHIDEE-MICT should improve the model performance at high latitudes. However, I did not find in the manuscript any comparison with the previous ORCHIDEE version in this regard. Could you please show with 2 or 3 examples how ORCHIDEE-MICT represents an improvement over the previous version, e.g. with respect to simulation of runoff, snow

patterns, carbon fluxes etc.?

Specific comments

p.3,l.4: Is the correct buildup of soil carbon pools during the spin-up the only important factor for the correct short-term (100yr) prediction of soil carbon fluxes? I would argue that accurate representation of decomposition is at least equally important. Please explain shortly in the discussion why you did not revise the decomposition scheme.

p.4,l.3: If transpiration is calculated per PFT, some averaging has to take place in order to calculate the energy balance per grid cell; Please explain.

p.5,l.13: To what extent does soil water content fluctuate in 11th soil layer? If significant changes occur, these are transferred with unlimited speed through the whole soil column down to 38m. This may lead to unrealistic dynamics of thermal properties.

p.9,l.14: Why do you assume that the residual soil moisture and the Van Genuchten coefficients are independent of soil carbon content? Does soil carbon have no effect on soil texture? Please explain shortly.

p.24,l.10: It is suggested that low speed of infiltration is the reason for the underestimation of soil water content in the deep soil. In addition to the mentioned deficiencies in the representation of infiltration into frozen soils, I would like to know whether the sensitivity of deep soil moisture to the parametrisation of soil hydraulic conductivity was tested? I think the importance of the soil water deficit should be pointed out a bit more in sect. 7, since it has far-reaching effects such as reduced transpiration, increased surface temperature, and reduced productivity.

p.26,l.30 Why is peak GPP overestimated? Please explain.

p.27,l.4: NPP is underestimated (Fig 16) due to water stress and lack of nitrogen fertilisation. GPP, however, is overestimated, which should lead to an underestimation of CUE. However, CUE is overestimated, and this is explained with a lack of nutrient limitation ('too much' nutrients), which is inconsistent with the lack of nitrogen ('too little'

**[GMDD](GMDD)**

Interactive
comment

nutrients) mentioned above. Please explain.

p.28,l.30: The respiration could also originate from a moss/lichen layer which may show some activity at low temperatures and under snow.

p.29,l.6: Biomass is significantly overestimated (see also Fig S5), and, in my opinion, the difference between climate forcing data sets cannot explain this easily: Precipitation in CRUNCEP is lower than in GSWP3, but biomass is higher, which seems counter-intuitive. Could you please explain?

p.32,l.23: I think biomass compares well to observations only for GSWP3 forcing (Fig 21).

p.33,l.18: The model already has a cold bias, so even lower soil temperatures would be required for a more realistic (higher) soil carbon content. Why do you not mention explicit simulation of cryoturbation as a potential missing process?

supplement: Please indicate if CRUNCEP is subtracted from GSWP3 or vice versa.

---

## Author Comment (AC1) · 1 Oct 2017

Executive editor's comments are in bold. Modifications done in the new submitted version of the manuscript are in red.

**The link https://forge.ipsl.jussieu.fr/orchidee/browser/branches/ORCHIDEE-MICT nominated as the access to the code in the manuscript does not work. It seems to require a login. This needs to be fixed.**

The access of the source code is restricted. A login and password are thus necessary. We modified the "Code availability" section to include this information: "The source code for ORCHIDEE-MICT version 8.4.1 is available online but its access is restricted. Consequently, it is required to communicate to the corresponding author for a username and password. The source code can be found at the following address:
https://forge.ipsl.jussieu.fr/orchidee/browser/branches/ORCHIDEE-MICT/tags/ORCHIDEE_MICT_8.4.1"

**It is also a bit usual to use an SVN version number to refer to a model version. I would like to suggest that the authors tag the particular revision in SVN and use this tag as a reference instead. Obviously releases of ORCHIDEE have been tagged before although the last release seems to be a few years old. Nevertheless I am strongly encouraging the authors to do this and to change the title accordingly. In order to guarantee persistent access to the release the use of a DOI is strongly encouraged but not enforced.**

We tagged the version of ORCHIDEE-MICT used in this study into ORCHIDEE-MICT v8.4.1. We changed the title accordingly.

**It is also recommended to add a brief statement on license of use.**

We added a new section called "Code license" after the "Code availability" one: "This software is governed by the CeCILL  license under French law and abiding by the rules of distribution of free software.  You can  use, modify and/ or redistribute the software under the terms of the CeCILL license as circulated by CEA, CNRS and INRIA at the following URL:  http://www.cecill.info.".

**If possible we also strongly encourage authors to provide data and scripts as supplements for the manuscript again in order to guarantee persistent access to the information.**

The code is tagged and the running environment is also versioned. All input files are stored on four data repositories that are synchronized every night. The meteorological forcing files are versioned; the CRU-NCEP files are freely available but not the GSWP3 ones. We are working within the IPSL (Institut Pierre Simon Laplace) framework, which is preparing the CMIP6 simulations, so, even if nothing is perfect, we are working with pretty high standards of reproducibility. Input and output files for these simulations amount to several hundred of Gb, so it is quite difficult to provide them. Even a dedicated data publisher, as for example PANGAEA (https://pangaea.de/), mentions: 'max. size per file = 100 MB' (https://wiki.pangaea.de/wiki/Data_submission).

---

## Editor Comment (EC1) · H. Sato (Editor) · 26 Oct 2017

Dear Dr. Guimberteau,

I am requesting an anonymous reviewer to check whether you addressed all of their concerns adequately. Although you have shorten the main body of the manuscript much, it is still long. So, for reducing the required effort for checking it, I asked this work to only one of the two previous referees.

Well, I have questions on the revised manuscript. Page and line numbers correspond to the revised manuscript.

[Figure]

(1) Page 17, Lines 6∼7: This sentence seems to refer the Fig 10b, not the Fig 11b. Anyhow, I cannot find any obvious differences in root-soil-moisture between GLEAM and simulations of ORCHIDEE-MICT.

(2) Page 25, Line 12∼14: This sentence should refer the Fig 6.

Best, Hisashi SATO, topical editor of the GMD

---

## Author Response (AR1)

**RESPONSE TO THE REFEREES COMMENTS**

Title: ORCHIDEE-MICT (revision 4126), a land surface model for the high-latitudes: model description and validation
Author(s): Matthieu Guimberteau et al.
MS No.: gmd-2017-122
MS Type: Model evaluation paper
Iteration: Revised Submission

We would like to express our gratitude towards the two anonymous Referees for their constructive comments. We value very much their help in our effort toward a revised version of the manuscript which is at the end of this document. In the following, we write our point by point response.

- Reviewer's comments are in bold
- Modifications done in the new submitted version of the manuscript are in blue
- Figure and Table numbers, line numbers and pages all correspond to the initial manuscript version

**REFEREE #1**

**This paper presents and evaluates the ORCHIDEE "high latitude" model version which is known as ORCHIDEE-MICT. The difference compared with the trunk version of ORCHIDEE is a vertically discretized soil carbon scheme and the coupling of this scheme to the soil thermal/hydrological properties, along with a representation of fire. What I really like about the paper is the coincident evaluation of so many variables (in order to correctly interpret interrelated biases) and the use of multiple datasets for the same variable when those are available (in order to give an idea of observational uncertainties). It is also great that two different forcing datasets were used, and I think it is a valuable conclusion that the uncertainty in forcing datasets needs to be taken into account to avoid "over-calibrating". Clearly, this paper is extremely long. I think a suggestion of splitting it could be rightly met with the argument that all of the components are interacting so it would be difficult to split. However, depending on what the other reviewers or editors think, there could be a reasonable split into two linked papers that cover thermal/hydrological processes (in one paper) and carbon cycle processes (in another). Given the size of the work, it is well written so that it does not become too confusing to the reader. So for now (aside from the idea of splitting into two papers) I suggest only minor changes.**

We agree that the paper is long but we think that splitting it into two linked papers would not be necessary, as mentioned by the reviewer #2 in his first general comment. Following the comments of this reviewer, we propose to remove the last paragraph on page 4 and replace it by a table of

content between the abstract and the introduction. Moreover, Figure 4 will be moved to the supplementary section, as you suggested in point 3. We propose also to put the description of the evaluation datasets (sections 5.2 to 5.4, P.12 to 20) into an appendix section after the conclusions for a better clarity of the paper.

**1. A theme that runs through the paper is the late response of LAI in the spring. The reason that is suggested (several times) in the paper is that this could be linked to the late persistence of snow cover. However, from my experience of such land surface models, they often don't incorporate a direct influence of snow on vegetation - perhaps this is incorporated in ORCHIDEE? But if so, can you make it more clear in the paper how the snow cover influences the vegetation in the model? The late LAI in spring also occurs in ORCHIDEE simulations where the snow does not stay too late (Chadburn et al., 2017), and I have heard that bud burst is simply triggered by the number of growing degree days and requires a rather large number to initiate bud burst. Therefore, I suggest you consider this alternative (possibly, more likely?) explanation, and recommend further study on the phenology scheme.**

The reviewer is right, the snow cover is not directly taken into account in the vegetation phenology models used for the different PFTs of ORCHIDEE. After the sentences on Page 26, Lines 13-15: "In all the basins, the LAI simulated by ORCHIDEE-MICT has a phase delay of up to one month compared to both products. This is due to a delay in the start of the growing season, which may be related to excessive persistence of the snow cover (Fig. 10)" in section 8.1, we thus detailed the arguments to support our hypothesis: "The phenological models in ORCHIDEE (detailed in MacBean et al., 2015, Appendix A) do not explicitly take into account this influence, unlike what is done in Van Wijk et al. (2003), who model the link of the start of the tussock tundra growing season to the soil thaw at 10-cm depth. However there is a first indirect link between the snow cover and the vegetation phenology through air temperature, which influences both the start of the growing season, determined in ORCHIDEE using growing degree days (GDD)-based phenological models for deciduous species, and the start of the snowmelt season. There is a second indirect link through snowmelt. While there is still a large amount of snow, the soil surface temperature is kept at zero degree Celsius or below and the soil cannot thaw. Only when snowmelt occurs and when the snow fraction is small enough, will the soil start thawing, thus increasing soil liquid water content. This impacts the start of the growing season for grasses and crops, which use both a GDD and a soil moisture thresholds, and also reduces water stress, thus favoring photosynthesis for all PFTs."

In the discussion part, section 10.3, after the sentences Lines 32-33 P.33: "The phase of simulated LAI in spring lags satellite observations, in particular for BONA and BOAS sub-regions (Fig. 14).", we nevertheless acknowledge the reference to Chadburn et al. (2017), cited by the reviewer: "We argued that this lag is related to the late persistence of the snow cover. However a recent work (Chadburn et al., 2017) shows a late onset even in the absence of snow persistence on site simulations by ORCHIDEE. This calls for a revisit of the phenology-related thresholds in the high-latitudes, perhaps by introducing new PFTs (arctic C3 grass and shrub, and non-vascular plants), with their separate set of parameters calibrated, to better represent Arctic vegetation and their phenology (Druel et al., 2017).".

**2. Another bias is the deep active layer, which really suggests that the soil properties should be better representative of the organic carbon content. The high organic content at the surface is quite well simulated (Figure 22), and this should have a great impact on the soil temperature. I would suggest that the problem might be the use of the linear weighted average for thermal conductivity. In terms of water and ice, the geometric mean is used - Equation 4 - so it might make sense to use that form for weighting the soil thermal conductivity as well, particularly the dry soil thermal conductivity which can be very low for an organic soil.**

We would like to note that the soil carbon concentrations used in the thermal and hydrological modules were prescribed from the empirical soil databases of NCSCD and HWSD (as mentioned on Page 8 Lines 15-16), similar to the treatment in other land surface models like CLM (Lawrence and Slater, 2008) and JULES (Chadburn et al., 2015). So the modeled SOC by the carbon cycle module did not yet feed back on the soil physical properties. Although the coupling between soil thermodynamics and the prognostically modeled SOC is readily achievable by changing a configuration in the model set-up, we tend to exclude the bias of the carbon cycle module in the physical processes in this study as a first step.

The simulated ALT is indeed generally overestimated compared to the site observations from CALM network and the regional ALT map for Yakutia. However, apart from the bias in the parameterization of soil thermal properties, the fact that we did not use the site-specific organic layer thickness for the CALM sites, and that we did not prescribe regionally a fixed thickness for organic layer (to mimic the insulating effect of moss layer), also contributed to the bias in modeled ALT. Now we conducted, following the comment by Reviewer #2, additional CALM site simulations in which site-specific organic layer thicknesses were prescribed. These site runs then produced significantly shallower ALTs compared to the previous regional simulation. Please refer to our second response to Reviewer #2 for details.

As for the averaging method for soil thermal conductivity, it makes sense indeed to use geometric mean, which will be tested for the next steps of model development. We added the following sentence after the sentence on Page 9 Line 2: "Note that here we followed Lawrence and Slater (2008) to use linear weighting organic and mineral soil properties, while in some other models like JULES (Chadburn et al., 2015) and ISBA (Decharme et al., 2016), soil thermal conductivities are calculated as geometric averages of organic and mineral soils, consistent with the treatment for soil water and ice (Eq. 4). The geometric averaging method increases the effect of the organic fraction compared to arithmetic averages, and would be tested in ORCHIDEE-MICT in future developments."

**In the seasonal cycle of NEE is an unrealistic peak of emissions in spring, which can be partly due to the late LAI already discussed, but also partly because of soil decomposition starting too early. You talked about $CO_2$ trapping in the soil (P34 Line 16) but in fact it may be much more simply that the ground is thawing too quickly - again, due to the lack of thermal insulation from the organic layer. Although when the seasonal cycle of soil temperature is studied (Figure 6), this is not obviously the case, I think there might be a bias in the Russian dataset as it seemed to behave differently from other in-situ data. The problem is potentially with the removal of vegetation from the surfaces and site disturbance, which can result in the insulation of the 'organic layer' being removed (Frauenfeld et al., 2004). Certainly when comparing with in situ data in Chadburn et al. (2017), the soil in ORCHIDEE is thawing too early and likewise the soil respiration starts a bit too early in the spring.**
**I suggest adding discussion of the above points relating to the link between soil carbon and soil thermal properties.**

We would like first to note that the Russian dataset for soil temperatures between 0.2 and 3.2 m depths (that we used in this study) have been measured under natural vegetation (mainly grass) and undisturbed snow; although the grasses were regularly mowed to keep their height below 20 cm (as described in http://nsidc.org/data/docs/fgdc/ggd251_soiltemp_fsu/), the main insulating effect of the organic matter near the ground surface has been kept. Therefore, bias of these measurements induced by human disturbance are generally very small.

Figure 6cd indeed shows a cold bias of soil temperature during spring; however, this figure is the average for all sites located in continuous permafrost region, while spatially, there is a warm bias in the Lena basin and a cold bias for the further eastern sites (please also see our second response to Reviewer #2), which partly offset each other in Fig. 6cd.

We agree that the spring peak in NEE may be partly explained by a too big soil respiration. But also note that the spring GPP is underestimated, especially in BOAS and BOEU (Fig. 15). To

address these issues, we modified the text on Page 34 Lines 15-17: "Interestingly the reasonable seasonal phase of simulated GPP (Fig. 14) can be contrasted with the larger lag of spring NEE uptake compared to inversions results (Fig. 19). This may be partly explained by the underestimated GPP at the beginning of the growing season (Fig. 14), and also by a possible too big soil respiration in spring. The moss/lichen surface coverage could be over 70% under the vast boreal forests (Porada et al., 2016), but we did not prescribe an additional moss layer in the regional simulations, which could lead to a too early thawing of the soil in spring."

**3. These two issues that I have mentioned: The phenology and the organic soil thermal properties, both seem quite important to me, and worthy of being mentioned in the conclusion, along with the issue of snow thermal conductivity which is certainly too high. These issues are extensively discussed in the text but not mentioned in the conclusions. (In particular, the organic soil properties are the 'new' process that is included in the paper so it seems important to include them in the conclusion.) It seems to me that the other processes are appropriately discussed (at least, as far as my expertise goes: I can't comment on fires or say much about forests.)**

We add these sentences in the conclusion:
- P. 34 L.27 "... in the ORCHIDEE-MICT land surface model. The effects of soil organic matter on soil thermal and hydraulic properties are incorporated."
- P. 34 L.30 "Naturally, there remains significant room for improvement. The model appears to underestimate evapotranspiration and overestimate surface temperature, particularly in the southern portion of the boreal zone. Simulated phenology shows generally a delay in the onset of growing season. And the snow module underestimates the thermal insulation of snow."

**I would like to suggest some kind of reduction in the text, as the same points are sometimes made a few times, but it is hard to envisage how to do this- I'm sure you have thought about the same thing! However, to shorten I suggest at least moving Figure 4 to the supplementary as it doesn't contain observations and doesn't seem so informative as the others.**

Yes, Figure 4 is moved to the supplementary section. We also propose to remove the last paragraph on page 4 and replace it by a table of content between the abstract and the introduction. We put the description of the evaluation datasets (sections 5.2 to 5.4) into an appendix section after the conclusion for a better clarity of the paper.

**Small comments:**

**P29 Lines 25: "SOC stocks simulated by the model fit the spatial pattern from observed inventory data" - this does not seem convincing to me looking at Figure 22, I think this statement should be more qualified e.g. 'to some extent' !**

Yes, we revised it as : "SOC stocks simulated by the model fit to some extent the spatial pattern from observed inventory data ...".

**P28, line 27/8: just says "see 7" - should this be "see Figure 7"?**

Corrected now in the text.
* * *
**REFEREE #2**

**The manuscript presents a new version of the global land surface model ORCHIDEE, which aims at a more realistic representation of hydrological processes and carbon fluxes at high latitudes and is called ORCHIDEE-MICT. To this end, several new components are introduced, such as a vertical soil carbon profile, influence of soil carbon on soil thermal properties, and a revised scheme for plant water stress. The new model is thoroughly evaluated by comparing multiple variables, such as snow properties, soil moisture and temperature, runoff and evapotranspiration, GPP, NPP, biomass and soil carbon. Explanations for mismatches between model estimates and observations are provided. The paper is well written and of good scientific quality. I therefore recommend to publish it with some minor revisions (see below).**

**General comments**

**(1) I agree with reviewer #1 that the paper is quite long, but I do not think it is necessary to split the manuscript. Instead, I suggest replacing the last paragraph on page 4 with a table or schematic, similar to a table of contents, which lists the sections of the paper. This might help the reader to get a better overview of the paper.**

Thank you for this good suggestion. We will remove the last paragraph on page 4 and replace it by a table of content between the abstract and the introduction. Moreover, Figure 4 is moved to the supplementary section, as suggested by the Reviewer #1. We propose also to put the description

of the evaluation dataset (sections 5.2 to 5.4) into an appendix section after the conclusion for a better clarity of the paper.

**(2) The simulated soil temperatures show, in general, a cold bias, only for one soil depth in combination with the GSWP3 forcing data the temperature is overestimated. However, ALT is significantly overestimated, which does not make much sense to me. Even if soil thermal conductivity was overestimated, leading to a overestimation of ALT, this does not explain why soil temperature is underestimated. Could you please explain this a bit more? Furthermore, I do not understand why spatial heterogeneity should lead to underestimated ALT in the field measurements (page 22, line 14)?**

Simulated ALT is indeed generally overestimated compared to the site observations from CALM network and the empirically-derived map for Yakutia (Beer et al., 2013). This seems inconsistent with the cold bias in soil temperature compared to the Russian meteorological stations' observations as shown in Fig. 6. However, we would like to clarify several points:

First, there is a mismatch for the locations of CALM sites for ALT and of Russian stations for soil temperature, the former mostly within the Arctic regions near the coast (for the Eurasian sites), and the latter more scattered throughout Russia. Therefore, the slight overestimation in peak summer-time soil temperature (and even underestimation with the CRUNCEP forcing) shown in Fig. 6cd for the Russian sites does not necessarily contradict with the significantly overestimated ALT compared to the CALM sites.

As for the regional maps for soil temperature and ALT, we acknowledge that Fig. 6ab may be a bit misleading if readers connect this figure to Fig. S3 which shows on the contrary a generally deeper ALT compared to Beer et al., (2013). This is because Fig. 6ab shows the annual mean temperature at 0.2 m depth, in which the larger cold bias in winter outweigh the warm bias in summer. To complement Fig. 6ab, we added maps for the maximum monthly soil temperature for the four depths in the Supplementary, also shown below, to facilitate a link between bias in soil temperature and bias in ALT. This figure shows an overestimation in maximum soil temperature below 0.8 m in the Lena basin, consistent with the deeper ALT for the same region shown in Fig. S3.

[Figure]

**Figure S4.** Maximum monthly soil temperature at different depths in **(a)** GSWP3 and **(b)** CRUNCEP-forced simulation (background maps), compared to the site observations (color filled circles), averaged over the period 1981-2000.

To clarify it, we revised the following sentences on Page 21 Lines 32-33: "Summer soil temperatures are higher in the GSWP3-forced simulation relative to those of CRUNCEP, and warmer than observations from the Russian meteorological stations in continuous permafrost region by 1~2 °C on average at 0.8 and 1.6 m depths (Fig. 5c,d). Spatially however, the bias in peak summer soil temperature varies for different regions, with a large warm bias for the Lena basin below 0.8 m, and some cold bias for the further eastern sites (Fig. S4). This is consistent with the overestimation of ALT for Yakutia (Fig. S5) compared to the empirically-derived map by Beer et al., (2013) (see Section 6.4). Differences between the two simulations…" And added one sentence at the end of the caption for Fig. 6: "…over the period 1981-2000. The spatial patterns of maximum monthly soil temperature are also shown in Fig. S4."

Second, the mismatch between the local-scale ALT measurements at the CALM sites and the modeled ALT could be partly explained by the fact that we did not use the site information of the soil organic layer thickness in the calculation of soil physical properties, but used the gridded soil carbon database from NCSCD (Hugelius et al., 2013) for permafrost regions (as mentioned on Page 8 Lines 15-16), which was upscaled in the model to match the spatial resolution for each simulation (here 1° by 1°).

To further explore the impact of the site-specific organic layer thickness on modeled ALT, we conducted additional runs at the CALM sites, in which we assumed $f_{i,soc}$ in Eq. 9 equaling to one for the soil layers above the organic layer thickness at each site. While the other model inputs including climate and soil texture are the same as the previous northern hemisphere simulation. There are in total 69 sites that provide explicit organic layer thickness (https://www2.gwu.edu/~calm/data/north.html). Some sites, e.g. the Dot Lake in Alaska, also have a thick sphagnum layer above the decomposing organic soil; in such cases, we summed up their depths to derive a total organic layer thickness. The figure below shows the result, which is now added to the Supplementary.

[Figure]

**Figure S6. (a)** Scatter plot of modeled ALT forced by CRUNCEP compared with observed ALT from the CALM network, averaged over the period 1990-2007. The black circles represent the grid cells taken from the regional simulation (shown in Fig. 8b). The blue circles represent a subset of the CALM sites for which we performed additional runs using site-specific organic layer thicknesses, with the result shown by the red circles. **(b)** Illustration for the difference of the additional site simulations. Each grey arrow connects the same site, pointing from the blue circles using soil carbon content from NCSCD (upscaled at 1° by 1° resolution) to calculate soil physical properties, to the red circles using the organic layer thickness provided by the sites. See text for further information.

Accordingly, the following discussions were added on Page 22 Line 10: "…whereas CRUNCEP-forced output shows relatively better agreement with the observations. Apart from the uncertainty induced by climate forcing, the model-data mismatch may also arise from scale differences for the organic carbon content that is used to calculate soil physical properties for each grid cell. As mentioned in Section 4.2, the empirical SOC map from NCSCD (Hugelius et al., 2013) is prescribed for permafrost regions in the soil thermal and hydrological modules, which is upscaled

by the model to the target spatial resolution (1° by 1° in this study). These SOC values thus do not represent the site-level soil conditions, aside from the uncertainty of the NCSCD database itself. To further investigate this impact, we conducted additional simulations for the sites that provide explicit organic layer thicknesses (in total 69 sites), forced by CRUNCEP. In these runs, we assumed pure organic soil, i.e. $f_{i,soc}$ in Eq. 9 equaling to one, for the soil layers above the site-specific organic layer thickness, while kept the SOC concentration unchanged below this thickness, i.e. from NCSCD. Note that the moss layer, vegetation mat, and/or organic root zone as described in some sites were all summed to derive a total organic layer thickness. The other configurations including climate forcing and soil texture were the same as the regional simulation. The result is displayed in Fig. S6, showing significantly shallower ALTs simulated by these site runs which better match the observations (Fig. S6a), with different magnitudes of ALT reductions among the sites (Fig. S6b)."

**(3) In the abstract, it is stated that the new processes put ORCHIDEE-MICT at the forefront of land surface modelling at high latitudes. However, I would expect more comparison at the process level to other models (e.g. JULES or JSBACH) to substantiate this statement, maybe through a short paragraph in the discussion. I would also like to know why the inclusion of an organic layer or a moss/lichen layer, which has been done in JULES (Chadburn et al, 2015, TC) and JSBACH (Porada et al, 2016, TC) was not considered? Could you please explain in this context the relation of ORCHIDEE-MICT to another ORCHIDEE version which is currently in review in GMD (Druel et al, 'Towards a more detailed representation of high-latitude vegetation in the global land surface model ORCHIDEE (ORC-HL-VEGv1.0)') ?**

The current model described in this study indeed lacks explicit representation of moss and lichen growth. For an inclusion of organic layer, however, the multi-layer structure of the model enables it to approximate the effect of an organic layer by assuming 100% organic soil above the prescribed organic layer thickness, as we did for the additional CALM site simulations. Actually, this method could also approximate the effect of moss/lichen on soil thermodynamics, assuming similar thermal properties of the moss/lichen layer to that of the organic layer, as the implementation in JULES (Chadburn et al., 2015, GMD). Therefore, the current ORCHIDEE-MICT is able to represent the insulating effect of moss/lichen and organic layer in a simplified way, given an input of their thickness.

However, for large-scale simulations to account for moss/lichen layer, what is indeed lacking in current ORCHIDEE-MICT is a prognostic modeling of moss/lichen surface cover, considering the lack of a gridded map for moss/lichen coverage, especially in the boreal forest understory (Chadburn et al., 2015, TC; Porada et al, 2016, TC). Chadburn et al. (2015) used functions of

temperature, moisture, light and snow to empirically determine the ground cover of moss to be used in the soil thermal module, but did not simulate the carbon cycle of mosses. Porada et al., (2016) modeled the productivity and expansion of moss/lichen, which then, combined with fire disturbance, determined the dynamic surface coverage of moss/lichen.

Druel et al. (2017, GMDD) implemented an explicit representation of the high-latitude vegetation types including shrubs, boreal grasses, and non-vascular plants that are missing in standard ORCHIDEE. Processes and parameters regarding the growth of these new PFTs were defined, and the main biogeochemical results were evaluated. At the moment, their work is in parallel with the recent developments in ORCHIDEE-MICT described in this study, but could be merged within ORCHIDE-MICT in the next step.

To discuss these issues, we added a paragraph at the end of Section 10.2, Page 32 Line 15: "Previous land surface modeling studies have shown the critical role of organic matter in soil thermodynamics in permafrost regions (e.g. Lawrence et al., 2008; Chadburn et al., 2015), while different parameterizations of such effects are implemented in different models. Most of the recent models, like CLM (Lawrence et al., 2008), JULES (Chadburn et al., 2015), ISBA (Decharme et al., 2016), and ORCHIDEE-MICT in this study, assume weighted combinations of organic soil and mineral soil in the calculation of soil physical parameters for each soil layer in the model. This structure is more flexible than a fixed thickness of organic layer or moss layer as the implementation in JSBACH (Ekici et al., 2014), since the former could approximate the latter by assuming 100% organic soil above the prescribed thickness. Note that for the insulating effect of moss/lichen layer, the same values of thermal properties to that of the organic soil are usually used in recent models (Chadburn et al., 2015; Porada et al., 2016). In this study, however, we did not apply a fixed moss layer in the thermal module for the regional simulations, due to the lack of a gridded map for moss/lichen ground covers especially on the boreal forest floor, and to the lack of a representation for dynamic moss/lichen coverage as in JULES (Chadburn et al., 2015) and JSBACH (Porada et al., 2016). This could partly explain the regionally overestimated ALT compared to the empirical map for Yakutia (Fig. S5). An explicit representation for non-vascular plants in ORCHIDEE (Druel et al., 2017) has been worked in parallel with this study at the moment, but would be incorporated in ORCHIDEE-MICT in the future developments."

**(4) I agree with the authors that the new processes implemented in ORCHIDEE-MICT should improve the model performance at high latitudes. However, I did not find in the manuscript any comparison with the previous ORCHIDEE version in this regard. Could you please show with 2 or 3 examples how ORCHIDEE-MICT represents an improvement over the previous version, e.g. with respect to simulation of runoff, snow patterns, carbon fluxes etc.?**

ORCHIDEE-MICT is a new branch building on several separated former works with important processes for high latitudes for both the physical processes (e.g. Gouttevin et al., 2012a; Wang et al., 2013) and the carbon cycle (e.g. Koven et al., 2009). This paper is thus like the birth certificate of this new branch, demonstrating the effectiveness of all these combined processes in reproducing important observed variables. A comparison with the TRUNK, which does not yet incorporate all of these processes like the soil carbon discretization, would be unfair. An outcome of this paper will indeed be the integration of these high-latitude processes in the standard TRUNK version.

**Specific comments**

**p.3,l.4: Is the correct buildup of soil carbon pools during the spin-up the only important factor for the correct short-term (100yr) prediction of soil carbon fluxes? I would argue that accurate representation of decomposition is at least equally important. Please explain shortly in the discussion why you did not revise the decomposition scheme.**

We fully agree with the reviewer that a good representation of SOC is highly important to represent the carbon fluxes to the atmosphere. At some points we wondered how we could improve the scheme in ORCHIDEE-MICT, and from a short literature review (well summarized in Manzoni and Porporato, 2009, SBB, Wutzler and Reichstein 2008, BG or more recently in Luo et al. 2016 GBC) there is still no consensus in the soil science community, and the different approach with their own underlying assumptions can hardly impact the model outputs. The representation of SOC decomposition in land surface models is still under debate and within this debate we choose to be conservative and keep the scheme used for decades now based on CENTURY (Parton et al., 1988). Nevertheless, some actions are ongoing in our group to improve the SOC decomposition scheme including the impact of priming, dissolved organic carbon, etc.

We add these sentences in the discussion on Page 33 Line 18: "Other studies (Koven et al., 2013; Burke et al., 2017a) further limited the rate of decomposition of SOC at depth, to reproduce the lack of oxygen inhibiting decomposition. It should be noted also that the decomposition scheme of SOC is still based on Parton et al., (1988) as classically done in land surface models (Friedlingstein et al., 2006). Different approaches were proposed in models focusing only on SOC decomposition (Manzoni and Porporato, 2009; Wutlzer and Reichstein, 2008) based on different assumptions (substrate driven, decomposer driven, etc.), but no clear consensus emerged up to date to revise the SOC decomposition scheme in land surface models (Luo et al., 2016)."

**p.4,l.3: If transpiration is calculated per PFT, some averaging has to take place in order to calculate the energy balance per grid cell; Please explain.**

A transpiration flux is calculated for each PFT in each tile (forest and short vegetation). In each tile, a weighted spatial average of the different PFTs is performed. Then, the model calculates a weighted spatial average across the soil tiles to obtain a total representative flux of the grid-cell.

**p.5,l.13: To what extent does soil water content fluctuate in 11th soil layer? If significant changes occur, these are transferred with unlimited speed through the whole soil column down to 38m. This may lead to unrealistic dynamics of thermal properties.**

The soil water content fluctuates in a variation range given by the hydrodynamic parameters of the different soils prescribed in the model. Thus it can vary from 0.034 to 0.460 m³/m³ (residual and saturated water content for a silt soil respectively). This corresponds to a maximum variation range of 68 to 920 mm for the 2-meter depth soil of the model. Yet, the most important variations of water content occurs in the superficial layers of the soil column that are submitted to precipitation events and infiltration. The non-linear decrease of the hydraulic conductivity with soil water content leads to reasonable values of gravitational drainage at the bottom and thus cannot lead to unlimited speed through the whole soil column.

**p.9,l.14: Why do you assume that the residual soil moisture and the Van Genuchten coefficients are independent of soil carbon content? Does soil carbon have no effect on soil texture? Please explain shortly.**

Soil organic matter may indeed contribute to the variation of the Van Genuchten equation coefficients (except for α) (Wang et al., CLEAN-Soil Air Water, 2014). However, we chose as a first step to keep the coefficients unchanged, and the resulted relationship of field capacity/wilting point versus SOC (Fig. S2) could capture the first-order characters in the observations by Hudson (1994). Modifications of the Van Genuchten equation coefficients would require an in-depth sensitivity study of soil hydrology in the model. This could be considered for further developments of the model.

**p.24,l.10: It is suggested that low speed of infiltration is the reason for the underestimation of soil water content in the deep soil. In addition to the mentioned deficiencies in the representation of infiltration into frozen soils, I would like to know whether the sensitivity of deep soil moisture to the parameterization of soil hydraulic conductivity was tested? I think the importance of the soil water deficit should be pointed out a bit more in sect. 7, since it has far-reaching effects such as reduced transpiration, increased surface temperature, and reduced productivity.**

In our parameterization of SOM in the model, the soil hydraulic conductivity can be affected by SOM through the increase of the porosity. However, we didn't perform any sensitivity tests of deep soil moisture to the parameterization of soil hydraulic conductivity. We point out the importance of the soil water deficit at the beginning of the paragraph "In the root zone" of sub-section 7.3 "Soil moisture", Page 24 Line 16: "The soil water deficit is of primary importance during spring and summer in the high latitudes because of its potential impacts on the vegetation transpiration, leading to a surface temperature increase and a reduction in the productivity. Yet, a soil water comparison between GLEAM and ORCHIDEE-MICT is difficult because ..."

**p.26,l.30 Why is peak GPP overestimated? Please explain.**

The $CO_2$ fertilization seems indeed too important in ORCHIDEE. We have at least two leads to improve this behavior. Kuppel et al. (2012) used FLUXNET daily observations to optimize several photosynthetic parameters like the maximum carboxylation rate Vcmax. Given all the modifications brought to the model since this first FLUXNET optimization, mainly regarding the physics, it is probably time to recalibrate these photosynthetic parameters to get a more realistic GPP. Second, plants grown under elevated $CO_2$ show a photosynthetic downregulation (Sellers et al., 1996 ; Bounoua et al., 1999 ; Bounoua et al., 2010). This downregulation calibration is coded in the model to rectify Vcmax under different atmospheric $CO_2$ levels to empirically limit the $CO_2$ fertilization effect, but this option was not activated in these simulations.

- Kuppel, S., Peylin, P., Chevallier, F., Bacour, C., Maignan, F., and Richardson, A. D.: Constraining a global ecosystem model with multi-site eddy-covariance data, Biogeosciences, 9, 3757-3776, DOI 10.5194/bg-9-3757-2012, 2012.
- Sellers, P. J., Bounoua, L., Collatz, G. J., Randall, D. A., Dazlich, D. A., Los, S. O., Berry, J. A., Fung, I., Tucker, C. J., Field, C. B., and Jensen, T. G.: Comparison of radiative and physiological effects of doubled atmospheric co2 on climate, Science, 271, 1402-1406, DOI 10.1126/science.271.5254.1402, 1996.
- Bounoua, L., Collatz, G. J., Sellers, P. J., Randall, D. A., Dazlich, D. A., Los, S. O., Berry, J. A., Fung, I., Tucker, C. J., Field, C. B., and Jensen, T. G.: Interactions between vegetation and climate: Radiative and physiological effects of doubled atmospheric co2, Journal of Climate, 12, 309-324, Doi 10.1175/1520-0442(1999)012<0309:Ibvacr>2.0.Co;2, 1999.
- Bounoua, L., Hall, F. G., Sellers, P. J., Kumar, A., Collatz, G. J., Tucker, C. J., and Imhoff, M. L.: Quantifying the negative feedback of vegetation to greenhouse warming: A modeling approach, Geophysical Research Letters, 37, Artn L23701, Doi 10.1029/2010gl045338, 2010.

**p.27,l.4: NPP is underestimated (Fig 16) due to water stress and lack of nitrogen fertilisation. GPP, however, is overestimated, which should lead to an underestimation of CUE. However, CUE is overestimated, and this is explained with a lack of nutrient limitation ('too much' nutrients), which is inconsistent with the lack of nitrogen ('too little' nutrients) mentioned above. Please explain.**

GPP is overestimated at the regional scale largely for BONA, slightly for BOEU and is correct for BOAS (see Fig. 15), but GPP is underestimated over the Campioli sites (see Fig. 17), where the CUE is computed. We can thus infer that the Campioli database is not a representative sample of our domain, however this database offers the advantage of having collocated GPP and NPP observations, and thus access to realistic CUE estimates.

Nevertheless the reviewer is right, as we use the nutrients argument in opposite manners to explain a too low NPP and a too high CUE over the Campioli sites. We thus removed the following sentences in section 8.2.2 on Page 27 Line 6: "... or due to a lack of N-deposition combined with soil fertility effects in modeled NPP" and on Page 27 Lines 18-19 "This high CUE bias can be expected, as ORCHIDEE-MICT omits the effects of low nutrient availability on CUE (Vicca et al. 2012)". We now just stick to our water stress hypothesis, to explain these low NPP and GPP, resulting in a high CUE.

**p.28,l.30: The respiration could also originate from a moss/lichen layer which may show some activity at low temperatures and under snow.**

Yes. We added these sentences in the text:
- at Line 27 P.28: "...(ii) insufficient snow insulation of soils (See Fig. 7); and (iii) the lack of the carbon cycle of mosses and lichens which could have respiration under winter low temperatures (Atanasiu, 1971)."
- at Lines 29-31 p.28: "...improve the snow insulating, prescribe an organic layer of insulating topsoil (e.g. mosses, O-horizons observed in boreal forests, see O'Donnell et al., 2011) into the thermal module, or explicitly represent the moss/lichen plants including their carbon cycle and physical effects (Porada et al., 2016; Druel et al., 2017)."

**p.29,l.6: Biomass is significantly overestimated (see also Fig S5), and, in my opinion, the difference between climate forcing data sets cannot explain this easily: Precipitation in CRUNCEP is lower than in GSWP3, but biomass is higher, which seems counterintuitive. Could you please explain?**

The higher biomass by CRUNCEP than by GSWP3 could be partly explained by the higher GPP (Fig. 15). Indeed, precipitation in CRUNCEP is lower than in GSWP3; however, the specific air humidity in CRUNCEP during summer is higher than in GSWP3 (Fig. S14). A lower air humidity leads to a higher atmospheric vapour pressure deficit (VPD). ORCHIDEE being VPD-dependent, the stomatal conductance of the vegetation decreases to prevent excessive water loss when VPD is high, and consequently reduces photosynthetic rates. To address it, we add the following paragraph on Page 26, below Line 31: "Interestingly, comparing GPP forced by the two climate datasets shows higher values by CRUNCEP than by GSWP3 (Fig. 14), despite a generally lower precipitation in CRUNCEP (Fig. S17). This could be explained by the higher specific air humidity during summer in CRUNCEP than in GSWP3 (Fig. S14). A low air humidity increases the atmospheric vapor pressure deficit (VPD) and the leaf to air vapour pressure difference; plants then partially close the stomata to constrain a potentially fast transpiration (Oren et al., 1999; McAdam et al., 2015), which leads to reduced photosynthetic rate. The photosynthesis module in ORCHIDEE largely follows Yin and Struik (2009) in which stomata conductance decreases with an increasing VPD, thus is able to simulate a lower GPP under dry air conditions. A recent study (Novick et al., 2016) showed that, between the two factors that impact plant water stress, i.e. soil moisture supply and atmospheric demand for water (reflected by VPD), the latter limits evapotranspiration to a greater extent than the former in relatively wet forested ecosystems. In spite of its importance, the effect of VPD on vegetation productivity has been far less studied than soil water availability (Konings et al., 2017), warranting further investigations in both observations and land surface models."

Apart from a higher GPP, the higher biomass by CRUNCEP could also be because of the allocation scheme in ORCHIDEE. As detailed in Krinner et al., (2005), if LAI is above a PFT-specific maximum value (about 4 for boreal tree PFTs), carbon will not be allocated to leaves but to the sapwood which slowly converts to heartwood. Therefore, a higher LAI simulated by CRUNCEP leads to more carbon allocation to the wood for some areas, the turnover time of which is much longer than leaves. To address it, the paragraph on Page 29, Lines 13-14 is revised as: "Both model output and observation data are subject to the spatial uncertainties introduced by the use of satellite-derived land cover maps. We thus used the forest cover map as prescribed in the model for both data sets, and calculated latitudinal averages to compare with model results (Fig. S8). GSWP3-forced model output agrees well with observations averaged over the whole study region, while CRUNCEP-forced output overestimates biomass at all latitudes (Fig. S8a). For the sub-regions, the overestimation of biomass in BONA is consistent with that of GPP (Fig. 14), while biomass is more overestimated in BOEU compared with GPP, probably because of the lack of forest management and forest age structure for Europe. Comparing the two model results, CRUNCEP-forced biomass is much higher than GSWP3-forced biomass, which cannot solely be explained by the higher GPP by CRUNCEP (Fig. 14), indicating a bias in the allocation scheme in ORCHIDEE. As detailed in Krinner et al., (2005), the photosynthates are partitioned into leaves,

roots, sapwood, and carbohydrate reserve, dependent on soil moisture etc. If LAI is above a PFT-specific maximum value, carbon will not be allocated to leaves but to the sapwood, which slowly converts to heartwood. Therefore, a higher LAI forced by CRUNCEP leads to more carbon allocation to the wood for some areas, the turnover time of which is much longer than leaves. A new allocation scheme based on the pipe model was implemented in another branch of ORCHIDEE (Naudts et al., 2015), which provides a physiologically meaningful relationship between foliage, roots, and wood. This would be incorporated in ORCHIDEE-MICT in the future developments. Simulated total forest biomass for the whole domain is 95 PgC under GSWP3 forcing (165 PgC under CRUNCEP), close to estimates from forest inventory data in Pan et al. (2011) of 92.1 PgC. Somewhat lower estimates are derived by Avitabile et al. (2016) and Thurner et al. (2014), of 73 and 84 PgC, respectively."

**p.32,l.23: I think biomass compares well to observations only for GSWP3 forcing (Fig 21).**

Yes, you are right. This is now corrected in the text: "ORCHIDEE-MICT performs generally well for biomass with GSWP3 forcing, including the latitudinal profile, ..."

**p.33,l.18: The model already has a cold bias, so even lower soil temperatures would be required for a more realistic (higher) soil carbon content. Why do you not mention explicit simulation of cryoturbation as a potential missing process?**

Actually, we take into account the cryoturbation effect in ORCHIDEE-MICT using a diffusion term (see Eq. 2 in Section 4.1).

**supplement: Please indicate if CRUNCEP is subtracted from GSWP3 or vice versa.**

This is added now in the title of the section of supplementary Figures and in the caption of Figure S9

[revised manuscript text omitted]

Guimberteau et al.

**Soil thermal parameters**

| | $\theta_{sat}$ (or porosity) $(m^3\,m^{-3})$ | $\lambda_{dry}$ $(W\,m^{-1}\,K^{-1})$ | $\lambda_{solid}$ $(W\,m^{-1}\,K^{-1})$ | $c_{dry}$ $(10^6\,J\,K^{-1}\,m^{-3})$ |
|---|---|---|---|---|
| sand | 0.43 | 0.22 | 6.91 | 1.47 |
| loamy sand | 0.41 | 0.23 | 6.04 | 1.41 |
| sandy loam | 0.41 | 0.23 | 4.49 | 1.34 |
| silt loam | 0.45 | 0.20 | 2.80 | 1.27 |
| silt | 0.46 | 0.20 | 3.30 | 1.21 |
| loam | 0.43 | 0.22 | 3.43 | 1.21 |
| sandy clay loam | 0.39 | 0.25 | 4.49 | 1.18 |
| silty clay loam | 0.43 | 0.22 | 3.30 | 1.32 |
| clay loam | 0.41 | 0.23 | 3.21 | 1.23 |
| sandy clay | 0.38 | 0.26 | 4.03 | 1.18 |
| silty clay | 0.36 | 0.28 | 3.30 | 1.15 |
| clay | 0.38 | 0.26 | 2.80 | 1.09 |
| pure organic soil | 0.92 | 0.05 | 0.25 | 2.50 |

$\theta_{sat}$: volumetric moisture content at saturation (porosity); $\lambda_{dry}$ : dry soil thermal conductivity, calculated by the empirical equation:

$\lambda_{dry} = \frac{0.135\rho+64.7}{2700-0.947\rho}$ with $\rho = 2700\,(1-\theta_{sat})$

$\lambda_{solid}$ : thermal conductivity of soil solids; $c_{dry}$ : dry soil heat capacity.

[revised manuscript text omitted]

**Maximum monthly soil temperature**

[Figure]

**(a) GSWP3**

0.2 m  0.8 m  1.6 m  3.2 m

**(b) CRUNCEP**

0.2 m  0.8 m  1.6 m  3.2 m

-3 -1 1 3 5 7 9 11 13 15 17 19 21
Soil temperature (°C)

**Figure S4.** Maximum monthly soil temperature at different depths in **(a)** GSWP3 and **(b)** CRUNCEP-forced simulation (background maps), compared to the site observations (color filled circles), averaged over the period 1981-2000.

**Active layer thickness**

[Figure]

**(a) Beer et al., 2013**

**(b) GSWP3**

**(c) CRUNCEP**

active layer thickness (m)

**Figure S3****. (a)** Active layer thickness (ALT) of Yakutia, East Siberia, upscaled based on the map of landscapes and permafrost conditions in Yakutia by Beer et al. (2013). **(b,c)** Modeled ALT for the same region with GSWP3 and CRUNCEP, averaged over the period 1960-1987 to be consistent with Beer et al. (2013).

[Figure]

○ grid cells taken from Fig. 8b that correspond to the CALM sites
○ a subset of the black circles
○ same as the blue circles but use the site-specific organic layer thicknesses

**Figure S6. (a)** Scatter plot of modeled ALT forced by CRUNCEP compared with observed ALT from the CALM network, averaged over the period 1990-2007. The black circles represent the grid cells taken from the regional simulation (shown in Fig. 8b). The blue circles represent a subset of the CALM sites for which we performed additional runs using site-specific organic layer thicknesses, with the result shown by the red circles. **(b)** Illustration for the difference of the additional site simulations. Each grey arrow connects the same site, pointing from the blue circles using soil carbon content from NCSCD (upscaled at 1° by 1° resolution) to calculate soil physical properties, to the red circles using the organic layer thickness provided by the sites. See text for further information.

**Albedo**

[Figure]

**(a)** **(b)** **(c)**

**(d)** **(e)** **(f)**

**Figure S47.** Albedo **(left)** observed (ESA-GlobAlbedo) for the months of **(Upper panel)** February and **(Lower panel)** August, averaged over the period 1998-2011. Albedo bias with **(middle)** GSWP3 and **(right)** CRUNCEP climate forcing.

In February, when the snow cover is maximum, the plots show an overall underestimation of the modeled albedo up to 0.25 in the snow-covered regions particularly with CRUNCEP forcing and a slight overestimation elsewhere (Fig. S47). These differences may be partly explained by the SWE underestimation already highlighted and also by weaknesses in the parametrization of the albedo, especially over high vegetation (forested areas in North America, central Europe and Siberia). In these forested regions, the albedo seems to be overestimated in presence of snow. In August, the comparison shows very similar values in the modeled product whatever the atmospheric forcing used and the differences with the observations are strongly related to the land cover map. Grasslands in the northern regions presents underestimated albedo values up to 0.1, whereas the albedo is overestimated slightly elsewhere with a bias of about +0.02. These discrepancies may be explained by the LAI summer lag already discussed which leads to an overestimation of the vegetation cover in August and consequently an underestimation of the albedo (since the albedo of bare soils is generally larger than those of vegetation).

**Latitudinal distribution of fire carbon emissions**

[Figure]

**Figure S58.** Latitudinal distribution of total annual forest biomass carbon stocks (PgC degree$^{-1}$) in the GSWP3 and CRUNCEP-forced simulations, compared to observations by Avitabile et al. (2016) and Thurner et al. (2014) for **(a)** the whole model domain. High-latitude sub-regions of **(b)** BONA, **(c)** BOEU and **(d)** BOAS have been extended to 30°N here compared to their shapes in Fig. 2a.

**Seasonal cycle of burned area and fire carbon emissions**

[Figure]

**Figure S69.** Monthly mean seasonal **(a-c)** fire burned area (Mha month$^{-1}$) and **(d-f)** carbon emissions to the atmosphere (TgC month$^{-1}$) from the GFED4s dataset and the two simulations, averaged over the period 1997–2007. For carbon emissions, cropland and natural fires are represented separately.

**LAI and GPP seasonality between deciduous and evergreen forests**

[Figure]

**Figure S710.** Monthly mean seasonal **(first and third panels)** GPP ($gC\,m^{-2}\,day^{-1}$) and **(second and fourth panels)** LAI (-) for deciduous (light green) and conifer forests (dark green), over the three high-latitude sub-regions (shown in Fig. 2a), averaged over the period 2000-2007. The lower panel shows absolute values and the upper one values scaled for a better comparison.

**Land carbon fluxes and pools modeled and derived from observations**

[Figure]

**Figure S811.** Annual mean land carbon fluxes (PgC yr$^{-1}$) and pool sizes (PgC) over the three high-latitude sub-regions (shown in Fig. 2a) **(a)** BONA, **(b)** BOEU and **(c)** BOAS, averaged over the period 2000-2007. The red numbers are the model results forced by (left) GSWP3 and (right) CRUNCEP. The black numbers are from observation-based estimates used in the text : NPP from MODIS (NTSG), fire emissions from (left) GFED4s and (right) GFAS datasets, harvest fluxes include crop harvest and wood product decay in the model (for simplicity the change in wood product pools is not represented), NEE from the two atmospheric inversions (left) Jena CarboScope and (right) CAMS, biomass from (left) Avitabile et al. (2016) and (right) Thurner et al. (2014), litter from Pan et al. (2011), soil carbon from NCSCD (Hugelius et al., 2013) in permafrost regions and HWSD in non-permafrost regions.

**Differences of atmospheric variables between GSWP3 and CRUNCEP forcings (GSWP3 minus CRUNCEP)**

[Figure]

**Figure S912.** Difference in monthly air temperature (°C) between GSWP3 and CRUNCEP (GSWP3 minus CRUNCEP), averaged over the period 1981-2007.

[Figure]

**Figure S13.** Same as Fig. S12 but for snowfall ($\text{mm yr}^{-1}$)

[Figure]

**Figure S14.** Same as Fig. S12 but for total precipitation (mm yr$^{-1}$)

[Figure]

**Figure S15.** Same as Fig. S12 but for downward shortwave radiation (W m$^{-2}$)

[Figure]

**Figure S136.** Same as Fig. S912 but for downward longwave radiation ($\mathrm{W\,m}^{-2}$)

[Figure]

**Figure S14**7**.** Same as Fig. S9**12** but for specific humidity $(\mathrm{g\,kg^{-1}})$

[Figure]

**Figure S15.** Same as Fig. S9 but for wind speed at 10 m (m s$^{-1}$)

[Figure]

**Figure S169.** Same as Fig. S912 but for wind speed at 2 m in ORCHIDEE outputs ($\mathrm{m\,s^{-1}}$)

---

## Author Response (AR2)

**ORCHIDEE-MICT** (v8.4.1), a land surface model for the high-latitudes: model description and validation**

Matthieu Guimberteau1\*, Dan Zhu1\*, Fabienne Maignan1, Ye Huang1, Chao Yue1, Sarah Dantec-Nédélec1, Catherine Ottlé1, Albert Jornet-Puig1, Ana Bastos1, Pierre Laurent1, Daniel Goll1, Simon Bowring1, Jinfeng Chang2, Bertrand Guenet1, Marwa Tifafi1, Shushi Peng3, Gerhard Krinner4, Agnès Ducharne5, Fuxing Wang6, Tao Wang7,8, Xuhui Wang1,9, Yilong Wang1, Zun Yin1, Ronny Lauerwald10,1,11, Emilie Joetzjer1,12, Chunjing Qiu1, Hyungjun Kim13 and Philippe Ciais1

1 Laboratoire des Sciences du Climat et de l'Environnement, LSCE/IPSL, CEA - CNRS - UVSQ, Université Paris-Saclay, 91191 Gif-sur-Yvette, France

2 Sorbonne Universités (UPMC), CNRS-IRD-MNHN, LOCEAN/IPSL, 4 place Jussieu, 75005 Paris, France

3 Sino-French Institute for Earth System Science, College of Urban and Environmental Sciences, Peking University, Beijing 100871, China

4 CNRS, Univ. Grenoble Alpes, Institut des Géosciences de l'Environnement (IGE), 38000 Grenoble, France

5 UMR 7619 METIS, Sorbonne Universités, UPMC, CNRS, EPHE, 4 place Jussieu, 75005 Paris, France

6 Laboratoire de Météorologie Dynamique, Ecole Polytechnique, F 91128 Palaiseau, France

7 Key Laboratory of Alpine Ecology and Biodiversity, Institute of Tibetan Plateau Research, Chinese Academy of Sciences, Beijing 100085, China

8 CAS Center for Excellence in Tibetan Plateau Earth Sciences, Chinese Academy of Sciences, Beijing 100085, China

9 Laboratoire de Météorologie Dynamique, Université Pierre et Marie Curie, 75005 Paris, France

10 Université Libre de Bruxelles, Belgium

11 University of Exeter, Exeter, United Kingdom

12 CNRS, Université Paul Sabatier, ENFA; UMR5174 EDB (Laboratoire Evolution et Diversité Biologique), 118 route de Narbonne, Toulouse F-31062, France

13 Institute of Industrial Science, The University of Tokyo, Tokyo, Japan

\* These authors contributed equally to this work

Correspondence to: matthieu.guimberteau@lsce.ipsl.fr and dan.zhu@lsce.ipsl.fr

**Abstract.** The high-latitude regions of the northern hemisphere are a nexus for the interaction between land surface physical properties and their exchange of carbon and energy with the atmosphere. At these latitudes, two carbon pools of planetary significance – those of the permanently frozen soils (permafrost), and of the great expanse of boreal forest – are vulnerable to destabilization in the face of currently observed climatic warming, the speed and intensity of which are expected to in-

5

crease with time. Improved projections of future Arctic and boreal ecosystem transformation require improved land surface models that integrate processes specific to these cold biomes. To this end, this study lays out relevant new parameterizations in the ORCHIDEE-MICT land surface model. These describe the interactions between soil carbon, soil temperature and hydrology, and their resulting feedbacks on water and CO2 fluxes, in addition to a recently-developed fire module. Outputs from

ORCHIDEE-MICT, when forced by two climate input data sets, are extensively evaluated against: (i) temperature gradients between the atmosphere and deep soils; (ii) the hydrological components comprising the water balance of the largest highlatitude basins, and (iii) CO2 flux and carbon stock observations. The model performance is good with respect to empirical data, despite a simulated excessive plant water stress and a positive land surface temperature bias. In addition, acute model

sensitivity to the choice of input forcing data suggests that the calibration of model parameters is strongly forcing-dependent. 5 Overall, we suggest that this new model design is at the forefront of current efforts to reliably estimate future perturbations to the high-latitude terrestrial environment.

**Contents**

[revised manuscript text omitted]

(3)

25 with:

$$\lambda_{i,\text{sat}} = \lambda_{i,\text{solid}}^{\left(1 - \theta_{i,\text{sat}}\right)} \lambda_{\text{liq}}^{\left(\theta_{i,\text{sat}} \frac{\theta_{i,\text{liq}}}{\theta_{i,\text{liq}} + \theta_{i,\text{ice}}}\right)} \lambda_{\text{ice}}^{\left(\theta_{i,\text{sat}} \frac{\theta_{i,\text{ice}}}{\theta_{i,\text{liq}} + \theta_{i,\text{ice}}}\right)}$$

(4)

(5)

where  $\lambda_{i,sat}$  and  $\lambda_{i,dry}$  are saturated and dry thermal conductivities for layer i;  $\lambda_{liq}$  and  $\lambda_{ice}$  are thermal conductivities of liquid water and ice, equaling to 0.57 and 2.2 respectively (W m-1 K-1);  $\lambda_{i,solid}$  is thermal conductivity of soil solids (see Table S1);  $c_{liq}$  and  $c_{ice}$  are heat capacities of liquid water and ice, equaling to 4.18 106 and 2.11 106 respectively (J K-1m-3);  $c_{dry}$  is dry soil heat capacity depending on soil texture;  $\theta_{i,sat}$  is volumetric moisture content at saturation (porosity), and it varies with soil textures;  $\theta_{i,liq}$  and  $\theta_{i,ice}$  are prognostic volumetric liquid water and ice contents (m3 m-3) that are computed by the soil hydrology model; Kei is the Kersten number given by:

For unfrozen soils:

$$Ke_{i} = \begin{cases} \log_{10}(S_{r}) + 1 & \\ 0.7\log_{10}(S_{r}) + 1 & if \\ 0 & \\ 0 & \\ S_{r} \le 0.05 \end{cases} \begin{pmatrix} S_{r} > 0.1 \\ 0.05 < S_{r} \le 0.1 \\ S_{r} \le 0.05 \end{pmatrix}$$
(6)

10 with:

5

$$S_{r} = \frac{\theta_{i}}{\theta_{i,sat}}$$
(7)

For frozen soils:

$$Ke_i = S_r \tag{8}$$

where  $S_r$  is the degree of saturation.

- 15
  - To account for the impacts of organic carbon on soil thermal properties in ORCHIDEE-MICT, we follow Lawrence and Slater (2008) in assuming that soil physical properties are weighted averages of mineral soil (as the default values in standard ORCHIDEE) and pure organic soil, with the organic soil fraction  $f_{i,soc}$  calculated as:

$$f_{i,soc} = \min\left(1, \frac{\rho_{i,soc}}{\rho_{soc,max}}\right)$$
(9)

where  $\rho_{i,soc}$  is the carbon content for layer i (kgCm-3), derived from observation-based soil organic carbon map from 20 NCSCD (Hugelius et al., 2013) in permafrost regions and from HWSD (FAO, 2012) in non-permafrost regions, after linear

vertical interpolation from their original soil horizons to fit ORCHIDEE-MICT vertical layers;  $\rho_{soc,max}$  equals to 130 kgC m-3, a typical soil carbon density of peat (Lawrence and Slater, 2008). Therefore, the red-colored parameters in Eqs. 3-7 are calculated as:

$$P_{i} = (1 - f_{i,soc}) P_{mineral} + f_{i,soc} P_{soc}$$
(10)

where Pi represents different properties λi,dry, λi,solid, ci,dry, and θi,sat. The values of Pmineral for each soil texture and Psoc 
[revised manuscript text omitted]